# WHEN IS TASK VECTOR *PROVABLY* EFFECTIVE FOR MODEL EDITING? A GENERALIZATION ANALYSIS OF NONLINEAR TRANSFORMERS

**Hongkang Li**[1], **Yihua Zhang**[2], **Shuai Zhang**[3], **Pin-Yu Chen**[4], **Sijia Liu**[2,4], **Meng Wang**[1,*]
[1]Rensselaer Polytechnic Institute, [2]Michigan State University, [3]New Jersey Institute of Technology, [4]IBM Research

## ABSTRACT

Task arithmetic refers to editing the pre-trained model by adding a weighted sum of task vectors, each of which is the weight update from the pre-trained model to fine-tuned models for certain tasks. This approach recently gained attention as a computationally efficient inference method for model editing, e.g., multi-task learning, forgetting, and out-of-domain generalization capabilities. However, the theoretical understanding of why task vectors can execute various conceptual operations remains limited, due to the highly non-convexity of training Transformer-based models. To the best of our knowledge, this paper provides the first theoretical characterization of the generalization guarantees of task vector methods on nonlinear Transformers. We consider a conceptual learning setting, where each task is a binary classification problem based on a discriminative pattern. We theoretically prove the effectiveness of task addition in simultaneously learning a set of irrelevant or aligned tasks, as well as the success of task negation in unlearning one task from irrelevant or contradictory tasks. Moreover, we prove the proper selection of linear coefficients for task arithmetic to achieve guaranteed generalization to out-of-domain tasks. All of our theoretical results hold for both dense-weight parameters and their low-rank approximations. Although established in a conceptual setting, our theoretical findings were validated on a practical machine unlearning task using the large language model Phi-1.5 (1.3B).

## 1 INTRODUCTION

Large pre-trained models (Chowdhery et al., 2022; Touvron et al., 2023; Achiam et al., 2023) have recently served as a foundational module in deep learning systems. Under the pre-training-and-fine-tuning paradigm, although the traditional and straightforward full-parameter fine-tuning can demonstrate superior performance in downstream tasks, its immense computational and memory costs have become a serious practical issue. Consequently, many Parameter-Efficient Fine-Tuning (PEFT) methods (Li & Liang, 2021; Hu et al., 2022; Jia et al., 2022; Wei et al., 2022b;a) have been proposed to address this concern. Among them, the recent *task vector* approach receives increasing attention (Ilharco et al., 2022a; Ortiz-Jimenez et al., 2023; Hendel et al., 2023; Todd et al., 2024).

The task vector approach first fine-tunes a pre-trained model on several simpler tasks to obtain task vectors, which represent the weight differences between the fine-tuned models and the pre-trained model. To handle more complex tasks, a proper model can be edited by adding a linear combination of these task vectors to the pre-trained model. Since this approach only requires determining the appropriate arithmetic hyperparameters, with no need for further fine-tuning on complicated tasks, the task vector method offers a significant efficiency advantage and is particularly effective when adapting to a wide range of downstream tasks. Empirical evidence shows that adding multiple task vectors can improve the model's performance on corresponding tasks, while subtracting certain task vectors allows the model to forget associated tasks. A proper linear combination of task vectors can even enable the model to generalize on an out-of-domain task that has an analogous relationship with the given task vectors, without needing labeled data. Additionally, it has been found that using low-

---

*Corresponding author. Email: wangm7@rpi.edu.

rank and/or sparse task vectors can further improve efficiency while maintaining the performance (Yadav et al., 2023; Chitale et al., 2023; Yu et al., 2024; He et al., 2025).

Despite empirical successes, theoretical analysis of task vectors is less investigated. In particular, we ask the following question:

*When and why can the task vector approach perform well in multi-task learning, unlearning, and out-of-domain generalization successfully and efficiently?*

Some related theoretical works focus on analyzing the performance of machine unlearning from a purely optimization perspective (Ginart et al., 2019; Neel et al., 2021; Guo et al., 2020; Mu & Klabjan, 2024). However, these analyses do not apply to Transformer-based neural networks, which are key components of large pre-trained models. Moreover, these works cannot be extended to study multi-task learning or out-of-domain generalization to new tasks. Frankle et al. (2020) proposes the concept of linear mode connectivity, suggesting that there exists a small-loss connected region in the loss landscape of the model, thereby demonstrating that linear interpolation between models can yield good performance. The most relevant workto this paper is (Ortiz-Jimenez et al., 2023), which uses the Neural Tangent Kernel (NTK) framework (Jacot et al., 2018) to study neural networks as linearized models under specific assumptions, to justify the use of linear arithmetic on task vectors for targeted model editing. However, this work does not have generalization guarantees and cannot explain the success of task vectors in nonlinear models without NTK assumptions.

## 1.1 MAJOR CONTRIBUTIONS

To the best of our knowledge, this work is the first theoretical generalization analysis of task arithmetic on a nonlinear Transformer model for multi-task learning, unlearning, and out-of-domain generalization. Focusing on binary classification tasks, we provide a quantitative analysis of the dependence of the task arithmetic effect on arithmetic hyperparameters. Although our analysis is centered on a simplified single-head and one-layer nonlinear Transformer, our theoretical insights are validated on practical architectures. Our major contributions include:

1. **A fine-grained feature-learning analysis of the effectiveness of task addition and negation.** We consider a data model in which binary labels are determined by the majority of discriminative tokens, rather than their opposing discriminative counterparts, while other tokens do not affect the labels. We begin by analyzing the learning dynamics of fine-tuning a Transformer and characterize the properties of the resulting task vectors. Next, we provide sufficient conditions on the arithmetic hyperparameters for the task vector approach to be successful. We prove that task addition is effective for multi-task learning when the tasks are either irrelevant or aligned. Aligned tasks are those where solving one task contributes positively to solving the other. In contrast, task negation is provably successful for unlearning tasks that are either irrelevant or contradictory. Contradictory tasks are defined as those where improving performance on one task harms the performance of the other.

2. **The first provable out-of-domain generalization guarantees through task arithmetic**. Focusing on task vectors representing a set of irrelevant tasks, we prove a linear combination of these task vectors can generalize to a wide range of new tasks by properly selecting the arithmetic coefficients. Additionally, we characterize the range of suitable arithmetic coefficients sufficient for successful generalization. This is the first theoretical justification of task vectors' ability to adapt to new tasks.

3. **Theoretical justification of low-rank approximation and magnitude-based pruning for task vectors.** We construct low-rank and sparse approximations to task vectors and prove that the generalization guarantees are minimally affected by these approximations. This provides the first theoretical support for the practice of using low-rank and sparse approximations to task vectors in order to reduce computational complexity.

## 1.2 RELATED WORKS

**Weight interpolation technique**. Weight interpolation or model merging (Matena & Raffel, 2022; Ilharco et al., 2022b; Yadav et al., 2023; Yu et al., 2024; He et al., 2025) refers to the practice of linearly interpolating weights of multiple models, where these models may be fine-tuned from different downstream tasks or using different hyperparameters (model soups (Wortsman et al., 2022a)). Weight interpolation is empirically observed to be able to guide the model towards wider optima (Izmailov et al., 2018; Frankle et al., 2020) and better generalization in both single-task performance and multi-task ablities, even surpassing fine-tuning methods in some cases (Rame et al.,

2022; Wortsman et al., 2022b; Ramé et al., 2023). Task arithmetic can be viewed as a special type of weight interpolation, where linear operations are performed on task vectors.

**Feature learning analysis for Transformers**. Several recent works study the optimization and generalization analysis of Transformers following the feature learning framework, which describes how neural networks gradually focus on important features while discarding unimportant features during training. Jelassi et al. (2022); Li et al. (2023e); Oymak et al. (2023); Ildiz et al. (2024); Nichani et al. (2024); Chen et al. (2024); Li et al. (2023a; 2024c; 2023b); Huang et al. (2024); Luo et al. (2024) study the generalization of one-layer Transformers on different data models such as spatial association, semantic/contextual structure, causal structure/Markov Chain of data, and the majority voting of tokens in the data. However, no discussion was provided for merged models.

**Theoretical study of PEFT methods**. These are recent theoretical analyses on other PEFT methods. For example, in-context learning is analyzed from the perspective of expressive power (Bai et al., 2023; Akyürek et al., 2023; Von Oswald et al., 2023), the training dynamics or generalization (Xie et al., 2021; Zhang et al., 2023a; Li et al., 2023c; 2024a;b; Huang et al., 2023). Some other works focus on prompt engineering with a tunable prompt (Wei et al., 2021; Oymak et al., 2023; Zhang et al., 2024). Another line of work theoretically investigates the low-rank adaptation in terms of the implicit bias of the optimization process (Damian et al., 2022; Abbe et al., 2022; 2023; Boix-Adsera et al., 2023; Jang et al., 2024; Li et al., 2024d) or model pruning with generalization analysis (Zhang et al., 2021; Yang & Wang, 2023; Yang et al., 2023; Zhang et al., 2023b; Li et al., 2024a). However, none of these works involve the task vector method or related approaches.

## 2 TASK VECTOR: DEFINITION AND OBSERVATIONS

### 2.1 PRELIMINARIES

Let $f : \mathcal{X} \times \Theta \to \mathcal{Y}$ be a neural network that maps inputs $\boldsymbol{X} \in \mathcal{X}$ to labels $\boldsymbol{y} \in \mathcal{Y}$ with $\Psi \in \Theta$ as the model parameters. Denote $\Psi^{(0)}$ as the pre-trained model and $\Psi^*_{\mathcal{T}}$ as the fine-tuned model on a given task $\mathcal{T}$.

**Definition 1.** *(Task Vector) The task vector $\Delta\Psi_{\mathcal{T}}$ for the task $\mathcal{T}$ is computed as the element-wise difference between the pre-trained and fine-tuned weights, i.e., $\Delta\Psi_{\mathcal{T}} = \Psi^*_{\mathcal{T}} - \Psi^{(0)}$.*

**Task Arithmetic and Generalization**. Given the pre-trained model $\Psi^{(0)}$ and a set of task vectors $\{\Delta\Psi_{\mathcal{T}_i}\}_{i \in \mathcal{V}}$ on tasks $\{\mathcal{T}_i\}_{i \in \mathcal{V}}$, one can construct a merged model $\Psi = \Psi^{(0)} + \sum_{i \in \mathcal{V}} \lambda_i \Delta\Psi_{\mathcal{T}_i}$ for inference on downstream tasks, where $\lambda_i \in \mathbb{R}$ are arithmetic hyperparameters. Denote $\ell(\boldsymbol{X}, y; \Psi)$ as the loss function for the input $\boldsymbol{X} \in \mathcal{X}$, output $y \in \mathcal{Y}$, and the model $\Psi \in \Theta$. Hence, the **generalization error** on the task $\mathcal{T}'$ with data $(\boldsymbol{X}, y) \sim \mathcal{D}_{\mathcal{T}'}$ is defined as

$$\mathbb{E}_{(\boldsymbol{X}, y) \sim \mathcal{D}_{\mathcal{T}'}} \ell(\boldsymbol{X}, y; \Psi). \tag{1}$$

Existing works (Ilharco et al., 2022a; Ortiz-Jimenez et al., 2023) conclude that by controlling $\lambda_i$, the merged model $\Psi$ can generalize across different tasks. Specifically, *adding* several $\Delta\Psi_{\mathcal{T}_i}$ via making $\lambda_i > 0$, $i \in \mathcal{V}_A \subset \mathcal{V}$, leads to a model that exhibits desired performance on multiple tasks from $\mathcal{V}_A$. Such a successful *multi-task learning* result can be mathematically represented as

$$\mathbb{E}_{(\boldsymbol{X}, y) \sim \mathcal{D}_{\mathcal{T}_i}} \ell(\boldsymbol{X}, y; \Psi) \leq \Theta(\epsilon), \ \forall i \in \mathcal{V}_A. \tag{2}$$

Meanwhile, *negating* $\Delta\Psi_{\mathcal{T}_i}$ with $\lambda_i < 0$, $i \in \mathcal{V}_N \subset \mathcal{V}$, results in a *machine unlearning* model that performs poorly on $\mathcal{V}_N$ but roughly retains the accuracy on $\mathcal{V} \backslash \mathcal{V}_N$, i.e.,

$$\mathbb{E}_{(\boldsymbol{X}, y) \sim \mathcal{D}_{\mathcal{T}_i}} \ell(\boldsymbol{X}, y; \Psi) \geq \Theta(1), \ \mathbb{E}_{(\boldsymbol{X}, y) \sim \mathcal{D}_{\mathcal{T}_j}} \ell(\boldsymbol{X}, y; \Psi) \leq \Theta(\epsilon), \ \forall i \in \mathcal{V}_N, \forall j \in \mathcal{V} \backslash \mathcal{V}_N. \tag{3}$$

Moreover, task arithmetic is empirically (Ilharco et al., 2022a) shown to produce a model $\Psi = \Psi^{(0)} + \lambda \cdot \Delta\Psi_{\mathcal{T}'}$ that performs well on task analogy, in the form that "the target out-of-domain task $\mathcal{T}'(\notin \mathcal{V})$ is to $\mathcal{T}_A$ as $\mathcal{T}_B$ is to $\mathcal{T}_C$," by constructing a task vector $\Delta\Psi_{\mathcal{T}'} = \Delta\Psi_{\mathcal{T}_A} + (\Delta\Psi_{\mathcal{T}_B} - \Delta\Psi_{\mathcal{T}_C})$.

### 2.2 EMPIRICAL OBSERVATIONS

Note that experiments in (Ilharco et al., 2022a) only summarize the empirical findings when tasks are almost "orthogonal" to each other, while non-orthogonal cases are less explored. Therefore, in Table 1, we further construct binary classification tasks on the parity of digits of Colored-MNIST

(Arjovsky et al., 2019; Chapel et al., 2020). We control the colors of digits to generate a pair of two datasets so that the parity classification tasks on different pairs of datasets are conceptually "irrelevant," "aligned," or "contradictory" to each other, respectively.

For irrelevant tasks, odd and even digits are highly correlated with red and green colors in one dataset but independent of colors in the other. In aligned tasks, the odd and even digits are correlated with red and green colors in both datasets. In contradictory tasks, the color-parity correspondence is the opposite in the two datasets. Let $\mathcal{T}_1$ and $\mathcal{T}_2$ denote the parity classification task on two different datasets. $\Psi = \Psi^{(0)} + \Delta\Psi_{\mathcal{T}_1} + \lambda\Delta\Psi_{\mathcal{T}_2}$ is used to evaluate the performance of $\mathcal{T}_1$ and $\mathcal{T}_2$.

A key finding from Table 1 is that **the task vector method performs quite differently with different task correlations**. To be concrete, given $\Delta\Psi_{\mathcal{T}_1}$ and $\Delta\Psi_{\mathcal{T}_2}$ for aligned tasks, the merged model $\Psi$ can acquire strong multi-task learning abilities but have poor unlearning capabilities. The conclusion is exactly opposite for contradictory tasks. For irrelevant tasks, using task arithmetic can result in good performance in both unlearning and multi-task learning. A question arises, i.e.,

> *(Q1) How does task correlation quantitatively affect the performance of task arithmetic in multi-task learning and unlearning?*

|  | "Irrelevant" Tasks | | "Aligned" Tasks | | "Contradictory" Tasks | |
|---|---|---|---|---|---|---|
|  | Multi-Task | Unlearning | Multi-Task | Unlearning | Multi-Task | Unlearning |
| Best $\lambda$ | 1.4 | -0.6 | 0.2 | 0.0 | 0.6 | -1.0 |
| $\mathcal{T}_1$ Acc | 91.83 (-3.06) | 95.02 (-0.56) | 95.62 (0.00) | 95.20 (-0.42) | 79.54 (-16.70) | 94.21 (-0.61) |
| $\mathcal{T}_2$ Acc | 88.40 (-5.65) | 50.34 (-45.24) | 92.46 (-3.23) | 90.51 (-5.18) | 62.52 (-33.72) | 4.97 (-89.85) |

Table 1: Test accuracy (%) of $\Psi = \Psi^{(0)} + \Delta\Psi_{\mathcal{T}_1} + \lambda\Delta\Psi_{\mathcal{T}_2}$ on task $\mathcal{T}_1$ and $\mathcal{T}_2$ with $\lambda \in \{-1, -0.8, -0.6, \cdots, 2\}$. Multi-task learning aims to achieve good performance on both tasks, while unlearning is to decrease the accuracy on $\mathcal{T}_2$ but maintain the accuracy on $\mathcal{T}_1$. The best $\lambda$ is selected based on the largest accuracy summation (or gap) of $\mathcal{T}_1$ and $\mathcal{T}_2$ for multi-task learning (or unlearning). The accuracy gap (%) using $\Psi$ to the fine-tuned models $\Psi^*_{\mathcal{T}_1}$ or $\Psi^*_{\mathcal{T}_2}$ is reported in the bracket.

We then explore the use of task arithmetic with two tasks $\mathcal{T}_1$ and $\mathcal{T}_2$ for an out-of-domain task $\mathcal{T}'$. We construct tasks and data with Colored-MNIST, where we make $\mathcal{T}'$ more aligned with $\mathcal{T}_1$ and contradictory to $\mathcal{T}_2$. This is a new out-of-domain setting different from task analogies in (Ilharco et al., 2022a). Table 2 indicates that **the optimal $\lambda_1$ and $\lambda_2$ results in a testing performance better than using any separately trained model $\Psi^*_{\mathcal{T}_1}$ or $\Psi^*_{\mathcal{T}_2}$.** This implies that task arithmetic is powerful in domain generalization and can be extended to more general scenarios beyond analogous tasks. Hence, another question occurs, i.e.,

> *(Q2) Why do the arithmetic operations of task vectors perform well for out-of-domain generalization, and how to choose the arithmetic hyperparameter $\lambda_i$ for a desired performance?*

|  | Fine-Tuning | $\Psi^*_{\mathcal{T}_1}$ | $\Psi^*_{\mathcal{T}_2}$ | Searching $\lambda_1, \lambda_2$ in $[-2, 3]$ |
|---|---|---|---|---|
| $(\lambda_1, \lambda_2)$ | N/A | $(1, 0)$ | $(0, 1)$ | $(1.2, -0.6)$ |
| $\mathcal{T}'$ Acc | 92.21 | 88.10 | 45.06 | **91.74** |

Table 2: Comparison between the test accuracy (%) by different methods with $\Delta\Psi_{\mathcal{T}_1}$ and $\Delta\Psi_{\mathcal{T}_2}$. Searching $\lambda_1$ and $\lambda_2$ refers to evaluating $\Psi = \Psi^{(0)} + \lambda_1\Delta\Psi_{\mathcal{T}_1} + \lambda_2\Delta\Psi_{\mathcal{T}_2}$ on $\mathcal{T}'$ with $\lambda_1, \lambda_2 \in \{-2, -1.8, -1.6, \cdots, 3\}$.

## 3 A DEEP DIVE INTO TASK VECTORS

We first summarize the main insights in Section 3.1. Section 3.2 introduces the mathematical formulation of data and model. Sections 3.3 and 3.4 present the formal theoretical results on task arithmetic for multi-task learning, unlearning, and out-of-domain generalization. Section 3.5 theoretically proves the existence of a low-rank approximation or a sparse version of task vectors to maintain the performance.

### 3.1 MAIN THEORETICAL INSIGHTS

We focus on a set of binary classification tasks, where the labels in each task are determined by the majority between the *discriminative* tokens versus their opposite tokens in each data. This follows

the theoretical setting in (Cao et al., 2022; Kou et al., 2023; Li et al., 2023a; 2024c). We consider one-layer single-head Transformers. Our major takeaways are:

**P1. Quantitative Analysis of Multi-Task Learning and Unlearning via Task Addition and Negation.** Let $\alpha$ represent the correlations between two tasks $\mathcal{T}_1$ and $\mathcal{T}_2$, where positive, negative, and zero values correspond to aligned, contradictory, and irrelevant tasks, respectively. We prove that the merged model, $\Psi = \Psi^{(0)} + \Delta\Psi_{\mathcal{T}_1} + \lambda\Delta\Psi_{\mathcal{T}_2}$, is successful for multi-task learning if $\lambda \geq 1 - \alpha + \beta$ for some small constant $\beta$. Moreover, the merged model is successful in unlearning $\mathcal{T}_2$ if $\lambda \leq 0$ for irrelevant tasks or if $\lambda \in [-\Theta(\alpha^{-2}), O(\alpha^{-1})]$ for contradictory tasks.

**P2. Successful Out-of-domain Generalization through Task Arithmetic.** Given the correlation $\gamma_i$ between each existing task $\mathcal{T}_i$ and the target task $\mathcal{T}'$, we prove that as long as not all $\mathcal{T}_i$ are irrelevant to $\mathcal{T}'$, we can achieve a desired out-of-domain generalization on $\mathcal{T}'$ using task arithmetic. We explicitly quantify the arithmetic hyperparameter as functions of $\gamma_i$'s.

**P3. Low-rank Approximation and Magnitude-Based Pruning Preserves the Model Editing Performance.** We provide the first theoretical generalization guarantees for the practical techniques of low-rank approximation and task vector sparsity that reduce computation. Focusing on binary classification tasks based on discriminative patterns, we demonstrate that both sparsification of task vectors in the MLP layer (by removing rows with small magnitudes) and low-rank approximations of task vectors offer guaranteed generalization through task arithmetic.

## 3.2 PROBLEM FORMULATION

Suppose that data $\boldsymbol{X} = (\boldsymbol{x}_1, \boldsymbol{x}_2, \cdots, \boldsymbol{x}_P) \in \mathbb{R}^{d \times P}$ contains $P$ tokens, where each token is $d$-dimensional and $\|\boldsymbol{x}_i\| = 1$ for $i \in [P]$. The label $y \in \{+1, -1\}$ is a scalar. We consider the **learning model** as a single-head one-layer Transformer with one self-attention layer and one two-layer perceptron, which is mathematically written as

$$f(\boldsymbol{X}; \Psi) = \frac{1}{P}\sum_{l=1}^{P} \boldsymbol{a}_{(l)}^{\top} \text{Relu}(\boldsymbol{W}_O \sum_{s=1}^{P} \boldsymbol{W}_V \boldsymbol{x}_s \text{softmax}_l(\boldsymbol{x}_s^{\top} \boldsymbol{W}_K^{\top} \boldsymbol{W}_Q \boldsymbol{x}_l)), \tag{4}$$

where $\Psi = \{\{\boldsymbol{a}_{(l)}\}_{l=1}^{P}, \boldsymbol{W}_O, \boldsymbol{W}_V, \boldsymbol{W}_K, \boldsymbol{W}_Q\}$ denotes the set of all the model parameters. $\boldsymbol{a}_{(l)} \in \mathbb{R}^m$ and $\boldsymbol{W}_O \in \mathbb{R}^{m \times m_a}$ are the weights in the MLP layer. $\boldsymbol{W}_V \in \mathbb{R}^{m_a \times d}$, $\boldsymbol{W}_K, \boldsymbol{W}_Q \in \mathbb{R}^{m_b \times d}$ are weights in the self-attention layer. $\text{softmax}_l((\boldsymbol{W}_K \boldsymbol{x}_i)^{\top} \boldsymbol{W}_Q \boldsymbol{x}_l) = e^{(\boldsymbol{W}_K \boldsymbol{x}_i)^{\top} \boldsymbol{W}_Q \boldsymbol{x}_l} / \sum_{j=1}^{P} e^{(\boldsymbol{W}_K \boldsymbol{x}_j)^{\top} \boldsymbol{W}_Q \boldsymbol{x}_l}$. $\min\{m_a, m_b\} > d$.

**Fine-tuning algorithm for task vectors**. Denote $\{\boldsymbol{X}^n, y^n\}_{n=1}^{N}$ as a dataset with $N$ data points for the task function $\mathcal{T}$, i.e., $y^n = \mathcal{T}(\boldsymbol{X}^n)$ for $n \in [N]$. We fine-tune the model by minimizing the empirical risk function, i.e., $\min_{\Psi} \frac{1}{N}\sum_{n=1}^{N} \ell(\boldsymbol{X}^n, y^n; \Psi)$, via stochastic gradient descent (SGD) to obtain the task vector $\Delta\Psi_{\mathcal{T}}$ for $\mathcal{T}$. We use the Hinge loss $\ell(\boldsymbol{X}, y, \Psi) = \max\{1 - y \cdot f(\boldsymbol{X}; \Psi), 0\}$ as the loss function. For simplicity of analysis, we let $\boldsymbol{W} = \boldsymbol{W}_K^{\top} \boldsymbol{W}_Q \in \mathbb{R}^{d \times d}$ and $\boldsymbol{V} = \boldsymbol{W}_O \boldsymbol{W}_V \in \mathbb{R}^{m \times d}$ as (Jelassi et al., 2022; Huang et al., 2023; Zhang et al., 2023a). At the $t$-th iteration, $t = 0, 1, \cdots, T - 1$, the gradient is computed using a mini-batch $\mathcal{B}_t$ with $|\mathcal{B}_t| = B$. The step size is $\eta \leq O(1)$. Every entry of $\boldsymbol{W}$ and $\boldsymbol{V}$ is initialized from $\mathcal{N}(0, \xi^2)$ where $\xi \leq 1/\sqrt{m}$. Each $a_{(l)_i}$ is sampled from $\{+1/\sqrt{m}, -1/\sqrt{m}\}$. $\boldsymbol{a}_{(l)}$ does not update during the fine-tuning.

Following (Cao et al., 2022; Bu et al., 2024), we consider the **data formulation** as in Definition 2.

**Definition 2.** *Denote $\boldsymbol{\mu}_{\mathcal{T}} \in \mathbb{R}^d$ as the discriminative pattern for the task $\mathcal{T}$. Let $\{\boldsymbol{v}_1, \boldsymbol{v}_2, \cdots, \boldsymbol{v}_M\}$ be a set of $d$-dimensional orthonormal vectors that spans the subspace of task-irrelevant tokens $\boldsymbol{v}_j \perp \boldsymbol{\mu}_{\mathcal{T}}, j \in [M]$. Then, each $(\boldsymbol{X}, y) \sim \mathcal{D}_{\mathcal{T}}$ is generated as follows:*

- *Randomly generate the label $y$ from $\{+1, -1\}$ with an equal probability.*

- *Each token is randomly chosen from $\{\boldsymbol{\mu}_{\mathcal{T}}, -\boldsymbol{\mu}_{\mathcal{T}}\} \cup \{\boldsymbol{v}_1, \cdots, \boldsymbol{v}_M\}$. If $y = 1$ (or $-1$), the number of tokens equal to $\boldsymbol{\mu}_{\mathcal{T}}$ (or $-\boldsymbol{\mu}_{\mathcal{T}}$) is larger than that of $-\boldsymbol{\mu}_{\mathcal{T}}$ (or $\boldsymbol{\mu}_{\mathcal{T}}$)[1]. $\boldsymbol{\mu}_{\mathcal{T}}$ and $-\boldsymbol{\mu}_{\mathcal{T}}$ (or "$-\boldsymbol{\mu}_{\mathcal{T}}$ and $\boldsymbol{\mu}_{\mathcal{T}}$") are referred to **label-relevant** and **confusion patterns** for $y = 1$*

---

[1]This is motivated by empirical observations that embeddings of data with opposite labels, such as anonymous words, are significantly distinct (Engler et al., 2022) and even in opposite directions (Liu et al., 2024).

*(or $y = -1$), respectively. The average fractions of label-relevant, confusion tokens, and each $\boldsymbol{v}_i, i \in [M]$ are $\delta_*$, $\delta_\#$, and $(1 - \delta_* - \delta_\#)/M$, respectively.*

The basic idea of Definition 2 is that each label is determined by the dominant tokens with $\pm\boldsymbol{\mu}_\mathcal{T}$ patterns while all $\boldsymbol{v}_i$ do not affect labels.

### 3.3 HOW DO TASK ADDITION AND NEGATION AFFECT THE PERFORMANCE?

Next, we investigate the generalization of task addition and negation with task vectors obtained by fine-tuning. Consider the setting where $\mathcal{V} = \{1, 2\}$ with $\Delta\Psi_{\mathcal{T}_1}$ and $\Delta\Psi_{\mathcal{T}_2}$ as the task vectors for two binary tasks $\mathcal{T}_1$ and $\mathcal{T}_2$, respectively. $\mathcal{T}_1$ (or $\mathcal{T}_2$) is defined based on $\boldsymbol{\mu}_{\mathcal{T}_1}$ (or $\boldsymbol{\mu}_{\mathcal{T}_2}$) as the discriminative pattern following Definition 2. Hence, $\Psi = \Psi^{(0)} + \Delta\Psi_{\mathcal{T}_1} + \lambda\Delta\Psi_{\mathcal{T}_2}$.

Denote $\alpha = \boldsymbol{\mu}_{\mathcal{T}_1}^\top \boldsymbol{\mu}_{\mathcal{T}_2} \in [-1, 1]$, $\beta = \text{poly}(\eta\delta_*) + \Theta(\epsilon\sqrt{M})(< \Theta(1))$. Suppose the number of neurons $m \gtrsim M^2 \log M$ with $M = \Theta(d)$. Motivated by experiments in Table 1, we discuss three cases, i.e., $\alpha > 0$, $\alpha < 0$, and $\alpha = 0$, which corresponds to an "aligned", "contradictory", or "irrelevant" relationship between $\mathcal{T}_1$ and $\mathcal{T}_2$, respectively. Then, we state Theorem 1 for multi-task learning with the merged model $\Psi$.

**Theorem 1.** *(Success of Multi-Task Learning on Irrelevant and Aligned Tasks) For any $\epsilon \in (0, 1)$ and task $\mathcal{T}$, suppose the following conditions hold when fine-tuning a pre-trained model: (i) the batch size $B \geq \Omega(\epsilon^{-2} \log M)$, (ii) the step size $\eta \leq O(1)$, (iii) the number of training iterations $t \geq T = \Theta(\eta^{-1}\delta_*^{-2})$, then the returned model $\Psi_\mathcal{T}^*$ achieves a generalization error $\mathbb{E}_{(\boldsymbol{X}, y) \sim \mathcal{D}_\mathcal{T}}[\ell(\boldsymbol{X}, y; \Psi_\mathcal{T}^*)] \leq \Theta(\epsilon)$.*

*Moreover, given task vectors $\Delta\Psi_{\mathcal{T}_1}$ and $\Delta\Psi_{\mathcal{T}_2}$ obtained by fine-tuning as above for tasks $\mathcal{T}_1$ and $\mathcal{T}_2$, the resulting $\Psi = \Psi^{(0)} + \Delta\Psi_{\mathcal{T}_1} + \lambda\Delta\Psi_{\mathcal{T}_2}$ satisfies*
$$\mathbb{E}_{(\boldsymbol{X}, y) \sim \mathcal{D}_{\mathcal{T}_1}} \ell(\boldsymbol{X}, y; \Psi) \leq \Theta(\epsilon) + |\lambda| \cdot \beta, \quad \text{and} \ \ \mathbb{E}_{(\boldsymbol{X}, y) \sim \mathcal{D}_{\mathcal{T}_2}} \ell(\boldsymbol{X}, y; \Psi) \leq \Theta(\epsilon) \tag{5}$$

*provided that $\alpha \geq 0$, $\lambda \geq 1 - \alpha + \beta$.*

**Remark 1.** *Theorem 1 first states the sufficient conditions during the fine-tuning stage to obtain proper task vectors. Then, it characterizes the region of $\lambda$ to ensure both tasks achieve $\Theta(M^{-1})$ or $\Theta(\epsilon)$ generalization error by adding task vectors. For irrelevant tasks with $\alpha = 0$, a constant $\lambda \geq 1 - \beta$ is required. This implies that adding up the task vector $\Delta\Psi_{\mathcal{T}_2}$ in $\Psi$ results in a desired performance of multi-task learning. For aligned tasks with $\alpha > 0$, we can obtain a good multi-task learning performance if $\lambda \geq 1 - \alpha + \beta$. For contradictory tasks with $\alpha < 0$, we cannot find the proper $\lambda$ such that $\Psi$ obtains a small error on both $\mathcal{T}_1$ and $\mathcal{T}_2$ simultaneously, which means $\Psi$ can hardly generalize well on contradictory tasks.*

We then study the unlearning using the merged model $\Psi$ in different cases of $\alpha$.

**Theorem 2.** *(Success of Unlearning on Irrelevant and Contradictory Tasks) Given task vectors $\Delta\Psi_{\mathcal{T}_1}$ and $\Delta\Psi_{\mathcal{T}_2}$ that are fine-tuned following conditions (i)-(iii) in Theorem 1, the resulting $\Psi = \Psi^{(0)} + \Delta\Psi_{\mathcal{T}_1} + \lambda\Delta\Psi_{\mathcal{T}_2}$ satisfies*
$$\mathbb{E}_{(\boldsymbol{X}, y) \sim \mathcal{D}_{\mathcal{T}_1}} \ell(\boldsymbol{X}, y; \Psi) \leq \Theta(\epsilon) + |\lambda| \cdot \beta, \quad \text{and} \ \ \mathbb{E}_{(\boldsymbol{X}, y) \sim \mathcal{D}_{\mathcal{T}_2}} \ell(\boldsymbol{X}, y; \Psi) \geq \Theta(1) \tag{6}$$

*when (A) $\alpha = 0$, $\lambda \leq 0$; or (B) $\alpha < 0$, and $-\Theta(\alpha^{-2}) \leq \lambda \leq \text{poly}(\eta\delta_*)\alpha$, or (C) $0 < \alpha < 1 - c$ for some $c = \Theta(1)$, and $0 \leq \lambda \leq c/2$;*

**Remark 2.** *For irrelevant tasks with $\alpha = 0$, a constant $\lambda \leq 0$ can ensure a perfect unlearning on $\mathcal{T}_2$ while retaining on $\mathcal{T}_1$. For contradictory tasks with $\alpha < 0$, the unlearning performance is desired if a negative $\lambda$ is in $[-\Theta(\alpha^{-2}), -\text{poly}(\eta\delta_*)/\alpha]$, i.e., negating $\Delta\Psi_{\mathcal{T}_2}$. For aligned tasks with $\alpha > 0$, a proper $\lambda$ for unlearning to be successful only exists when $\alpha$ is small, indicating that unlearning becomes more challenging when tasks are more aligned.*

**Remark 3.** *Theorem 1 and 2 generally justify the validity of task addition, i.e., $\lambda > 0$ for multi-task learning and negation, i.e., $\lambda < 0$, for unlearning as long as $|\lambda|$ is not too large. The appropriate region for $\lambda$ is determined by $\alpha$, the correlation between the tasks.*

### 3.4 CAN A MODEL PROVABLY GENERALIZE OUT-OF-DOMAIN WITH TASK ARITHMETIC?

Consider $\{\Delta\Psi_{\mathcal{T}_i}\}_{i \in \mathcal{V}_\Psi}$ as a set of task vectors fine-tuned on $\Psi^{(0)}$ for binary classification tasks $\{\mathcal{T}_i\}_{i \in \mathcal{V}_\Psi}$. Each task $\mathcal{T}_i$ is defined with $\boldsymbol{\mu}_{\mathcal{T}_i}, i \in \mathcal{V}_\Psi$ as the discriminative pattern following Definition 2. Given the observation that task vectors are usually orthogonal to each other in practice (Ilharco et al., 2022a), we study the setup where $\{\boldsymbol{\mu}_{\mathcal{T}_i}\}_{i \in \mathcal{V}_\Psi}$ forms a set of orthonormal vectors.

We analyze the out-of-domain generalization on data $(\boldsymbol{X}, y) \sim \mathcal{D}_{\mathcal{T}'}$ for the task $\mathcal{T}'$, where the discriminative pattern is denoted by $\boldsymbol{\mu}_{\mathcal{T}'}$, and $\boldsymbol{\mu}_{\mathcal{T}'} = \sum_{i \in \mathcal{V}_{\Psi}} \gamma_i \boldsymbol{\mu}_{\mathcal{T}_i} + \kappa \cdot \boldsymbol{\mu}'_{\perp}$ with $\boldsymbol{\mu}'_{\perp} \perp \{\boldsymbol{\mu}_{\mathcal{T}_i}\}_{i \in \mathcal{V}_{\Psi}}$, $\|\boldsymbol{\mu}_{\mathcal{T}'}\| = \|\boldsymbol{\mu}'_{\perp}\| = 1$, $\gamma_i, \kappa \in \mathbb{R}$ for $i \in \mathcal{V}_{\Psi}$. Note that $\boldsymbol{\mu}_{\mathcal{T}'}$ contains a component $\boldsymbol{\mu}'_{\perp}$ that is orthogonal to all discriminative patterns of existing tasks, characterizing it as an out-of-domain task.

The following theorem summarizes the required conditions for out-of-domain generalization on $\mathcal{T}'$.

**Theorem 3.** *(Out-of-domain generalization using task arithmetic) Suppose $\boldsymbol{\mu}_{\mathcal{T}_i} \perp \boldsymbol{\mu}_{\mathcal{T}_j}$ for $i \neq j, i, j \in \mathcal{V}_{\Psi}$. Let $\Psi = \sum_{i \in \mathcal{V}_{\Psi}} \lambda_i \Delta \Psi_{\mathcal{T}_i} + \Psi^{(0)}, \lambda_i \neq 0$. Then, given that each $\Delta \Psi_{\mathcal{T}_i}$ is fine-tuned to achieve $\Theta(\epsilon)$ error following conditions (i)-(iii) in Theorem 1, as long as the following conditions (A) there exists $i \in \mathcal{V}_{\Psi}$ s.t., $\gamma_i \neq 0$, and (B)*

$$
\begin{cases}
\sum_{i \in \mathcal{V}_{\Psi}} \lambda_i \gamma_i \geq 1 + c, \\
\sum_{i \in \mathcal{V}_{\Psi}} \lambda_i \gamma_i^2 \geq 1 + c, \\
|\lambda_i| \cdot \beta \leq c, & \text{for some } c \in (0, 1) \text{ and all } i \in \mathcal{V}_{\Psi},
\end{cases}
\tag{7}
$$

*we have*
$$
\mathbb{E}_{(\boldsymbol{X}, y) \sim \mathcal{D}_{\mathcal{T}'}} \ell(\boldsymbol{X}, y; \Psi) \leq \Theta(\epsilon).
\tag{8}
$$

**Remark 4.** *Theorem 3 implies that linear operations of task vectors can produce a model that can generalize well on out-of-domain tasks $\mathcal{T}'$ that has a distribution shift from tasks $\mathcal{T}_i, i \in \mathcal{V}_{\Psi}$. With properly fine-tuned task vectors, the conditions to make out-of-domain generalization successful are (1) the discriminative pattern of the target task $\mathcal{T}'$ has a non-zero projection onto at least one of the discriminative pattern of tasks $\mathcal{T}_i, i \in \mathcal{V}_{\Psi}$; (2) the weighted summation of $\gamma_i$ and $\gamma_i^2$ with $\lambda_i$ as the coefficient should be greater than the margin of the binary classification task; (3) the absolute value of each $\lambda_i$ is not too large to avoid large errors to the resulting model $\Psi$.*

**Remark 5.** *Note that $\lambda_i$ satisfying (7) exists under mild conditions. In (75) of Appendix, we provide a closed-form solution that meets (7). We omit them from the main paper to simplify the presentation.*

### 3.5 CAN TASK VECTORS BE IMPLEMENTED EFFICIENTLY?

In this section, we theoretically investigate how to improve the computation efficiency of task vector techniques during inference. We focus on two properties of task vectors, low rankness and sparsity.

Consider the fine-tuned model $\Psi_{\mathcal{T}}^* = \{\{\boldsymbol{a}_{(l)}\}_{l=1}^P, \boldsymbol{W}_{O\mathcal{T}}^*, \boldsymbol{W}_{V\mathcal{T}}^*, \boldsymbol{W}_{K\mathcal{T}}^*, \boldsymbol{W}_{Q\mathcal{T}}^*\}$ with $\boldsymbol{W}_{\mathcal{T}}^* = \boldsymbol{W}_{K\mathcal{T}}^{*\top} \boldsymbol{W}_{Q\mathcal{T}}^*$ and $\boldsymbol{V}_{\mathcal{T}}^* = \boldsymbol{W}_{O\mathcal{T}}^* \boldsymbol{W}_{V\mathcal{T}}^*$ from Lemma 1. Denote $\Delta \boldsymbol{W}_{\mathcal{T}} = \boldsymbol{W}_{\mathcal{T}}^* - \boldsymbol{W}^{(0)}$ and $\Delta \boldsymbol{V}_{\mathcal{T}} = \boldsymbol{V}_{\mathcal{T}}^* - \boldsymbol{V}^{(0)}$. We have the following conclusions.

**Corollary 1.** *(Low-rank approximation) For any task $\mathcal{T}$ defined in Section 3.2, there exists $\Delta \boldsymbol{W}_{LR} \in \mathbb{R}^{d \times d}$ and $\Delta \boldsymbol{V}_{LR} \in \mathbb{R}^{m \times d}$ with $rank(\Delta \boldsymbol{W}_{LR}) = rank(\Delta \boldsymbol{V}_{LR}) = 1$, such that*

$$
\|\Delta \boldsymbol{W}_{\mathcal{T}} - \Delta \boldsymbol{W}_{LR}\|_F \leq M \cdot \epsilon + \frac{1}{\log M}, \quad \text{and} \quad \|\Delta \boldsymbol{V}_{\mathcal{T}} - \Delta \boldsymbol{V}_{LR}\|_F \leq \delta_*^{-1} \epsilon,
\tag{9}
$$

*hold. Moreover, Theorems 1-3 hold by replacing $\Delta \boldsymbol{W}_{\mathcal{T}}$ and $\Delta \boldsymbol{V}_{\mathcal{T}}$ with $\Delta \boldsymbol{W}_{LR}$ and $\Delta \boldsymbol{V}_{LR}$ in the task vectors and replacing $\epsilon$ with $\epsilon_{LR} = (\log \eta^{-1} + \delta_*^{-1}) \epsilon$ in the results.*

**Remark 6.** *Corollary 1 states that when $\epsilon \in (0, (M \log M)^{-1})$, we can find a rank-$1^2$ approximation of $\boldsymbol{W}^*$ and $\boldsymbol{V}^*$ with an error less than $\Theta(\log^{-1} M)$ to ensure that all Theorems hold with roughly the same generalization error. Specifically, with $\epsilon$ error derived in Theorems 1-3, using rank-1 approximation leads to $\epsilon_{LR} = (\log \eta^{-1} + \delta_*^{-1}) \epsilon$, which equals $\Theta(\epsilon)$ given $\eta$ and $\delta_*$ as constants. Hence, Corollary 1 indicates that low-rank approximation of individual task vectors generally preserves the performance of the model after applying task arithmetic.*

We also prove that task vectors are approximately sparse in Corollary 2, which implies that pruning task vectors does not change the generalization.

**Corollary 2.** *(Sparsity of task vectors) There exists $\mathcal{L} \subset [m]$ with $|\mathcal{L}| = \Theta(m)$ s.t.,*

$$
\|\boldsymbol{u}_i\| \geq \Omega(m^{-1/2}), i \in \mathcal{L}; \quad \|\boldsymbol{u}_i\| \leq O(m^{-1/2} \sqrt{\log B / B}), i \in [m] \backslash \mathcal{L},
\tag{10}
$$

*where $\boldsymbol{u}_i$ is the $i$-th row of $\Delta \boldsymbol{V}_{\mathcal{T}}^*$ and $B$ is the batch size of fine-tuning lower bounded in condition (i) of Lemma 1. Then, pruning all rows in $[m] \backslash \mathcal{L}$ of $\Delta \boldsymbol{V}_{\mathcal{T}}^*$ ensures Theorems 1-3 to hold.*

---

[2]The rank-1 approximation results from our simplified model that has one discriminative pattern per task. Our result indicates that the proper rank for approximation depends on the number of discriminative patterns for each task, which is far smaller than the model dimension in practice.

**Remark 7.** *Corollary 2 illustrates that a constant fraction of rows in $\Delta V_{\mathcal{T}}^*$ in $\mathcal{L}$ has a large magnitude, while the remaining ones in $[m]\backslash\mathcal{L}$ have much smaller magnitude. Then, we prove that removing rows in $[m]\backslash\mathcal{L}$ does not hurt the performance of multi-task learning, unlearning, and out-of-domain generalization by task arithmetic. This indeed justifies the existence of redundancy in "Delta parameters," a similar notion of task vectors, defined in (Yu et al., 2024), and verifies the validity of magnitude-based pruning on task vectors like TIES (Yadav et al., 2023) or DARE (Yu et al., 2024).*

### 3.6   Proof Sketch and Technical Novelty

We first provide the following informal lemma for the fine-tuned task vector. Lemma 1 provides the convergence of the fine-tuning process and the properties the obtained task vector satisfies.

**Lemma 1.** *(informal) A model $\Psi$ has a generalization error $\Theta(\epsilon)$ on task $\mathcal{T}$ (with the discriminative pattern $\mu_{\mathcal{T}}$) if $\Delta\Psi := \Psi - \Psi^{(0)} = \{\Delta W, \Delta V\}$ satisfy both conditions as follows:*

*(A) the attention weights between two label-relevant patterns are dominant, while the attention values between a label-relevant pattern and any other pattern are close to zero;*

*(B) A constant fraction of rows in $\Delta V$ in the MLP layer has a large magnitude with a direction either close to $\mu_{\mathcal{T}}$ or $-\mu_{\mathcal{T}}$, while the remaining rows have small weights.*

*Moreover, any task vector obtained by fine-tuning on task $\mathcal{T}$ satisfying conditions (i)-(iii) in Theorem 1 satisfy conditions (A) and (B) for task $\mathcal{T}$.*

The proof ideas of Theorems 1 and 2 are as follows. To ensure a successful multi-task learning stated in (2), we need $\Delta\Psi_{\mathcal{T}_1} + \lambda\Delta\Psi_{\mathcal{T}_2}$ satisfying both conditions (A) and (B) in Lemma 1 for tasks $\mathcal{T}_1$ and $\mathcal{T}_2$. To ensure unlearning $\mathcal{T}_2$ and maintaining the generalization in $\mathcal{T}_1$ as stated in (3), we need $\Delta\Psi_{\mathcal{T}_1} + \lambda\Delta\Psi_{\mathcal{T}_2}$ satisfying (A) and (B) for $\mathcal{T}_1$ but failing either (A) or (B) for $\mathcal{T}_2$. When $\alpha = 0$, the component of $\Delta\Psi_{\mathcal{T}_i}$ in $\Psi$ has negligibile effect on data from $\mathcal{T}_j$, for any $i \neq j, i, j \in \{1, 2\}$. When $\alpha > 0$, both $\mathcal{T}_1$ and $\mathcal{T}_2$ should tend to favor $\lambda > 0$ for a good generalization. When $\alpha < 0$, $\mathcal{T}_1$ prefers a negative $\lambda$, while $\mathcal{T}_2$ prefers a positive $\lambda$.

To prove the out-of-domain generalization in Theorem 3, we need to find a proper set of $\lambda_i, i \in \mathcal{V}_\Psi \cap \mathcal{V}'$ such that $\sum_{i\in\mathcal{V}_\Psi} \lambda_i\Delta\Psi_{\mathcal{T}_i}$ hold for conditions (A) and (B) in Lemma 1 for the task $\mathcal{T}'$. The proof idea for Corollaries 1 and 2 comes from an observation from Lemma 1. That is, Conditions (A) and (B) demonstrate that the rows in $\Delta V$ and the matrix $\Delta W$ only enlarge tokens in the direction of label-relevant pattern or its opposite. This implies the sparsity of $\Delta V$ and the low-rank property of the entire $\Delta\Psi$. The proofs for Theorems 1 and 2 and 3 and Corollaries 1 and 2 can be found in Appendix D, respectively.

**Technical Novelty**. Compared with (Li et al., 2023a), Lemma 1 establishes a more fine-grained characterization of $\Delta\Psi_{\mathcal{T}}$, which allows us to perform a detailed analysis of layer-by-layer outputs of the merged model. Furthermore, Lemma 1 extends the theoretical analysis to training from random initialization with two merged trainable parameter matrices $W$ and $V$.

Moreover, to the best of our knowledge, we provide the first generalization analysis of task arithmetic in model editing (Theorems 1, 2, and 3). The merged model $\Psi$ preserves the nonlinearity of task vectors from the nonlinear model architecture rather than linearizing the model by impractical infinite wide network assumption in (Ortiz-Jimenez et al., 2023). This allows us to expand the understanding of task arithmetic beyond the NTK region as in (Ortiz-Jimenez et al., 2023), where the problem is extremely overparameterized.

## 4   Numerical Experiments

We conduct extensive experiments on image classification and natural language generation to verify the effectiveness of task vectors in different downstream tasks. For image classification, we use the ViT-Small/16 model (Dosovitskiy et al., 2020) pre-trained from ImageNet-21K (Russakovsky et al., 2015) for downstream tasks with Colored-MNIST (Arjovsky et al., 2019; Chapel et al., 2020). For natural language generation, we use the open-source Phi-1.5 (1.3B) language model (Gunasekar et al., 2023; Li et al., 2023d). We repeat the experiment using LoRA with Phi-3-small (7B) in Appendix B.

### 4.1 EXPERIMENTS ON IMAGE CLASSIFICATION

**Experiment Setup**. To control the correlation between tasks, we use Colored-MNIST for image classification tasks. We designed binary classification problems based on the parity of digits, where odd digits are labeled as $+1$ and even digits as $-1$. We utilize two colors, red and green, to construct different task correlations. Define $r_o$ and $r_e$ as the proportion of red colors in odd and even digits, respectively. Then, the proportion of green colors in odd and even digits are $1 - r_o$ and $1 - r_e$, respectively. Across all of our experiments, we set $r_e = 1 - r_o$. The correlation $\hat{\alpha}(\Psi^*_{\mathcal{T}_1}, \Psi^*_{\mathcal{T}_2})$ between two tasks $\mathcal{T}_1$ and $\mathcal{T}_2$, with $\mathcal{D}_1$ and $\mathcal{D}_2$ respectively as the corresponding test set, is approximated by their averaged cosine similarity between centered outputs from the two fine-tuned models, i.e.,

$$\hat{\alpha}(\Psi^*_{\mathcal{T}_1}, \Psi^*_{\mathcal{T}_2}) = 1/2(\hat{\alpha}(\Psi^*_{\mathcal{T}_1}, \Psi^*_{\mathcal{T}_2}, \mathcal{D}_1) + \hat{\alpha}(\Psi^*_{\mathcal{T}_1}, \Psi^*_{\mathcal{T}_2}, \mathcal{D}_2)),$$

$$\text{where } \hat{\alpha}(\Psi^*_{\mathcal{T}_1}, \Psi^*_{\mathcal{T}_2}, \mathcal{D}_j) = \sum_{i \in \mathcal{D}_j} \frac{\cos \left\langle \tilde{\boldsymbol{y}}^i_{1,j}, \tilde{\boldsymbol{y}}^i_{2,j} \right\rangle}{|\mathcal{D}_j|}, \tilde{\boldsymbol{y}}^i_{l,j} = \hat{\boldsymbol{y}}^i_{l,j} - \frac{1}{|\mathcal{D}_j|} \sum_{i \in \mathcal{D}_j} \hat{\boldsymbol{y}}^i_{l,j}, \ l,j \in \{1,2\}. \quad (11)$$

$\hat{\boldsymbol{y}}^i_{l,j}$ represents the $i$-th output of the fine-tuned model $\Psi^*_{\mathcal{T}_l}$ on the test set $\mathcal{D}_j$. Note that to compute $\hat{\alpha}(\Psi^*_{\mathcal{T}_1}, \Psi_{\mathcal{T}_2^*})$ by (11), we do not require the availability of extra models or datasets except $\Psi^*_{\mathcal{T}_1}$, $\Psi^*_{\mathcal{T}_1}$, and the test set $\mathcal{D}_1$ and $\mathcal{D}_2$.

**Experiment Results**. We first investigate the ability of task arithmetic using $\Psi = \Psi^{(0)} + \Delta\Psi_{\mathcal{T}_1} + \lambda\Delta\Psi_{\mathcal{T}_2}$ to handle multi-task learning and unlearning under three cases in terms of task correlations. Let $r_o = 0.95$ for $\mathcal{T}_1$. In case I, let $r_o = r_e = 0.5$ in $\mathcal{T}_2$. In case II, let $r_o = 0.9$ in $\mathcal{T}_2$, and in case III, let $r_o = 0.05$ in $\mathcal{T}_2$. The computed correlations $\hat{\alpha}(\Psi^*_{\mathcal{T}_1}, \Psi^*_{\mathcal{T}_2})$ of the above three settings are $0.164$, $0.891$, and $-0.849$, which corresponds to irrelevant ($\alpha \approx 0$), aligned ($\alpha > 0$), and contradictory ($\alpha < 0$) tasks discussed in Theorem 1, respectively. Figure 1 illustrates that when tasks are irrelevant, successful multi-task learning on both tasks and unlearning on task $\mathcal{T}_2$ can be achieved when $\lambda \geq 1$ and $\lambda \leq 0$, respectively. When tasks are aligned, the trend of testing accuracy of $\Psi$ on $\mathcal{T}_1$ and $\mathcal{T}_2$ are consistent. A superior multi-task learning performance can be observed when $\lambda > 0$, and one cannot find a region of $\lambda$ where $\mathcal{T}_2$ is unlearned while maintaining the accuracy for $\mathcal{T}_1$. When tasks are contradictory, one can obtain a good unlearning behavior when $\lambda \leq 0$, and no selection of $\lambda$ can achieve multi-task learning. This result verifies Theorems 1 and 2 for $\alpha = 0$, $\alpha > 0$, and $\alpha < 0$, respectively.

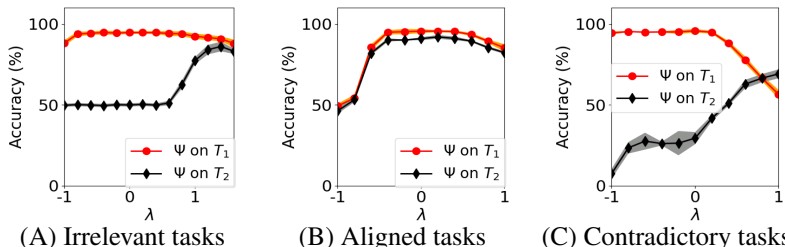

(A) Irrelevant tasks      (B) Aligned tasks      (C) Contradictory tasks

Figure 1: Testing accuracy of the merged model $\Psi$ on task $\mathcal{T}_1$ and $\mathcal{T}_2$.

We then study the out-of-domain generalization capability of task arithmetic. We consider a merged model $\Psi = \Psi^{(0)} + \lambda_1\Delta\Psi_{\mathcal{T}_1} + \lambda_2\Delta\Psi_{\mathcal{T}_2}$ constructed by two task vectors. In $\mathcal{T}_1$, we let $r_o = 0.85$, while in $\mathcal{T}_2$, we let $r_o = 0.05$. In the target task $\mathcal{T}'$, $r_o = 0.9$. We compute that $\hat{\alpha}(\Psi^*_{\mathcal{T}_1}, \Psi^*_{\mathcal{T}_2}) = 0.115$, which means $\mathcal{T}_1$ and $\mathcal{T}_2$ are approximately irrelevant. Figure 2 (A) demonstrates that in a triangular region with the black dashed line of $\lambda_1$ and $\lambda_2$, we can achieve a good generalization performance. This region is consistent with the red region in Figure 2 (B), which is produced by condition (7)[3] where $\gamma_1$ and $\gamma_2$ are estimated by $\hat{\alpha}(\Psi^*_{\mathcal{T}_1}, \Psi^*_{\mathcal{T}'}) = 0.792$ and $\hat{\alpha}(\Psi^*_{\mathcal{T}_2}, \Psi^*_{\mathcal{T}'}) = -0.637$. We choose small values $\beta = 0.01$, $c = 0.02$. The

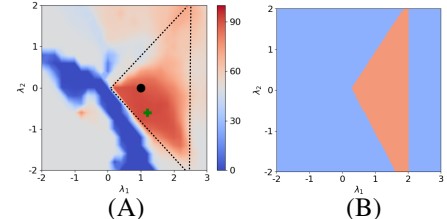

(A)      (B)

Figure 2: (A) The heatmap of the testing accuracy (the color bar %) on $\mathcal{T}'$ using the merged model $\Psi$. The black dot is the baseline, while the green cross is the best $\lambda_1, \lambda_2$. (B) The red region satisfies (7), while the blue region does not.

result justifies the sufficient conditions for a successful out-of-domain generalization in Theorem 3.

---

[3]Since the practical classification margin might be smaller than that of Hinge loss used in our theoretical analysis, we replace $1 + c$ in (7) with $0.2 + c$.

## 4.2 EXPERIMENT ON LANGUAGE GENERATION TASK

**Experiment setup**. We study the unlearning performance using three datasets, "Harry Potter 1" (HP1), "Harry Potter 2" (HP2) by J.K. Rowling, and "Pride and Prejudice" (PP) by Jane Austen. We consider HP1 and HP2 as semantically similar and aligned books due to the shared authors ($\hat{\alpha}(\Psi^*_{\mathcal{T}_{HP1}}, \Psi^*_{\mathcal{T}_{HP2}}) = 0.498$ by (11)) following Dou et al. (2024), while PP is less aligned with HP1 than HP2 ($\hat{\alpha}(\Psi^*_{\mathcal{T}_{HP1}}, \Psi^*_{\mathcal{T}_{PP}}) = 0.239$ by (11)). We study Next Token Prediction on these three datasets separately as three different tasks, denoted by $\mathcal{T}_{HP1}$, $\mathcal{T}_{HP2}$, and $\mathcal{T}_{PP}$, respectively. Then $\mathcal{T}_{HP1}$ and $\mathcal{T}_{HP2}$ are greatly aligned, while $\mathcal{T}_{HP1}$ and $\mathcal{T}_{PP}$ are less aligned.

Denote the pre-trained Phi-1.5 model as $\Psi^{(0)}$. We first fine-tune $\Psi^{(0)}$ on all three datasets jointly to obtain $\Psi^{(0)'}$, which has favorable generalization for all tasks $\mathcal{T}_{HP1}$, $\mathcal{T}_{HP2}$, and $\mathcal{T}_{PP}$. Initialized from $\Psi^{(0)}$, we fine-tune on dataset HP1 to obtain model $\Psi^*_{HP1}$. The task vector for $\mathcal{T}_{HP1}$ is computed as: $\Delta\Psi_{HP1} = \Psi^*_{HP1} - \Psi^{(0)}$. The merged model is $\Psi = \Psi^{(0)'} + \lambda \cdot \Delta\Psi_{HP1}$.

**Experiment results**. We vary $\lambda$ and evaluate the performance on $\mathcal{T}_{HP1}$, $\mathcal{T}_{HP2}$, and $\mathcal{T}_{PP}$, respectively. The evaluation metric is the Rouge-L score used in (Dou et al., 2024), which measures the ratio of the longest common sequence between the original book and the LLM's generation. A higher score indicates a better generation performance. As shown in Table 3, when $\lambda$ becomes negative, the Rouge-L score for $\mathcal{T}_{HP1}$ decreases, indicating the success of unlearning. When $\lambda$ is the smallest value in the experimental selection ($\lambda = -1$), the unlearning performance is the best, with the Rouge-L decreasing by $37.23\%$ from $\Psi^{(0)'}$. Moreover, when $\mathcal{T}_{HP1}$ is unlearned, the performance of $\mathcal{T}_{HP2}$ also degrades significantly, with the Rouge-L score decreasing by $34.71\%$. In contrast, the performance degradation on $\mathcal{T}_{PP}$ is much smaller, with a decrease by $15.13\%$[4]. This verifies Theorem 2 that unlearning a task $\mathcal{T}_{HP1}$ can effectively degrade the performance of the aligned task ($\mathcal{T}_{HP2}$) as well, while the performance degradation on the less aligned task ($\mathcal{T}_{PP}$) is relatively smaller.

| $\lambda$ | 0 (baseline) | $-0.2$ | $-0.4$ | $-0.6$ | $-0.8$ | $-1$ |
|---|---|---|---|---|---|---|
| $\mathcal{T}_{HP1}$ | 0.2213 | 0.2211 | 0.1732 | 0.1866 | 0.1572 | **0.1389** ($37.23\%\downarrow$) |
| $\mathcal{T}_{HP2}$ | 0.2302 | 0.2032 | 0.2111 | 0.2034 | 0.1695 | **0.1503** ($34.71\%\downarrow$) |
| $\mathcal{T}_{PP}$ | 0.1983 | 0.1888 | 0.1877 | 0.1802 | 0.1932 | **0.1683** ($15.13\%\downarrow$) |

Table 3: Rouge-L scores of $\mathcal{T}_{HP1}$, $\mathcal{T}_{HP2}$, and $\mathcal{T}_{PP}$ by $\Psi = \Psi^{(0)'} + \lambda \cdot \Delta\Psi_{HP1}$ using full-rank task vector $\Delta\Psi_{HP1}$.

We also implement our experiment using LoRA in fine-tuning to compute the task vector. We set the rank of each parameter as 32, which requires to tune only $0.35\%$ of total parameters and reduces the peak memory consumption by $54\%$. Let $\Delta\Psi^{LR}_{HP1}$ denote the resulting low-rank task vector for $\mathcal{T}_{HP1}$. We repeat the experiments by replacing $\Delta\Psi_{HP1}$ with $\Delta\Psi^{LR}_{HP1}$. Comparing Table 4 to Table 3, on can see that all the insights still hold when using a low-rank task vector, verifying Corollary 1.

| $\lambda$ | 0 (baseline) | $-0.2$ | $-0.4$ | $-0.6$ | $-0.8$ | $-1$ |
|---|---|---|---|---|---|---|
| $\mathcal{T}_{HP1}$ | 0.2432 | 0.2033 | 0.1857 | 0.1665 | 0.1439 | **0.1568** ($35.53\%\downarrow$) |
| $\mathcal{T}_{HP2}$ | 0.2335 | 0.1932 | 0.2065 | 0.1813 | 0.1664 | **0.1772** ($24.11\%\downarrow$) |
| $\mathcal{T}_{PP}$ | 0.2111 | 0.2001 | 0.1884 | 0.1963 | 0.1849 | **0.1819** ($13.83\%\downarrow$) |

Table 4: Rouge-L scores of $\mathcal{T}_{HP1}$ $\mathcal{T}_{HP2}$, and $\mathcal{T}_{PP}$ by $\Psi = \Psi^{(0)'} + \lambda \cdot \Delta\Psi^{LR}_{HP1}$ using low-rank task vector $\Delta\Psi^{LR}_{HP1}$.

## 5 CONCLUSIONS

In this paper, we theoretically investigate the generalization ability of the task vector technique. Based on feature learning analysis of a one-layer nonlinear Transformer, we quantitatively characterize the selection of arithmetic hyperparameters and their dependence on task correlations so that the resulting task vectors achieve desired multi-task learning, unlearning, and out-of-domain generalization. We also demonstrate the validity of using sparse or low-rank task vectors. Theoretical results are justified on large language models. Future directions include analyzing the performance of task vectors in more complex models and designing more robust task vector selection methods.

---

[4]Note that the task vector method leads to a $13.1\%$ decrease in Rouge-L score on BOOKS dataset on average (Shi et al., 2024). The state-of-the-art unlearning methods are empirically shown to result in a performance drop in utility (Maini et al., 2024; Shi et al., 2024).

ACKNOWLEDGMENTS

This work was supported by National Science Foundation(NSF) #2430223, Army Research Office (ARO) W911NF-25-1-0020, and the Rensselaer-IBM Future of Computing Research Collaboration (http://airc.rpi.edu). The work of Yihua Zhang and Sijia Liu was also supported by the National Science Foundation (NSF) CISE Core Program Award IIS-2207052, the NSF CAREER Award IIS-2338068, the ARO Award W911NF2310343, the Cisco Research Award, and the Amazon Research Award for AI in Information Security. The work of Shuai Zhang was supported by National Science Foundation (NSF) #2349879. We also thank all anonymous reviewers for their constructive comments.

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

## A  ADDITIONAL DISCUSSION

It was brought to our attention after the acceptance of ICLR 2025 in January 2025, that there is a recent submission on arxiv in February 2025 (Zeng et al., 2025) that also considers the theoretical generalization analysis of task vectors in multi-task learning, unlearning, and out-of-domain generalization. Their analysis is built upon assumptions that (i) the studied models are already fine-tuned (Assumption 4.1); (ii) the norm of task vectors is upper bounded (Assumption 4.1); (iii) different task vectors are almost orthogonal to each other (Assumption 4.2). In contrast, although our analysis is based on a one-layer single-head Transformer, we do not rely on the aforementioned assumptions. Our results show that the convergent models trained with SGD yield task vectors that support multi-task learning, unlearning, and out-of-distribution (OOD) generalization. We analyze the behavior of task arithmetic under aligned, irrelevant, and contradictory task relationships without requiring the orthogonality assumption between task vectors. Moreover, unlike (Zeng et al., 2025) that assumes sparsity of task vectors, we theoretically prove that task vectors obtained via fine-tuning can exhibit both low-rank structure and sparsity.

## B  ADDITIONAL EXPERIMENTS

We repeat the language generation experiment in Section 4.2 with Phi-3-small (7B). The task vectors are obtained by LoRA (Hu et al., 2022). Table 5 shows that the insight of Theorem 2 still holds, i.e., unlearning a certain task (HP1) can effectively forget the aligned task (HP2) with a $52.29\%$ decrease of Rouge-L scores, while the Rouge-L score for the less-aligned task (PP) has a decrease of only $20.65\%$. Moreover, by using a larger model than Phi-1.5, the unlearning performance of the aligned task HP2 is improved from $37.23\%$ decrease to $55.61\%$ decrease. In comparison, the performance difference on the less-aligned PP is much smaller, from $15.13\%$ decrease to $20.65\%$ decrease.

| $\lambda$ | 0 (baseline) | $-0.2$ | $-0.4$ | $-0.6$ | $-0.8$ | $-1$ |
|---|---|---|---|---|---|---|
| $\mathcal{T}_{\text{HP1}}$ | 0.2573 | 0.1989 | 0.1933 | 0.1888 | 0.1572 | **0.1142** ($55.61\% \downarrow$) |
| $\mathcal{T}_{\text{HP2}}$ | 0.2688 | 0.2113 | 0.1993 | 0.1938 | 0.1622 | **0.1563** ($52.29\% \downarrow$) |
| $\mathcal{T}_{\text{PP}}$ | 0.1942 | 0.1825 | 0.1644 | 0.1687 | 0.1592 | **0.1541** ($20.65\% \downarrow$) |

Table 5: Rouge-L scores of $\mathcal{T}_{\text{HP1}}$ $\mathcal{T}_{\text{HP2}}$, and $\mathcal{T}_{\text{PP}}$ by $\Psi = \Psi^{(0)'} + \lambda \cdot \Delta\Psi_{\text{HP1}}^{\text{LR}}$ using low-rank task vector $\Delta\Psi_{\text{HP1}}^{\text{LR}}$ with Phi-3-small (7B).

## C  PRELIMINARIES OF THEORY

We first summarize the notations we use in this paper in Table (6).

**Definition 3.** *For a task based on any discriminative pattern $\boldsymbol{\mu}_1$,*

1. $q_1(t) = \boldsymbol{\mu}_1^\top \boldsymbol{W}^{(t)} \boldsymbol{\mu}_1$.

2. $\mathcal{S}^n$: *the set of tokens in the $n$-th data.* $\mathcal{S}_1^n$: *the set of tokens of $\boldsymbol{\mu}_1$ in the $n$-th data.* $\mathcal{S}_2^n$: *the set of tokens of $-\boldsymbol{\mu}_1$ in the $n$-th data.* $\mathcal{R}_k^n$: *the set of tokens of $\boldsymbol{v}_k$ in the $n$-th data.*

3. $\phi_n(t) = \frac{1}{|\mathcal{S}_1^n| e^{q_1(t)^2} + P - |\mathcal{S}_1|}$.

4. $p_n(t) = \sum_{s,l \in \mathcal{S}_1^n \text{ or } s,l \in \mathcal{S}_2^n} softmax_l(\boldsymbol{x}_s^n \boldsymbol{W}^{(t)} \boldsymbol{x}_l^n)$.

5. $\zeta_{i,1,t} = \boldsymbol{V}_{(i,\cdot)}^{(t)} \boldsymbol{x}_s^n$ for $s \in \mathcal{S}_1^n$.

6. $\zeta_{1,t} = \min_{i \in [m]} \zeta_{i,1,t}$.

7. $softmax_l(\boldsymbol{X}^{n\top} \boldsymbol{W} \boldsymbol{x}_l) = (softmax_l(\boldsymbol{x}_1^{n\top} \boldsymbol{W} \boldsymbol{x}_l), \cdots, softmax_l(\boldsymbol{x}_P^{n\top} \boldsymbol{W} \boldsymbol{x}_l))$.

**Definition 4.** *Define*

$$\boldsymbol{R}_l^n(t) := \sum_{s=1}^{P} \boldsymbol{V}^{(t)} \boldsymbol{x}_s^n softmax_l(\boldsymbol{x}_s^{n\top} \boldsymbol{W}^{(}t) \boldsymbol{x}_l^n), \tag{12}$$

Table 6: Summary of Notations

| Notations | Annotation |
|---|---|
| $\boldsymbol{X}, \boldsymbol{x}_i, \boldsymbol{X}^n, y^n$ | $\boldsymbol{X}$ is the input data, which contains $P$ tokens. $\boldsymbol{x}_i$ is the $i$-th token of $\boldsymbol{X}$. $\boldsymbol{X}^n$ is the $n$-th input data with $y^n$ as the corresponding label. |
| $\Psi$ | $\Psi = \{\{\boldsymbol{a}_{(l)}\}_{l=1}^{P}, \boldsymbol{W}_O, \boldsymbol{W}_V, \boldsymbol{W}_K, \boldsymbol{W}_Q\}$ denotes the set of all the model parameters. $\boldsymbol{a}_{(l)} \in \mathbb{R}^m$ and $\boldsymbol{W}_O \in \mathbb{R}^{m \times m_a}$ are the weights in the MLP layer. $\boldsymbol{W}_V \in \mathbb{R}^{m_a \times d}, \boldsymbol{W}_K, \boldsymbol{W}_Q \in \mathbb{R}^{m_b \times d}$ are weights in the self-attention layer. |
| $\Psi^{(0)}, \Psi_{\mathcal{T}}^*, \Delta\Psi_{\mathcal{T}}$ | $\Psi^{(0)}$ is the pre-trained model. $\Psi_{\mathcal{T}}^*$ is the fine-tuned model on a given task $\mathcal{T}$. $\Delta\Psi_{\mathcal{T}}$ is the task vector of the task $\mathcal{T}$, which is computed as $\Delta\Psi_{\mathcal{T}} = \Psi_{\mathcal{T}}^* - \Psi^{(0)}$. |
| $\boldsymbol{\mu}_{\mathcal{T}}, \boldsymbol{v}_j$ | $\boldsymbol{\mu}_{\mathcal{T}}$ is the discriminative pattern of the task $\mathcal{T}$. $\boldsymbol{v}_j$ is the $j$-th task-irrelevant pattern, $j \in [M]$. |
| $\delta_*, \delta_\#$ | $\delta_*$ is the average fraction of label-relevant pattern in the input data. $\delta_\#$ is the average fraction of confusion pattern in the input data. |
| $q_1(t), \zeta_{1,t}, p_n(t)$ | $q_1(t) = \boldsymbol{\mu}_1^\top \boldsymbol{W}^{(t)} \boldsymbol{\mu}_1$ denotes the value of the product, where the patterns on both sides of $\boldsymbol{W}^{(t)}$ are the same. $\zeta_{1,t}$ denotes the modified value embedding of $\boldsymbol{\mu}_1$ at the $t$-th iteration. $p_n(t)$ refers to the summation of attention weights where the key and the query are the same discriminative pattern. |
| $\mathcal{W}_{n,l}, \mathcal{U}_{n,l}$ | $\mathcal{W}_{n,l}$ and $\mathcal{U}_{n,l}$ respectively represent of sets of positive or negative neurons so that the Relu activation is activated with $\boldsymbol{x}_l^n$ as the query. |
| $\mathcal{B}_b$ | $\mathcal{B}_b$ is the SGD batch at the $b$-th iteration. |
| $\mathcal{O}(), \Omega(), \Theta()$ | We follow the convention that $f(x) = O(g(x))$ (or $\Omega(g(x)), \Theta(g(x)))$) means that $f(x)$ increases at most, at least, or in the order of $g(x)$, respectively. |
| $a$ | $a = |\boldsymbol{a}_{(l)_i}| = 1/\sqrt{m}$ for $i \in [m]$. |
| $\gtrsim, \lesssim$ | $f(x) \gtrsim g(x)$ (or $f(x) \lesssim g(x)$ ) means that $f(x) \geq \Omega(g(x))$ (or $f(x) \lesssim \mathcal{O}(g(x))$). |

*Define $\mathcal{W}_{n,l}, \mathcal{U}_{n,l}$ as the sets of lucky neurons such that*

$$\mathcal{W}_{n,l} = \{i : \boldsymbol{V}_{(i,\cdot)}^\top \boldsymbol{R}_{n,l}(0) > 0, l \in \mathcal{S}_1^n, a_i > 0\}, \tag{13}$$

$$\mathcal{U}_{n,l} = \{i : \boldsymbol{V}_{(i,\cdot)}^\top \boldsymbol{R}_{n,l}(0) > 0, l \in \mathcal{S}_2^n, a_i < 0\}. \tag{14}$$

**Definition 5** ((Vershynin, 2010)). *We say $X$ is a sub-Gaussian random variable with sub-Gaussian norm $K > 0$, if $(\mathbb{E}|X|^p)^{\frac{1}{p}} \leq K\sqrt{p}$ for all $p \geq 1$. In addition, the sub-Gaussian norm of X, denoted $\|X\|_{\psi_2}$, is defined as $\|X\|_{\psi_2} = \sup_{p \geq 1} p^{-\frac{1}{2}} (\mathbb{E}|X|^p)^{\frac{1}{p}}$.*

**Lemma 2** (Vershynin (2010) Proposition 5.1, Hoeffding's inequality). *Let $X_1, X_2, \cdots, X_N$ be independent centered sub-gaussian random variables, and let $K = \max_i \|X_i\|_{\psi_2}$. Then for every $\boldsymbol{a} = (a_1, \cdots, a_N) \in \mathbb{R}^N$ and every $t \geq 0$, we have*

$$\Pr\left(\left|\sum_{i=1}^N a_i X_i\right| \geq t\right) \leq e \cdot \exp\left(-\frac{ct^2}{K^2\|\boldsymbol{a}\|^2}\right), \tag{15}$$

*where $c > 0$ is an absolute constant.*

**Lemma 3.** *For task $\mathcal{T}$ based on any $\boldsymbol{\mu}_1$, $0 \leq t \leq T$, there exists $K(t) > 0$, such that*

$$\boldsymbol{W}^{(t+1)}\boldsymbol{\mu}_1 = \boldsymbol{W}^{(t+1)}\boldsymbol{\mu}_1 + K(t)\boldsymbol{\mu}_1 + \sum_{l=1}^M \iota_l' \boldsymbol{\mu}_l, \tag{16}$$

*where*

$$K(t) \gtrsim \eta \frac{1}{B} \sum_{n \in \mathcal{B}_b} \frac{m|\mathcal{S}_1^n|}{aP} \zeta_{1,t} p_n(t) \phi_n(t)(P - |\mathcal{S}_1^n|), \tag{17}$$

$$\iota_l' \leq K(t) \cdot e^{-q_1(t)}. \tag{18}$$

For $k \in [M]$,

$$\|\boldsymbol{\mu}_1^\top \boldsymbol{W}^{(t)} \boldsymbol{v}_k\| \lesssim \sqrt{\frac{\log B}{B}} \sum_{b=0}^{t} K(b), \tag{19}$$

and for $j \neq k$, $j \in [M]$,

$$\|\boldsymbol{v}_j^\top \boldsymbol{W}^{(t)} \boldsymbol{v}_k\| \lesssim K(t)e^{-q_1(t)}, \tag{20}$$

For any $\boldsymbol{\mu}'$ such that $\boldsymbol{\mu}_1^\top \boldsymbol{\mu}' = \alpha$ and $\boldsymbol{\mu}' \perp \boldsymbol{v}_1, \boldsymbol{v}_2, \cdots, \boldsymbol{v}_M$, we have

$$\boldsymbol{\mu}'^\top \boldsymbol{W}^{(t)} \boldsymbol{\mu}' = \alpha^2 \boldsymbol{\mu}_1^\top \boldsymbol{W}^{(t)} \boldsymbol{\mu}_1 \cdot (1 \pm \Theta(\epsilon)), \tag{21}$$

if $B \geq \epsilon^{-2} \log M$ for some $\epsilon < 1$.

**Lemma 4.** *Given a task $\mathcal{T}$ based on any $\boldsymbol{\mu}_1$, $0 \leq t \leq T$. Then, for $i \in \mathcal{W}_{n,l}$,*

$$\boldsymbol{V}_{(i,\cdot)}^{(t)} \boldsymbol{\mu}_1 \gtrsim \eta \sum_{b=0}^{t-1} \frac{1}{B} \sum_{n \in \mathcal{B}_b} \frac{|\mathcal{S}_1^n|}{aP} \cdot p_n(b), \tag{22}$$

$$\boldsymbol{V}_{(i,\cdot)}^{(t)} \boldsymbol{v}_k \lesssim \eta \sum_{b=0}^{t-1} \frac{1}{B} \sum_{n \in \mathcal{B}_b} \frac{|\mathcal{S}_1^n|}{aPM}, \tag{23}$$

*for $k \in [M]$. For $i \in \mathcal{U}_{n,l}$, we similarly have*

$$-\boldsymbol{V}_{(i,\cdot)}^{(t)} \boldsymbol{\mu}_1 \gtrsim \eta \sum_{b=0}^{t-1} \frac{1}{B} \sum_{n \in \mathcal{B}_b} \frac{|\mathcal{S}_2^n|}{aP} \cdot p_n(b), \tag{24}$$

$$\boldsymbol{V}_{(i,\cdot)}^{(t)} \boldsymbol{v}_k \lesssim \eta \sum_{b=0}^{t-1} \frac{1}{B} \sum_{n \in \mathcal{B}_b} \frac{|\mathcal{S}_1^n|}{aPM}, \tag{25}$$

*for some $k \in [M]$. For $i \notin \mathcal{W}_{n,l} \cup \mathcal{U}_{n,l}$, we have that*

$$\boldsymbol{V}_{(i,\cdot)}^{(t)} \boldsymbol{\mu}_1 \lesssim \sqrt{\frac{\log B}{B}} \boldsymbol{V}_{(j,\cdot)}^{(t)} \boldsymbol{\mu}_1, \tag{26}$$

$$\boldsymbol{V}_{(i,\cdot)}^{(t)} \boldsymbol{v}_k \lesssim \sqrt{\frac{\log B}{B}} \boldsymbol{V}_{(j,\cdot)}^{(t)} \boldsymbol{v}_k, \tag{27}$$

*where $k \in [M]$, $j \in \mathcal{W}_{n,l} \cup \mathcal{U}_{n,l}$.*

**Lemma 5.** *(Full version of Lemma 1) Given a task $\mathcal{T}$ defined in Definition 2 based on the discriminative pattern $\boldsymbol{\mu}_{\mathcal{T}}$, we have that as long as conditions (i)-(iii) in Theorem 1 hold, then the returned model $\Psi_{\mathcal{T}}^*$ after $T$ iterations achieves a generalization error*

$$\mathbb{E}_{(\boldsymbol{X},y)\sim\mathcal{D}_{\mathcal{T}}}[\ell(\boldsymbol{X}, y; \Psi_{\mathcal{T}}^*)] \leq \Theta(\epsilon). \tag{28}$$

*The required sample complexity is $N = BT$, where $B$ is the batch size. We also have that*

1.

$$p_n(T) \geq 1 - (1 - \delta_*)\delta_*^{-1} T^{-C}, \tag{29}$$

   *for some constant $C > 1$.*

2.

$$\sum_{k=1}^{M} \|\boldsymbol{V}_{(i,\cdot)}^{(T)} \boldsymbol{v}_k\|^2 \lesssim \frac{1}{M} \|\boldsymbol{V}_{(i,\cdot)}^{(T)} \boldsymbol{\mu}_{\mathcal{T}}\|^2, \tag{30}$$

   *for $i \in \mathcal{W}_{n,l}$ with $l \in \mathcal{S}_1^n$ and for $i \in \mathcal{U}_{n,l}$ with $l \in \mathcal{S}_2^n$. We also have that (26) and (27) hold when $t = T$.*

# D PROOF OF MAIN THEOREMS AND COROLLARIES

## D.1 PROOF OF THEOREM 1 AND 2

*Proof.* Since the model is initialized close to zero, then $\Delta\Psi$ is close to $\Psi$. Denote $\Psi_1 = \{\{a_{(l,1)}\}_{l=1}^{P}\}, V_1, W_1\}$ and $\Psi_2 = \{\{a_{(l,2)}\}_{l=1}^{P}\}, V_2, W_2\}$. We consider three cases of this learning problem.

(1) Consider $\alpha = 0$. By (21) in Lemma 3, we know that

$$\boldsymbol{\mu}_{\mathcal{T}_1}^{\top}(\boldsymbol{W}_1^{(T)} + \lambda\boldsymbol{W}_2^{(T)})\boldsymbol{\mu}_{\mathcal{T}_1} = \boldsymbol{\mu}_{\mathcal{T}_1}^{\top}\boldsymbol{W}_1^{(T)}\boldsymbol{\mu}_{\mathcal{T}_1}(1 + \lambda\alpha^2(1 \pm \Theta(\epsilon))) = \boldsymbol{\mu}_{\mathcal{T}_1}^{\top}\boldsymbol{W}_1^{(T)}\boldsymbol{\mu}_{\mathcal{T}_1}, \tag{31}$$

$$-\boldsymbol{\mu}_{\mathcal{T}_1}^{\top}(\boldsymbol{W}_1^{(T)} + \lambda\boldsymbol{W}_2^{(T)})\boldsymbol{\mu}_{\mathcal{T}_1} = -\boldsymbol{\mu}_{\mathcal{T}_1}^{\top}\boldsymbol{W}_1^{(T)}\boldsymbol{\mu}_{\mathcal{T}_1}, \tag{32}$$

$$\boldsymbol{\mu}_{\mathcal{T}_2}^{\top}(\boldsymbol{W}_1^{(T)} + \lambda\boldsymbol{W}_2^{(T)})\boldsymbol{\mu}_{\mathcal{T}_2} = \lambda\boldsymbol{\mu}_{\mathcal{T}_2}^{\top}\boldsymbol{W}_2^{(T)}\boldsymbol{\mu}_{\mathcal{T}_2}, \tag{33}$$

$$-\boldsymbol{\mu}_{\mathcal{T}_2}^{\top}(\boldsymbol{W}_1^{(T)} + \lambda\boldsymbol{W}_2^{(T)})\boldsymbol{\mu}_{\mathcal{T}_2} = -\lambda\boldsymbol{\mu}_{\mathcal{T}_2}^{\top}\boldsymbol{W}_2^{(T)}\boldsymbol{\mu}_{\mathcal{T}_2}. \tag{34}$$

Then, for any $l \in [M]$ and for task $\mathcal{T}_1$,

$$\sum_{s \in \mathcal{S}_1^n} \text{softmax}_l(\boldsymbol{x}_s^{n\top}\boldsymbol{W}^{(T)}\boldsymbol{x}_l^n) \geq 1 - \frac{1-\delta_*}{\delta_*}T^{-C}, \tag{35}$$

for task $\mathcal{T}_2$,

$$\sum_{s \in \mathcal{S}_1^n} \text{softmax}_l(\boldsymbol{x}_s^{n\top}\boldsymbol{W}^{(T)}\boldsymbol{x}_l^n) \geq \frac{\delta_* T^{\lambda C}}{\delta_* T^{\lambda C} + (1 - \delta_*)} \geq 1 - \frac{1-\delta_*}{\delta_*}T^{-\lambda C}. \tag{36}$$

Since that $\boldsymbol{\mu}_{\mathcal{T}_2} \perp \{\boldsymbol{\mu}_{\mathcal{T}_1}, \boldsymbol{v}_1, \boldsymbol{v}_2, \cdots, \boldsymbol{v}_M\}$ and $\boldsymbol{\mu}_{\mathcal{T}_1} \perp \{\boldsymbol{\mu}_{\mathcal{T}_2}, \boldsymbol{v}_1, \boldsymbol{v}_2, \cdots, \boldsymbol{v}_M\}$, we have

$$\boldsymbol{V}_{(i,\cdot)}^{(T)}\boldsymbol{\mu}_{\mathcal{T}_2} = 0, \tag{37}$$

for $\boldsymbol{V} \in \Psi_1$, and

$$\boldsymbol{V}_{(i,\cdot)}^{(T)}\boldsymbol{\mu}_{\mathcal{T}_1} = 0, \tag{38}$$

for $\boldsymbol{V} \in \Psi_2$. Then, for data with the label $y = 1$, the network output for $\Psi_1 + \lambda\Psi_2$ is almost the same as that for $\Psi_1$ on task $\mathcal{T}_1$ when $|\lambda|$ is not too large. To see this, for $\boldsymbol{X}$ from $\mathcal{T}_1$, we have

$$1 - \frac{1}{P}\sum_{l=1}^{P}\sum_{i \in [m]}\frac{1}{a}\text{Relu}((\boldsymbol{V}_{1(i,\cdot)}^{(T)} + \lambda\boldsymbol{V}_{2(i,\cdot)}^{(T)})\boldsymbol{X}\,\text{softmax}_l(\boldsymbol{X}^{n\top}(\boldsymbol{W}_1^{(T)} + \lambda\boldsymbol{W}_2^{(T)})\boldsymbol{x}_l^n))$$

$$\leq |\lambda| \cdot \Theta(\eta\sum_{b=0}^{T-1}\sum_{i \in [m]}\frac{1}{B}\sum_{n \in \mathcal{B}_b}\frac{|\mathcal{S}_1^n|}{aPM}) \cdot \frac{1-\delta_*}{\delta_*}T^{-C} + |\lambda| \cdot \Theta(\sqrt{M\frac{\log B}{B}}) \tag{39}$$

$$\leq |\lambda| \cdot \Theta(1 - \delta_*) \cdot \text{poly}(\eta\delta_*) + |\lambda| \cdot \Theta(\epsilon\sqrt{M})$$

$$= |\lambda|\beta,$$

where the second to last step is by (26) and (27) and $B \gtrsim \epsilon^2 \log M$. Therefore, a larger $|\lambda|$ leads to a performance drop in task $\mathcal{T}_1$. For data of $\mathcal{T}_1$ with the label $y = -1$, we can choose $\lambda$ to be greater than around 1 to make the network output smaller than $-1$. Meanwhile, for $\boldsymbol{X}$ from $\mathcal{T}_2$, we have

$$f(\boldsymbol{X}^n, \Psi)$$
$$\gtrsim (1 - \frac{1-\delta_*}{\delta_*}T^{-C\lambda}) \cdot \lambda - \Theta(\sqrt{\frac{M\log B}{B}}) - \Theta(\frac{1-\delta_*}{\delta_*}) \cdot \text{poly}(\eta\delta_*), \tag{40}$$

where we need $\lambda \geq 1 + \beta$ so that $f(\boldsymbol{X}^n, \Psi) \geq 1 - \Theta(\epsilon)$.

If $\lambda \leq 0$, the attention map tends to be uniform. Then, for $\boldsymbol{X}^n$ in task $\mathcal{T}_2$, we have

$$f(\boldsymbol{X}^n; \Psi_1 + \lambda\Psi_2) \lesssim -\frac{1}{P}, \tag{41}$$

which leads to

$$\mathbb{E}_{(\boldsymbol{X},y)\sim\mathcal{D}_{\mathcal{T}_2}}\ell(\boldsymbol{X}, y; \Psi) \geq \Theta(1). \tag{42}$$

(2) Consider $\alpha > 0$. We first have

$$\boldsymbol{\mu}_{\mathcal{T}_1}^{\top}(\boldsymbol{W}_1^{(T)} + \lambda \boldsymbol{W}_2^{(T)})\boldsymbol{\mu}_{\mathcal{T}_1} = \boldsymbol{\mu}_{\mathcal{T}_1}^{\top}\boldsymbol{W}_1^{(T)}\boldsymbol{\mu}_{\mathcal{T}_1}(1 + \lambda\alpha^2), \tag{43}$$

$$\boldsymbol{\mu}_{\mathcal{T}_2}^{\top}(\boldsymbol{W}_1^{(T)} + \lambda \boldsymbol{W}_2^{(T)})\boldsymbol{\mu}_{\mathcal{T}_2} = (\lambda + \alpha^2)\boldsymbol{\mu}_{\mathcal{T}_2}^{\top}\boldsymbol{W}_2^{(T)}\boldsymbol{\mu}_{\mathcal{T}_2}. \tag{44}$$

Then, for $y^n = 1$ in task $\mathcal{T}_1$, we have that when $\lambda > 0$,

$$f(\boldsymbol{X}^n, \Psi)$$

$$\gtrsim (1 - \Theta(\epsilon)) \cdot (1 + \lambda\alpha) - |\lambda| \cdot \Theta(\eta \sum_{b=0}^{T-1} \sum_{i \in [m]} \frac{1}{B} \sum_{n \in \mathcal{B}_b} \frac{|\mathcal{S}_1^n|}{aPM}) \cdot \frac{1 - \delta_*}{\delta_*} T^{-\lambda C}$$

$$- |\lambda| \cdot \Theta(\sqrt{\frac{M \log B}{B}}) \tag{45}$$

$$\geq 1 + \Theta(\lambda\alpha) - |\lambda| \cdot \Theta(\frac{1 - \delta_*}{\delta_*}) \cdot \text{poly}(\eta\delta_*) - |\lambda| \cdot \Theta(\epsilon\sqrt{M})$$

$$= 1 + \Theta(\lambda\alpha) - |\lambda| \cdot \Theta(\frac{1 - \delta_*}{\delta_*}) \cdot \text{poly}(\eta\delta_*) - |\lambda| \cdot \Theta(\epsilon\sqrt{M}),$$

and for $y^n = 1$ in task $\mathcal{T}_2$, we have that when $\lambda \geq 0$,

$$f(\boldsymbol{X}^n, \Psi) \gtrsim (1 - \frac{1 - \delta_*}{\delta_*} T^{-C(\lambda + \alpha^2)}) \cdot (\lambda + \alpha) - \Theta(\sqrt{\frac{M \log B}{B}})$$

$$- \Theta(\frac{1 - \delta_*}{\delta_*}) \cdot \text{poly}(\eta\delta_*). \tag{46}$$

Therefore, when $\lambda \geq 1 - \alpha + \beta$, we have that for task $\mathcal{T}_1$,

$$f(\boldsymbol{X}^n, \Psi) \geq 1 - |\lambda|\beta - \Theta(\epsilon), \tag{47}$$

and for task $\mathcal{T}_2$,

$$f(\boldsymbol{X}^n, \Psi) \geq (1 - \Theta(\epsilon))(\lambda + \alpha) - \frac{1 - \delta_*}{\delta_*} \cdot \text{poly}(\eta\delta_*) - \Theta(\sqrt{\frac{M \log B}{B}})$$

$$\geq (1 - \Theta(\epsilon))(\lambda + \alpha) - \beta \tag{48}$$

$$\geq 1 - \Theta(\epsilon).$$

We can obtain corresponding conclusions for $y^n = -1$. Hence,

$$\mathbb{E}_{(\boldsymbol{X}, y) \sim \mathcal{D}_{\mathcal{T}_1}} \ell(\boldsymbol{X}, y; \Psi) \leq \Theta(\epsilon) + |\lambda|\beta, \tag{49}$$

$$\mathbb{E}_{(\boldsymbol{X}, y) \sim \mathcal{D}_{\mathcal{T}_2}} \ell(\boldsymbol{X}, y; \Psi) \leq \Theta(\epsilon). \tag{50}$$

Meanwhile, for $y^n = 1$ in task $\mathcal{T}_1$, we have that when $\lambda < 0$,

$$f(\boldsymbol{X}^n, \Psi) \gtrsim (1 - \frac{1 - \delta_*}{\delta_*} T^{-C} - (\frac{1 - \delta_*}{\delta_*} T^{-C(1 + \lambda\alpha^2)} - \frac{1 - \delta_*}{\delta_*} T^{-C})) \cdot (1 + \lambda\alpha)$$

$$- (|\lambda| + 1) \cdot \Theta(\frac{1 - \delta_*}{\delta_*}) \cdot \text{poly}(\eta\delta_*) - |\lambda| \cdot \Theta(\epsilon\sqrt{M})$$

$$\geq 1 + \lambda\alpha(1 - \frac{1 - \delta_*}{\delta_*} T^{-C(1 + \lambda\alpha^2)}) - (\frac{1 - \delta_*}{\delta_*} T^{-C(1 + \lambda\alpha^2)} - \frac{1 - \delta_*}{\delta_*} T^{-C}) \tag{51}$$

$$- (|\lambda| + 1) \cdot \Theta(\frac{1 - \delta_*}{\delta_*}) \cdot \text{poly}(\eta\delta_*) - |\lambda| \cdot \Theta(\epsilon\sqrt{M}),$$

and for $y^n = 1$ in task $\mathcal{T}_2$, we have that when $\lambda < 0$,

$$f(\boldsymbol{X}^n, \Psi) \gtrsim (1 - \frac{1 - \delta_*}{\delta_*} T^{-C(\lambda + \alpha^2)}) \cdot (\lambda + \alpha) - \Theta(\sqrt{\frac{M \log B}{B}}) - \Theta(\frac{1 - \delta_*}{\delta_*}) \cdot \text{poly}(\eta\delta_*)$$

$$\geq (1 - \frac{1 - \delta_*}{\delta_*} T^{-C} - (\frac{1 - \delta_*}{\delta_*} T^{-C(\lambda + \alpha^2)} - \frac{1 - \delta_*}{\delta_*} T^{-C})) \cdot (\lambda + \alpha)$$

$$- \Theta(\sqrt{\frac{M \log B}{B}}) - \Theta(\frac{1 - \delta_*}{\delta_*}) \cdot \text{poly}(\eta\delta_*) \tag{52}$$

$$\geq \lambda + \alpha(1 - \frac{1 - \delta_*}{\delta_*} T^{-C(\lambda + \alpha^2)}) - \lambda(\frac{1 - \delta_*}{\delta_*} T^{-C(\lambda + \alpha^2)} - \frac{1 - \delta_*}{\delta_*} T^{-C})$$

$$- \Theta(\sqrt{\frac{M \log B}{B}}) - \Theta(\frac{1 - \delta_*}{\delta_*}) \cdot \text{poly}(\eta\delta_*).$$

Then, for task $\mathcal{T}_1$, when $0 > \lambda \geq -\Theta(1/\alpha^2)$,

$$
\begin{aligned}
&\mathbb{E}_{(\boldsymbol{X},y)\sim\mathcal{D}_{\mathcal{T}_1}}\ell(\boldsymbol{X},y;\Psi) \\
&= \min\{\Theta(-\lambda\alpha(1 - \frac{1-\delta_*}{\delta_*}T^{-C(1+\lambda\alpha^2)}) + (\frac{1-\delta_*}{\delta_*}T^{-C(1+\lambda\alpha^2)} - \frac{1-\delta_*}{\delta_*}T^{-C}) + \epsilon \\
&\quad + (|\lambda|+1)\cdot\Theta(\frac{1-\delta_*}{\delta_*})\cdot\mathrm{poly}(\eta\delta_*) + |\lambda|\cdot\Theta(\epsilon\sqrt{M})), \Theta(1)\} \\
&\geq \min\{\Theta(-\lambda\alpha + (|\lambda|+1)\cdot\mathrm{poly}(\eta\delta_*) + |\lambda|\cdot\Theta(\epsilon\sqrt{M})), \Theta(1)\} \\
&= \min\{\Theta(-\lambda\alpha + |\lambda|\beta + \mathrm{poly}(\eta\delta_*)), \Theta(1)\},
\end{aligned}
\tag{53}
$$

Hence,

$$
\mathbb{E}_{(\boldsymbol{X},y)\sim\mathcal{D}_{\mathcal{T}_1}}\ell(\boldsymbol{X},y;\Psi) \geq \min\{\Theta(-\lambda\alpha + (1+|\lambda|)\beta), \Theta(1)\}. \tag{54}
$$

When $\lambda < -\Theta(1/\alpha^2)$,

$$
\begin{aligned}
&\mathbb{E}_{(\boldsymbol{X},y)\sim\mathcal{D}_{\mathcal{T}_1}}\ell(\boldsymbol{X},y;\Psi) \\
&= \Theta(1 - \frac{1}{M}\cdot\frac{1}{M}\cdot M) \\
&\geq \Theta(1).
\end{aligned}
\tag{55}
$$

For task $\mathcal{T}_2$, when $0 > \lambda \geq \Theta(1) - \alpha^2$,

$$
\begin{aligned}
&\mathbb{E}_{(\boldsymbol{X},y)\sim\mathcal{D}_{\mathcal{T}_2}}\ell(\boldsymbol{X},y;\Psi) \\
&= \min\{\Theta(1-\lambda-\alpha + \alpha\frac{1-\delta_*}{\delta_*}T^{-C(\lambda+\alpha^2)} + \lambda(\frac{1-\delta_*}{\delta_*}T^{-C(\lambda+\alpha^2)} - \frac{1-\delta_*}{\delta_*}T^{-C}) + \epsilon \\
&\quad + \Theta(\sqrt{\frac{M\log B}{B}}) + \Theta(\frac{1-\delta_*}{\delta_*})\cdot\mathrm{poly}(\eta\delta_*)), \Theta(1)\} \\
&\geq \min\{\Theta(1+\eta^C-\lambda-\alpha + \Theta(\mathrm{poly}(\eta\delta_*) + \epsilon\sqrt{M})), \Theta(1)\} \\
&= \min\{\Theta(1+\eta^C-\lambda-\alpha + \beta), \Theta(1)\},
\end{aligned}
\tag{56}
$$

where the second step is by $\lambda + \alpha \geq \Theta(1) + \alpha - \alpha^2 \geq \Theta(1)$. When $\lambda < \Theta(1) - \alpha^2 < 0$,

$$
\mathbb{E}_{(\boldsymbol{X},y)\sim\mathcal{D}_{\mathcal{T}_2}}\ell(\boldsymbol{X},y;\Psi) \geq \Theta(1). \tag{57}
$$

(3) Consider $\alpha < 0$. When $\lambda \in (-\Theta(1/\alpha^2), 0)$, we have that for task $\mathcal{T}_1$,

$$
\begin{aligned}
&f(\boldsymbol{X}^n, \Psi) \\
&\gtrsim (\frac{1 - \frac{1-\delta_*}{\delta_*}T^{-C(1+\lambda\alpha^2)}}{1 - \frac{1-\delta_*}{\delta_*}T^{-C}} - \Theta(\epsilon))\cdot(1+\lambda\alpha) - |\lambda|\cdot\Theta(\eta\sum_{b=0}^{T-1}\sum_{i\in[m]}\frac{1}{B}\sum_{n\in\mathcal{B}_b}\frac{|\mathcal{S}_1^n|}{aPM}) \\
&\quad \cdot\frac{1-\delta_*}{\delta_*}T^{-\lambda C} - |\lambda|\cdot\Theta(\sqrt{\frac{M\log B}{B}}) \\
&\geq (1-\Theta(\epsilon))\cdot(1+\lambda\alpha) - |\lambda|\cdot\Theta(\frac{1-\delta_*}{\delta_*})\cdot\mathrm{poly}(\eta\delta_*) - |\lambda|\cdot\Theta(\epsilon\sqrt{M}) \\
&\quad - \frac{\frac{1-\delta_*}{\delta_*}(T^{-C(1+\lambda\alpha^2)} - T^{-C})}{1 - \frac{1-\delta_*}{\delta_*}T^{-C}}(1+\lambda\alpha) \\
&\geq (1-\Theta(\epsilon))\cdot(1+\lambda\alpha) - |\lambda|\cdot\Theta(\frac{1-\delta_*}{\delta_*})\cdot\mathrm{poly}(\eta\delta_*) - |\lambda|\cdot\Theta(\epsilon\sqrt{M}) \\
&\quad - \mathrm{poly}(\eta\delta_*)\lambda\alpha^2(-\log\eta\delta_*)(1+\lambda\alpha),
\end{aligned}
\tag{58}
$$

Hence, if $\lambda \leq \mathrm{poly}(\eta\delta_*)\alpha$, we have

$$
f(\boldsymbol{X}^n, \Psi) \geq 1 - |\lambda|\beta - \Theta(\epsilon). \tag{59}
$$

$$
\mathbb{E}_{(\boldsymbol{X},y)\sim\mathcal{D}_{\mathcal{T}_1}}\ell(\boldsymbol{X},y;\Psi) \leq \Theta(\epsilon) + |\lambda|\beta. \tag{60}
$$

If $\lambda > \frac{\beta}{\alpha - \beta}$, we have

$$\mathbb{E}_{(\boldsymbol{X}, y) \sim \mathcal{D}_{\mathcal{T}_1}} \ell(\boldsymbol{X}, y; \Psi) \geq \min\{\Theta(1), \Theta(-\lambda \alpha + (|\lambda| + 1) \cdot \text{poly}(\eta \delta_*) + |\lambda| \cdot \Theta(\epsilon \sqrt{M}))\}. \quad (61)$$

If $\lambda \leq -\Theta(1/\alpha^2)$, we have

$$\mathbb{E}_{(\boldsymbol{X}, y) \sim \mathcal{D}_{\mathcal{T}_1}} \ell(\boldsymbol{X}, y; \Psi) \geq \Theta(1). \quad (62)$$

For task $\mathcal{T}_2$, we have that when $\lambda \geq 1 + \eta^C - \alpha + \beta$,

$$f(\boldsymbol{X}^n, \Psi) \gtrsim (1 - \eta^C)(\lambda + \alpha) - \frac{1 - \delta_*}{\delta_*} \cdot \text{poly}(\eta \delta_*) - \Theta(\sqrt{\frac{M \log B}{B}}) \geq 1, \quad (63)$$

$$\mathbb{E}_{(\boldsymbol{X}, y) \sim \mathcal{D}_{\mathcal{T}_2}} \ell(\boldsymbol{X}, y; \Psi) \leq \Theta(\epsilon). \quad (64)$$

When $\lambda \leq 1 + \eta^C - \alpha + \Theta(\text{poly}(\eta \delta_*) + \epsilon \sqrt{M})$,

$$\mathbb{E}_{(\boldsymbol{X}, y) \sim \mathcal{D}_{\mathcal{T}_2}} \ell(\boldsymbol{X}, y; \Psi) \geq \min\{\Theta(1), 1 + \eta^C - \lambda - \alpha + \beta\}. \quad (65)$$

One can easily find that there is no region of $\lambda$ such that $\Psi$ performs well on both $\mathcal{T}_1$ and $\mathcal{T}_2$. However, when $-\Theta(1/\alpha^2) < \lambda < \text{poly}(\eta \delta_*)\alpha < 1 + \eta^c - \alpha + \beta$, we can unlearn $\mathcal{T}_2$ and retain the performance of $\mathcal{T}_1$.

$\square$

## D.2 PROOF OF THEOREM 3

*Proof.* By Lemma 1, we know that

$$\begin{aligned}
&\boldsymbol{\mu}_{\mathcal{T}'}{}^\top \boldsymbol{W}^{(T)} \boldsymbol{\mu}_{\mathcal{T}'} \\
&= \sum_{i \in \mathcal{V}_\Psi} \gamma_i \boldsymbol{\mu}_{\mathcal{T}_i}^\top (\sum_{j=1} \lambda_j \boldsymbol{W}_j^{(T)}) \sum_{k \in \mathcal{V}_\Psi} \gamma_k \boldsymbol{\mu}_{\mathcal{T}_k} \\
&\gtrsim \sum_{i \in \mathcal{V}_\Psi} \gamma_i^2 \boldsymbol{\mu}_{\mathcal{T}_i}^\top \cdot \lambda_i \boldsymbol{W}_i^{(T)} \boldsymbol{\mu}_{\mathcal{T}_i}.
\end{aligned} \quad (66)$$

For positive neurons, we also have

$$\boldsymbol{V}^{(T)} \boldsymbol{\mu}_{\mathcal{T}'} = \sum_{i \in \mathcal{V}_\Psi} \lambda_i \boldsymbol{V}_{\mathcal{T}_i}{}^{(T)} \sum_{i \in \mathcal{V}'} \gamma_i \boldsymbol{\mu}_{\mathcal{T}_i} = \sum_{i \in \mathcal{V}_\Psi} \lambda_i \gamma_i \boldsymbol{V}_{\mathcal{T}_i}{}^{(T)} \boldsymbol{\mu}_{\mathcal{T}_i} \quad (67)$$

Then, we need

$$\sum_{i \in \mathcal{V}_\Psi} \lambda_i \gamma_i \geq 1 + c, \quad (68)$$

$$\sum_{i \in \mathcal{V}_\Psi} \lambda_i \gamma_i^2 \geq 1 + c, \quad (69)$$

$$|\lambda_i| (\Theta(\frac{1 - \delta_*}{\delta_*} \text{poly}(\eta \delta_*) + \epsilon \sqrt{M})) = |\lambda_i| \beta \leq c, \text{ for some } c > 0 \text{ and all } i \in \mathcal{V}_\Psi, \quad (70)$$

to hold simultaneously.

Then, when $\gamma_i = k$ does not hold for all $i \in \mathcal{V}_\Psi$ and for some fixed $k < 0$, we can find $\lambda_i$ in the middle of the normalized $\gamma_i$ and $\gamma_i^2$ to satisfy (68) and (69), i.e.,

$$\lambda_i \propto \frac{\gamma_i}{\sqrt{\sum_{i \in \mathcal{V}_\Psi} \gamma_i^2}} + \frac{\gamma_i^2}{\sqrt{\sum_{i \in \mathcal{V}_\Psi} \gamma_i^4}}. \quad (71)$$

By Cauchy–Schwarz inequality, we have

$$-\sqrt{\sum_{i \in \mathcal{V}_\Psi} \gamma_i^2} \cdot \sqrt{\sum_{i \in \mathcal{V}_\Psi} \gamma_i^4} < \sum_{i \in \mathcal{V}_\Psi} \gamma_i^3 < \sqrt{\sum_{i \in \mathcal{V}_\Psi} \gamma_i^2} \cdot \sqrt{\sum_{i \in \mathcal{V}_\Psi} \gamma_i^4}. \quad (72)$$

Hence,

$$\sum_{i \in \mathcal{V}_\Psi} \lambda_i \gamma_i \propto \sqrt{\sum_{i \in \mathcal{V}_\Psi} \gamma_i^2} + \frac{\sum_{i \in \mathcal{V}_\Psi} \gamma_i^3}{\sqrt{\sum_{i \in \mathcal{V}_\Psi} \gamma_i^4}} = \frac{\sqrt{\sum_{i \in \mathcal{V}_\Psi} \gamma_i^2} \cdot \sqrt{\sum_{i \in \mathcal{V}_\Psi} \gamma_i^4} + \sum_{i \in \mathcal{V}_\Psi} \gamma_i^3}{\sqrt{\sum_{i \in \mathcal{V}_\Psi} \gamma_i^4}} > 0, \quad (73)$$

$$\sum_{i \in \mathcal{V}_\Psi} \lambda_i \gamma_i^2 \propto \frac{\sum_{i \in \mathcal{V}_\Psi} \gamma_i^3}{\sqrt{\sum_{i \in \mathcal{V}_\Psi} \gamma_i^2}} + \sqrt{\sum_{i \in \mathcal{V}_\Psi} \gamma_i^4} = \frac{\sqrt{\sum_{i \in \mathcal{V}_\Psi} \gamma_i^2} \cdot \sqrt{\sum_{i \in \mathcal{V}_\Psi} \gamma_i^4} + \sum_{i \in \mathcal{V}_\Psi} \gamma_i^3}{\sqrt{\sum_{i \in \mathcal{V}_\Psi} \gamma_i^2}} > 0. \quad (74)$$

Therefore, by letting

$$\lambda_i = C_\gamma \cdot \left( \frac{\gamma_i}{\sqrt{\sum_{i \in \mathcal{V}_\Psi} \gamma_i^2}} + \frac{\gamma_i^2}{\sqrt{\sum_{i \in \mathcal{V}_\Psi} \gamma_i^4}} \right), \quad (75)$$

where

$$C_\gamma = \frac{(1+c)\sqrt{\sum_{i \in \mathcal{V}_\Psi} \gamma_i^4}}{\sqrt{\sum_{i \in \mathcal{V}_\Psi} \gamma_i^2} \cdot \sqrt{\sum_{i \in \mathcal{V}_\Psi} \gamma_i^4} + \sum_{i \in \mathcal{V}_\Psi} \gamma_i^3}, \quad (76)$$

we can obtain (68) and (69) hold if $C_\gamma \lesssim \beta^{-1}$.

When $\gamma_i = k$ hold for all $i \in \mathcal{V}_\Psi$ and for some fixed $k < 0$ with $|\mathcal{V}_\Psi| > 0$, we cannot find $\lambda_i$ such that both (68) and (69) hold.

$\square$

### D.3    PROOF OF COROLLARY 1

*Proof.* Let $\{\boldsymbol{\mu}_1, \boldsymbol{v}_1, \boldsymbol{v}_2, \cdots, \boldsymbol{v}_M\} \cup \{\boldsymbol{u}_1, \boldsymbol{u}_2, \cdots, \boldsymbol{u}_{d-M+1}\}$ form a set of orthonormal vectors, which is denoted by

$$\boldsymbol{U} = (\boldsymbol{\mu}_1, \boldsymbol{v}_1, \boldsymbol{v}_2, \cdots, \boldsymbol{v}_M, \boldsymbol{u}_1, \boldsymbol{u}_2, \cdots, \boldsymbol{u}_{d-M+1}). \quad (77)$$

Note that for any $\boldsymbol{a}, \boldsymbol{b} \in \{\boldsymbol{\mu}_1, \boldsymbol{v}_1, \boldsymbol{v}_2, \cdots, \boldsymbol{v}_M\} \cup \{\boldsymbol{u}_1, \boldsymbol{u}_2, \cdots, \boldsymbol{u}_{d-M+1}\}$,

$$\boldsymbol{a}^\top \boldsymbol{W}^{(0)} \boldsymbol{b} = \sum_{1 \le i,j \le d} a_i b_j W_{i,j}^{(0)} \sim \mathcal{N}(0, \sum_{1 \le i,j \le d} |a_i b_j| \xi^2), \quad (78)$$

where the last step comes from that each entry of $\boldsymbol{W}^{(0)} \sim \mathcal{N}(0, \xi^2)$. Given that $\|\boldsymbol{a}\| = \|\boldsymbol{b}\| = 1$, we have

$$\sum_{1 \le i,j \le d} |a_i b_j| = (|a_1|, \cdots, |a_d|)^\top (|b_1|, \cdots, |b_d|) \le 1. \quad (79)$$

By (90), we know that for $\boldsymbol{a} \in \{\boldsymbol{u}_1, \boldsymbol{u}_2, \cdots, \boldsymbol{u}_{d-M+1}\}$ and any $t = 0, 1, \cdots, T-1$,

$$\eta \frac{1}{B} \sum_{n \in \mathcal{B}_b} \frac{\partial \ell(\boldsymbol{X}^n, y^n; \Psi)}{\partial \boldsymbol{W}^{(t)}} \boldsymbol{a} = 0, \quad (80)$$

$$\boldsymbol{a}^\top \eta \frac{1}{B} \sum_{n \in \mathcal{B}_b} \frac{\partial \ell(\boldsymbol{X}^n, y^n; \Psi)}{\partial \boldsymbol{W}^{(t)}} = 0. \quad (81)$$

Then, we have that for some $C > 1$,

$$[\boldsymbol{U}^\top \boldsymbol{W}^{(T)} \boldsymbol{U}]_{i,j} = \begin{cases} \Theta(\log T), & i = j = 1, \\ O(\epsilon \cdot \frac{1}{e^{\Theta(\log T)} \cdot (1 - \frac{1-\delta_*}{\delta_*} T^{-C})}) = O(\epsilon \cdot T^{-C}), & j = 1, 1 \le i \le M-1, \\ O(\epsilon \cdot \log T), & j \in [2, M-1], i \in [1, M-1], \\ O(\xi), & \text{else.} \end{cases} \quad (82)$$

Let $\boldsymbol{E}_{i,j}$ be the matrix that only the $(i,j)$ entry equals 1, while all other entries are 0. Therefore,

$$
\begin{aligned}
&\|\boldsymbol{U}^\top \boldsymbol{W}^{(T)} \boldsymbol{U} - \boldsymbol{E}_{1,1} \cdot \Theta(\log T)\|_F^2 \\
&\leq (\epsilon \cdot T^{-C})^2 \cdot (M-1) + (\epsilon \cdot \log T)^2 \cdot (M-1)(M-2) + \xi^2 (d^2 - M^2) \\
&\leq \epsilon^2 \log^2 T \cdot M^2 + d^2/m \\
&\lesssim \epsilon^2 \cdot M^2 + \frac{1}{\log M},
\end{aligned}
\tag{83}
$$

where the last step comes from that $m \gtrsim M^2 \log M$ and $M = \Theta(d)$. Then,

$$
\begin{aligned}
&\|\boldsymbol{W}^{(T)} - \boldsymbol{U} \boldsymbol{E}_{1,1} \cdot \Theta(\log T) \cdot \boldsymbol{U}^\top\|_F \\
&\leq \|\boldsymbol{W}^{(T)} \boldsymbol{U} - \boldsymbol{U} \boldsymbol{E}_{1,1} \cdot \Theta(\log T)\|_F \cdot \|\boldsymbol{U}^\top\| \\
&\leq \|\boldsymbol{U}\| \cdot \|\boldsymbol{U}^\top \boldsymbol{W}^{(T)} \boldsymbol{U} - \boldsymbol{E}_{1,1} \cdot \Theta(\log T)\|_F \\
&\leq \epsilon M + 1/\log M.
\end{aligned}
\tag{84}
$$

Likewise, by (132), we know that neurons of $\boldsymbol{V}^{(T)}$ with a non-trivial magnitude are in the direction of the iterative summation of $\left( \sum_{s=1}^P \boldsymbol{x}_s^n \mathrm{softmax}_l(\boldsymbol{x}_s^{n\top} \boldsymbol{W} \boldsymbol{x}_l^n) \right)$. Hence, there exists $\hat{\boldsymbol{v}}_1 \in \mathbb{R}^m$ and $\hat{\boldsymbol{v}}_2 \in \mathbb{R}^d$ such that

$$
\|\boldsymbol{V}^{(T)} - \hat{\boldsymbol{v}}_1 \hat{\boldsymbol{v}}_2^\top\|_F \leq \Theta(1) \cdot \sqrt{m} \cdot \sqrt{\frac{\log B}{B}} \cdot \delta_*^{-2} \cdot \delta_* \cdot \frac{1}{\sqrt{m}} \leq \delta_*^{-1} \epsilon
\tag{85}
$$

Then, for $n$ such that $y^n = +1$, we have that the low-rank trained model, where $\boldsymbol{W}_{LR}^{(T)} = \boldsymbol{U} \boldsymbol{E}_{1,1} \cdot \Theta(\log T) \cdot \boldsymbol{U}^\top$, satisfies

$$
f(\boldsymbol{X}^n, \Psi_{LR}) \geq 1 \cdot (1 - \delta_* \epsilon) \cdot (1 - \Theta(\epsilon \log T)) = 1 - \Theta((\log T + \delta_*)\epsilon),
\tag{86}
$$

which leads to

$$
\ell(\boldsymbol{X}^n, y^n; \Psi_{LR}) \leq \Theta(\epsilon_{LR}), \text{ where } \epsilon_{LR} = (\log T + \delta_*)\epsilon.
\tag{87}
$$

$\square$

## D.4 PROOF OF COROLLARY 2

*Proof.* We know that from Lemma 1, there is a number of $\Omega(m)$ lucky neurons with large weights. We can denote the set of lucky neurons as $\mathcal{L} \subset [m]$. By combining (148) and (163), we have that for any lucky neuron $\boldsymbol{u}_i$,

$$
\|\boldsymbol{u}_i\| \geq \eta \eta^{-1} \delta_*^{-1} \cdot \delta_* \cdot \frac{1}{\sqrt{m}} = m^{-1/2}.
\tag{88}
$$

For any unlucky neurons, by (149), we have

$$
\|\boldsymbol{u}_i\| \leq m^{-1/2} \sqrt{\frac{\log B}{B}}.
\tag{89}
$$

Since that $B \geq \epsilon^{-2} \log M$ by Lemma 1, we have that if we remove neurons from $m \backslash \mathcal{L}$, the output in (158) and (159) will only be affected by a factor of $\epsilon$. Therefore, Lemma 1 still holds, so that Theorems 1-3 all hold. $\square$

# E PROOF OF KEY LEMMAS

## E.1 PROOF OF LEMMA 3

For ease of presentation, we sometimes use $\boldsymbol{\mu}_2$ to represent $-\boldsymbol{\mu}_1$ in the proof. We first investigate the gradient of $\boldsymbol{W}$, i.e.,

$$
\begin{aligned}
&\eta \frac{1}{B} \sum_{n \in \mathcal{B}_b} \frac{\partial \ell(\boldsymbol{X}^n, y^n; \Psi)}{\partial \boldsymbol{W}} \\
={}&\eta \frac{1}{B} \sum_{n \in \mathcal{B}_b} \frac{\partial \ell(\boldsymbol{X}^n, y^n; \Psi)}{\partial f(\boldsymbol{X}^n; \Psi)} \frac{f(\boldsymbol{X}^n; \Psi)}{\partial \boldsymbol{W}} \\
={}&\eta \frac{1}{B} \sum_{n \in \mathcal{B}_b} (-y^n) \frac{1}{P} \sum_{l=1}^{P} \sum_{i=1}^{m} a_{(l)_i} \mathbb{1}[\boldsymbol{V}_{(i,\cdot)} \boldsymbol{X} \operatorname{softmax}_l(\boldsymbol{X}^{n\top} \boldsymbol{W} \boldsymbol{x}_l^n) \geq 0] \\
&\cdot \left( \boldsymbol{V}_{(i,\cdot)} \sum_{s=1}^{P} \boldsymbol{x}_s^n \operatorname{softmax}_l(\boldsymbol{x}_s^{n\top} \boldsymbol{W} \boldsymbol{x}_l^n) \sum_{r=1}^{P} \operatorname{softmax}_l(\boldsymbol{x}_r^{n\top} \boldsymbol{W} \boldsymbol{x}_l^n)(\boldsymbol{x}_s^n - \boldsymbol{x}_r^n)\boldsymbol{x}_l^{n\top} \right) \\
={}&\eta \frac{1}{B} \sum_{n \in \mathcal{B}_b} (-y^n) \frac{1}{P} \sum_{l=1}^{P} \sum_{i=1}^{m} a_{(l)_i} \mathbb{1}[\boldsymbol{V}_{(i,\cdot)} \boldsymbol{X}^n \operatorname{softmax}_l(\boldsymbol{X}^{n\top} \boldsymbol{W} \boldsymbol{x}_l^n) \geq 0] \\
&\cdot \left( \boldsymbol{V}_{(i,\cdot)} \sum_{s=1}^{P} \boldsymbol{x}_s^n \operatorname{softmax}_l(\boldsymbol{x}_s^{n\top} \boldsymbol{W} \boldsymbol{x}_l^n) \cdot (\boldsymbol{x}_s^n - \sum_{r=1}^{P} \operatorname{softmax}_l(\boldsymbol{x}_r^{n\top} \boldsymbol{W} \boldsymbol{x}_l^n)\boldsymbol{x}_r^n)\boldsymbol{x}_l^{n\top} \right)
\end{aligned}
\tag{90}
$$

For $j, l \in \mathcal{S}_1^n$, we have

$$
\operatorname{softmax}_l(\boldsymbol{x}_j^{n\top} \boldsymbol{W}^{(t)} \boldsymbol{x}_l^n) \gtrsim \frac{e^{\|\boldsymbol{q}_1(t)\|}}{|\mathcal{S}_1^n|e^{\|\boldsymbol{q}_1(t)\|} + (P - |\mathcal{S}_1^n|)}
\tag{91}
$$

For $j \notin \mathcal{S}_1^n$ and $l \in \mathcal{S}_1^n$, we have

$$
\operatorname{softmax}_l(\boldsymbol{x}_j^{n\top} \boldsymbol{W}^{(t)} \boldsymbol{x}_l^n) \lesssim \frac{1}{|\mathcal{S}_1^n|e^{\|\boldsymbol{q}_1(t)\|} + (P - |\mathcal{S}_1^n|)},
\tag{92}
$$

where $\|\boldsymbol{q}_1(0)\| = 0$. For $l \notin \mathcal{S}_1^n \cup \mathcal{S}_2^n$, $j \in [P]$, we have

$$
\operatorname{softmax}_l(\boldsymbol{x}_j^{n\top} \boldsymbol{W}^{(0)} \boldsymbol{x}_l^n) \lesssim \frac{1}{P}.
\tag{93}
$$

Therefore, for $s, r, l \in \mathcal{S}_1^n$, let

$$
\boldsymbol{x}_s^n - \sum_{r=1}^{P} \operatorname{softmax}_l(\boldsymbol{x}_r^{n\top} \boldsymbol{W}^{(t)} \boldsymbol{x}_l^n)\boldsymbol{x}_r^n := \beta_1^n(t)\boldsymbol{\mu}_1 + \boldsymbol{\beta}_2^n(t),
\tag{94}
$$

where

$$
\beta_1^n(t) \gtrsim \frac{P - |\mathcal{S}_1^n|}{|\mathcal{S}_1^n|e^{\|\boldsymbol{q}_1(t)\|} + P - |\mathcal{S}_1^n|} := \phi_n(t)(P - |\mathcal{S}_1^n|).
\tag{95}
$$

$$
\boldsymbol{\beta}_2^n(t) = \sum_{l=2}^{M_1} \iota_l' \boldsymbol{\mu}_l,
\tag{96}
$$

where

$$
|\iota_l'| \leq \beta_1^n(t) \frac{|\mathcal{S}_l^n|}{P - |\mathcal{S}_1^n|}.
\tag{97}
$$

Note that $|\iota_l'| = 0$ if $P = |\mathcal{S}_1^n|$, $l \geq 2$.
If $s \in \mathcal{S}_1^n$, we have

$$
\boldsymbol{V}_{(i,\cdot)}^{(t)} \boldsymbol{x}_s^n \operatorname{softmax}_l(\boldsymbol{x}_s^{n\top} \boldsymbol{W} \boldsymbol{x}_l^n) \geq \zeta_{i,1,t} \cdot \frac{p_n(t)}{|\mathcal{S}_1^n|}.
\tag{98}
$$

If $s \in \mathcal{S}_2^n$ and $j \in \mathcal{S}_1^n$, we have

$$\boldsymbol{V}_{(i,\cdot)}^{(t)} \boldsymbol{x}_s^n \mathrm{softmax}_l(\boldsymbol{x}_s^{n\top} \boldsymbol{W}^{(t)} \boldsymbol{x}_l^n) \lesssim \boldsymbol{V}_{(i,\cdot)}^{(t)} \boldsymbol{x}_j^n \mathrm{softmax}_l(\boldsymbol{x}_j^{n\top} \boldsymbol{W}^{(t)} \boldsymbol{x}_l^n) \phi_n(t) \cdot \frac{|\mathcal{S}_1^n|}{p_n(t)}. \tag{99}$$

If $s \notin (\mathcal{S}_1^n \cup \mathcal{S}_2^n)$ and $j \in \mathcal{S}_1^n$,

$$\boldsymbol{V}_{(i,\cdot)}^{(t)} \boldsymbol{x}_s^n \mathrm{softmax}_l(\boldsymbol{x}_s^{n\top} \boldsymbol{W}^{(t)} \boldsymbol{x}_l^n) \lesssim \boldsymbol{V}_{(i,\cdot)}^{(t)} \boldsymbol{x}_j^n \mathrm{softmax}_l(\boldsymbol{x}_j^{n\top} \boldsymbol{W}^{(t)} \boldsymbol{x}_l^n) \phi_n(t) \cdot \frac{|\mathcal{S}_1^n|}{\sqrt{B} p_n(t)}. \tag{100}$$

Then, by combining (94) to (100), we have that for $l \in \mathcal{S}_1^n$, $i \in \mathcal{W}_{n,l}$,

$$\boldsymbol{\mu}_1^\top \boldsymbol{V}_{(i,\cdot)} \sum_{s=1}^P \boldsymbol{x}_s^n \mathrm{softmax}_l(\boldsymbol{x}_s^{n\top} \boldsymbol{W} \boldsymbol{x}_l^n) \cdot (\boldsymbol{x}_s^n - \sum_{r=1}^P \mathrm{softmax}_l(\boldsymbol{x}_r^{n\top} \boldsymbol{W} \boldsymbol{x}_l^n) \boldsymbol{x}_r^n) \boldsymbol{x}_l^{n\top} \boldsymbol{\mu}_1 \tag{101}$$
$$\gtrsim \zeta_{i,1,t} \cdot p_n(t) \phi_n(t)(P - |\mathcal{S}_1^n|).$$

For $l \in \mathcal{S}_1^n$, $i \in \mathcal{W}_{n,l}$, we have that for $k \neq 1, 2$,

$$\boldsymbol{\mu}_2^\top \boldsymbol{V}_{(i,\cdot)} \sum_{s=1}^P \boldsymbol{x}_s^n \mathrm{softmax}_l(\boldsymbol{x}_s^{n\top} \boldsymbol{W} \boldsymbol{x}_l^n) \cdot (\boldsymbol{x}_s^n - \sum_{r=1}^P \mathrm{softmax}_l(\boldsymbol{x}_r^{n\top} \boldsymbol{W} \boldsymbol{x}_l^n) \boldsymbol{x}_r^n) \boldsymbol{x}_l^{n\top} \boldsymbol{\mu}_1 \tag{102}$$
$$= -\boldsymbol{\mu}_1^\top \boldsymbol{V}_{(i,\cdot)} \sum_{s=1}^P \boldsymbol{x}_s^n \mathrm{softmax}_l(\boldsymbol{x}_s^{n\top} \boldsymbol{W} \boldsymbol{x}_l^n) \cdot (\boldsymbol{x}_s^n - \sum_{r=1}^P \mathrm{softmax}_l(\boldsymbol{x}_r^{n\top} \boldsymbol{W} \boldsymbol{x}_l^n) \boldsymbol{x}_r^n) \boldsymbol{x}_l^{n\top} \boldsymbol{\mu}_1.$$

For $l \in \mathcal{S}_1^n$, $i \in \mathcal{W}_{n,l}$, we have that for $k \in [M]$,

$$\boldsymbol{v}_k^\top \boldsymbol{V}_{(i,\cdot)} \sum_{s=1}^P \boldsymbol{x}_s^n \mathrm{softmax}_l(\boldsymbol{x}_s^{n\top} \boldsymbol{W} \boldsymbol{x}_l^n) \cdot (\boldsymbol{x}_s^n - \sum_{r=1}^P \mathrm{softmax}_l(\boldsymbol{x}_r^{n\top} \boldsymbol{W} \boldsymbol{x}_l^n) \boldsymbol{x}_r^n) \boldsymbol{x}_l^{n\top} \boldsymbol{\mu}_1$$
$$\leq \boldsymbol{\mu}_1^\top \boldsymbol{V}_{(i,\cdot)} \sum_{s=1}^P \boldsymbol{x}_s^n \mathrm{softmax}_l(\boldsymbol{x}_s^{n\top} \boldsymbol{W} \boldsymbol{x}_l^n) \cdot (\boldsymbol{x}_s^n - \sum_{r=1}^P \mathrm{softmax}_l(\boldsymbol{x}_r^{n\top} \boldsymbol{W} \boldsymbol{x}_l^n) \boldsymbol{x}_r^n) \boldsymbol{x}_l^{n\top} \boldsymbol{\mu}_1 \tag{103}$$
$$\cdot \frac{|\mathcal{R}_k^n|}{P - |\mathcal{S}_1^n|} \cdot \frac{|\mathcal{S}_1^n| \phi_n(t)}{p_n(t)}.$$

For $i \in \mathcal{U}_{n,l}$, by the definition of $\mathcal{U}_{n,l}$ in Definition 4, we have

$$\mathbb{1}[\boldsymbol{V}_{(i,\cdot)} \boldsymbol{X}^n \mathrm{softmax}_l(\boldsymbol{X}^{n\top} \boldsymbol{W} \boldsymbol{x}_l^n) \geq 0] = 0. \tag{104}$$

For $i \notin \mathcal{W}_{n,l} \cup \mathcal{U}_{n,l}$, we have that for $j \in \mathcal{W}_{n,l}$, $k \in [M]$,

$$\boldsymbol{\mu}_1^\top \boldsymbol{V}_{(i,\cdot)} \sum_{s=1}^P \boldsymbol{x}_s^n \mathrm{softmax}_l(\boldsymbol{x}_s^{n\top} \boldsymbol{W} \boldsymbol{x}_l^n) \cdot (\boldsymbol{x}_s^n - \sum_{r=1}^P \mathrm{softmax}_l(\boldsymbol{x}_r^{n\top} \boldsymbol{W} \boldsymbol{x}_l^n) \boldsymbol{x}_r^n) \boldsymbol{x}_l^{n\top} \boldsymbol{\mu}_1$$
$$\leq \boldsymbol{\mu}_1^\top \boldsymbol{V}_{(j,\cdot)} \sum_{s=1}^P \boldsymbol{x}_s^n \mathrm{softmax}_l(\boldsymbol{x}_s^{n\top} \boldsymbol{W} \boldsymbol{x}_l^n) \cdot (\boldsymbol{x}_s^n - \sum_{r=1}^P \mathrm{softmax}_l(\boldsymbol{x}_r^{n\top} \boldsymbol{W} \boldsymbol{x}_l^n) \boldsymbol{x}_r^n) \boldsymbol{x}_l^{n\top} \boldsymbol{\mu}_1 \tag{105}$$
$$\cdot \phi_n(t) \frac{|\mathcal{S}_1^n|}{\sqrt{B} p_n(t)}.$$

$$\boldsymbol{\mu}_2^\top \boldsymbol{V}_{(i,\cdot)} \sum_{s=1}^P \boldsymbol{x}_s^n \mathrm{softmax}_l(\boldsymbol{x}_s^{n\top} \boldsymbol{W} \boldsymbol{x}_l^n) \cdot (\boldsymbol{x}_s^n - \sum_{r=1}^P \mathrm{softmax}_l(\boldsymbol{x}_r^{n\top} \boldsymbol{W} \boldsymbol{x}_l^n) \boldsymbol{x}_r^n) \boldsymbol{x}_l^{n\top} \boldsymbol{\mu}_1 \tag{106}$$
$$= -\boldsymbol{\mu}_1^\top \boldsymbol{V}_{(i,\cdot)} \sum_{s=1}^P \boldsymbol{x}_s^n \mathrm{softmax}_l(\boldsymbol{x}_s^{n\top} \boldsymbol{W} \boldsymbol{x}_l^n) \cdot (\boldsymbol{x}_s^n - \sum_{r=1}^P \mathrm{softmax}_l(\boldsymbol{x}_r^{n\top} \boldsymbol{W} \boldsymbol{x}_l^n) \boldsymbol{x}_r^n) \boldsymbol{x}_l^{n\top} \boldsymbol{\mu}_1.$$

$$\boldsymbol{v}_k^\top \boldsymbol{V}_{(i,\cdot)} \sum_{s=1}^P \boldsymbol{x}_s^n \mathrm{softmax}_l(\boldsymbol{x}_s^{n\top} \boldsymbol{W} \boldsymbol{x}_l^n) \cdot (\boldsymbol{x}_s^n - \sum_{r=1}^P \mathrm{softmax}_l(\boldsymbol{x}_r^{n\top} \boldsymbol{W} \boldsymbol{x}_l^n) \boldsymbol{x}_r^n) \boldsymbol{x}_l^{n\top} \boldsymbol{\mu}_1$$
$$\leq \boldsymbol{\mu}_1^\top \boldsymbol{V}_{(j,\cdot)} \sum_{s=1}^P \boldsymbol{x}_s^n \mathrm{softmax}_l(\boldsymbol{x}_s^{n\top} \boldsymbol{W} \boldsymbol{x}_l^n) \cdot (\boldsymbol{x}_s^n - \sum_{r=1}^P \mathrm{softmax}_l(\boldsymbol{x}_r^{n\top} \boldsymbol{W} \boldsymbol{x}_l^n) \boldsymbol{x}_r^n) \boldsymbol{x}_l^{n\top} \boldsymbol{\mu}_1 \tag{107}$$
$$\cdot \phi_n(t) \frac{|\mathcal{S}_1^n|}{\sqrt{B} p_n(t)} \cdot \frac{|\mathcal{R}_k^n|}{P - |\mathcal{S}_1^n|}.$$

When $l \notin \mathcal{S}_1^n$, we have that $\boldsymbol{x}_l^{n\top}\boldsymbol{\mu}_1 = 0$. If $l \in \mathcal{S}_2^n$, we can obtain that

$$
\boldsymbol{\mu}_2^\top \boldsymbol{V}_{(i,\cdot)} \sum_{s=1}^P \boldsymbol{x}_s^n \mathrm{softmax}_l(\boldsymbol{x}_s^{n\top}\boldsymbol{W}\boldsymbol{x}_l^n) \cdot (\boldsymbol{x}_s^n - \sum_{r=1}^P \mathrm{softmax}_l(\boldsymbol{x}_r^{n\top}\boldsymbol{W}\boldsymbol{x}_l^n)\boldsymbol{x}_r^n)\boldsymbol{x}_l^{n\top}\boldsymbol{\mu}_2
$$
$$
\gtrsim \zeta_{i,1,t} \cdot \frac{p_n(t)|\mathcal{S}_2^n|}{|\mathcal{S}_1^n|}\phi_n(t)(P - |\mathcal{S}_1^n|),
$$
(108)

$$
\boldsymbol{\mu}_1^\top \boldsymbol{V}_{(i,\cdot)} \sum_{s=1}^P \boldsymbol{x}_s^n \mathrm{softmax}_l(\boldsymbol{x}_s^{n\top}\boldsymbol{W}\boldsymbol{x}_l^n) \cdot (\boldsymbol{x}_s^n - \sum_{r=1}^P \mathrm{softmax}_l(\boldsymbol{x}_r^{n\top}\boldsymbol{W}\boldsymbol{x}_l^n)\boldsymbol{x}_r^n)\boldsymbol{x}_l^{n\top}\boldsymbol{\mu}_2
$$
$$
= - \boldsymbol{\mu}_2^\top \boldsymbol{V}_{(i,\cdot)} \sum_{s=1}^P \boldsymbol{x}_s^n \mathrm{softmax}_l(\boldsymbol{x}_s^{n\top}\boldsymbol{W}\boldsymbol{x}_l^n) \cdot (\boldsymbol{x}_s^n - \sum_{r=1}^P \mathrm{softmax}_l(\boldsymbol{x}_r^{n\top}\boldsymbol{W}\boldsymbol{x}_l^n)\boldsymbol{x}_r^n)\boldsymbol{x}_l^{n\top}\boldsymbol{\mu}_2,
$$
(109)

$$
\boldsymbol{v}_k^\top \boldsymbol{V}_{(i,\cdot)} \sum_{s=1}^P \boldsymbol{x}_s^n \mathrm{softmax}_l(\boldsymbol{x}_s^{n\top}\boldsymbol{W}\boldsymbol{x}_l^n) \cdot (\boldsymbol{x}_s^n - \sum_{r=1}^P \mathrm{softmax}_l(\boldsymbol{x}_r^{n\top}\boldsymbol{W}\boldsymbol{x}_l^n)\boldsymbol{x}_r^n)\boldsymbol{x}_l^{n\top}\boldsymbol{\mu}_2
$$
$$
\leq \boldsymbol{\mu}_2^\top \boldsymbol{V}_{(i,\cdot)} \sum_{s=1}^P \boldsymbol{x}_s^n \mathrm{softmax}_l(\boldsymbol{x}_s^{n\top}\boldsymbol{W}\boldsymbol{x}_l^n) \cdot (\boldsymbol{x}_s^n - \sum_{r=1}^P \mathrm{softmax}_l(\boldsymbol{x}_r^{n\top}\boldsymbol{W}\boldsymbol{x}_l^n)\boldsymbol{x}_r^n)\boldsymbol{x}_l^{n\top}\boldsymbol{\mu}_2
$$
$$
\cdot \frac{|\mathcal{R}_k^n|}{P - |\mathcal{S}_2^n|}\frac{|\mathcal{S}_1^n|\phi_n(t)}{p_n(t)},
$$
(110)

where $k \in [M]$, $i \in \mathcal{U}_{n,l}$. If $i \in \mathcal{W}_{n,l}$,

$$
\mathbb{1}[\boldsymbol{V}_{(i,\cdot)}\boldsymbol{X}^n \mathrm{softmax}_l(\boldsymbol{X}^{n\top}\boldsymbol{W}\boldsymbol{x}_l^n) \geq 0] = 0.
$$
(111)

If $i \notin \mathcal{W}_{n,l} \cup \mathcal{U}_{n,l}$, we have that for $j \in \mathcal{U}_{n,l}$, $k \in [M]$,

$$
\boldsymbol{\mu}_2^\top \boldsymbol{V}_{(i,\cdot)} \sum_{s=1}^P \boldsymbol{x}_s^n \mathrm{softmax}_l(\boldsymbol{x}_s^{n\top}\boldsymbol{W}\boldsymbol{x}_l^n) \cdot (\boldsymbol{x}_s^n - \sum_{r=1}^P \mathrm{softmax}_l(\boldsymbol{x}_r^{n\top}\boldsymbol{W}\boldsymbol{x}_l^n)\boldsymbol{x}_r^n)\boldsymbol{x}_l^{n\top}\boldsymbol{\mu}_2
$$
$$
\leq \boldsymbol{\mu}_2^\top \boldsymbol{V}_{(j,\cdot)} \sum_{s=1}^P \boldsymbol{x}_s^n \mathrm{softmax}_l(\boldsymbol{x}_s^{n\top}\boldsymbol{W}\boldsymbol{x}_l^n) \cdot (\boldsymbol{x}_s^n - \sum_{r=1}^P \mathrm{softmax}_l(\boldsymbol{x}_r^{n\top}\boldsymbol{W}\boldsymbol{x}_l^n)\boldsymbol{x}_r^n)\boldsymbol{x}_l^{n\top}\boldsymbol{\mu}_2
$$
$$
\cdot \phi_n(t)\frac{|\mathcal{S}_1^n|}{\sqrt{B}p_n(t)}.
$$
(112)

$$
\boldsymbol{\mu}_1^\top \boldsymbol{V}_{(i,\cdot)} \sum_{s=1}^P \boldsymbol{x}_s^n \mathrm{softmax}_l(\boldsymbol{x}_s^{n\top}\boldsymbol{W}\boldsymbol{x}_l^n) \cdot (\boldsymbol{x}_s^n - \sum_{r=1}^P \mathrm{softmax}_l(\boldsymbol{x}_r^{n\top}\boldsymbol{W}\boldsymbol{x}_l^n)\boldsymbol{x}_r^n)\boldsymbol{x}_l^{n\top}\boldsymbol{\mu}_2
$$
$$
= - \boldsymbol{\mu}_2^\top \boldsymbol{V}_{(i,\cdot)} \sum_{s=1}^P \boldsymbol{x}_s^n \mathrm{softmax}_l(\boldsymbol{x}_s^{n\top}\boldsymbol{W}\boldsymbol{x}_l^n) \cdot (\boldsymbol{x}_s^n - \sum_{r=1}^P \mathrm{softmax}_l(\boldsymbol{x}_r^{n\top}\boldsymbol{W}\boldsymbol{x}_l^n)\boldsymbol{x}_r^n)\boldsymbol{x}_l^{n\top}\boldsymbol{\mu}_2.
$$
(113)

$$
\boldsymbol{v}_k^\top \boldsymbol{V}_{(i,\cdot)} \sum_{s=1}^P \boldsymbol{x}_s^n \mathrm{softmax}_l(\boldsymbol{x}_s^{n\top}\boldsymbol{W}\boldsymbol{x}_l^n) \cdot (\boldsymbol{x}_s^n - \sum_{r=1}^P \mathrm{softmax}_l(\boldsymbol{x}_r^{n\top}\boldsymbol{W}\boldsymbol{x}_l^n)\boldsymbol{x}_r^n)\boldsymbol{x}_l^{n\top}\boldsymbol{\mu}_2
$$
$$
\leq \boldsymbol{\mu}_2^\top \boldsymbol{V}_{(j,\cdot)} \sum_{s=1}^P \boldsymbol{x}_s^n \mathrm{softmax}_l(\boldsymbol{x}_s^{n\top}\boldsymbol{W}\boldsymbol{x}_l^n) \cdot (\boldsymbol{x}_s^n - \sum_{r=1}^P \mathrm{softmax}_l(\boldsymbol{x}_r^{n\top}\boldsymbol{W}\boldsymbol{x}_l^n)\boldsymbol{x}_r^n)\boldsymbol{x}_l^{n\top}\boldsymbol{\mu}_2
$$
$$
\cdot \phi_n(t)\frac{|\mathcal{S}_1^n|}{\sqrt{B}p_n(t)} \cdot \frac{|\mathcal{R}_k^n|}{P - |\mathcal{S}_1^n|}.
$$
(114)

If $l \in \mathcal{R}_k^n$, $k \in [M]$, we have that for $j \in \mathcal{W}_{n,l}$, if $\boldsymbol{V}_{(j,\cdot)}\sum_{s=1}^P \boldsymbol{x}_s^n \mathrm{softmax}_l(\boldsymbol{x}_s^{n\top}\boldsymbol{W}\boldsymbol{x}_l^n) > 0$, $l' \in \mathcal{S}_1^n$,

$$
0 \leq \boldsymbol{\mu}_1^\top \boldsymbol{V}_{(j,\cdot)} \sum_{s=1}^P \boldsymbol{x}_s^n \mathrm{softmax}_l(\boldsymbol{x}_s^{n\top}\boldsymbol{W}\boldsymbol{x}_l^n) \cdot (\boldsymbol{x}_s^n - \sum_{r=1}^P \mathrm{softmax}_l(\boldsymbol{x}_r^{n\top}\boldsymbol{W}\boldsymbol{x}_l^n)\boldsymbol{x}_r^n)\boldsymbol{x}_l^{n\top}\boldsymbol{v}_k
$$
$$
\leq \boldsymbol{\mu}_1^\top \boldsymbol{V}_{(j,\cdot)} \sum_{s=1}^P \boldsymbol{x}_s^n \mathrm{softmax}_{l'}(\boldsymbol{x}_s^{n\top}\boldsymbol{W}\boldsymbol{x}_{l'}^n) \cdot (\boldsymbol{x}_s^n - \sum_{r=1}^P \mathrm{softmax}_{l'}(\boldsymbol{x}_r^{n\top}\boldsymbol{W}\boldsymbol{x}_{l'}^n)\boldsymbol{x}_r^n)\boldsymbol{x}_{l'}^{n\top}\boldsymbol{\mu}_1,
$$
(115)

$$\boldsymbol{\mu}_2^\top \boldsymbol{V}_{(j,\cdot)} \sum_{s=1}^{P} \boldsymbol{x}_s^n \text{softmax}_l(\boldsymbol{x}_s^{n\top} \boldsymbol{W} \boldsymbol{x}_l^n) \cdot (\boldsymbol{x}_s^n - \sum_{r=1}^{P} \text{softmax}_l(\boldsymbol{x}_r^{n\top} \boldsymbol{W} \boldsymbol{x}_l^n) \boldsymbol{x}_r^n) \boldsymbol{x}_l^{n\top} \boldsymbol{v}_k$$

$$= -\boldsymbol{\mu}_1^\top \boldsymbol{V}_{(j,\cdot)} \sum_{s=1}^{P} \boldsymbol{x}_s^n \text{softmax}_l(\boldsymbol{x}_s^{n\top} \boldsymbol{W} \boldsymbol{x}_l^n) \cdot (\boldsymbol{x}_s^n - \sum_{r=1}^{P} \text{softmax}_l(\boldsymbol{x}_r^{n\top} \boldsymbol{W} \boldsymbol{x}_l^n) \boldsymbol{x}_r^n) \boldsymbol{x}_l^{n\top} \boldsymbol{v}_k, \tag{116}$$

$$\boldsymbol{v}_k^\top \boldsymbol{V}_{(j,\cdot)} \sum_{s=1}^{P} \boldsymbol{x}_s^n \text{softmax}_l(\boldsymbol{x}_s^{n\top} \boldsymbol{W} \boldsymbol{x}_l^n) \cdot (\boldsymbol{x}_s^n - \sum_{r=1}^{P} \text{softmax}_l(\boldsymbol{x}_r^{n\top} \boldsymbol{W} \boldsymbol{x}_l^n) \boldsymbol{x}_r^n) \boldsymbol{x}_l^{n\top} \boldsymbol{v}_k$$

$$\leq \boldsymbol{\mu}_1^\top \boldsymbol{V}_{(j,\cdot)} \sum_{s=1}^{P} \boldsymbol{x}_s^n \text{softmax}_{l'}(\boldsymbol{x}_s^{n\top} \boldsymbol{W} \boldsymbol{x}_{l'}^n) \cdot (\boldsymbol{x}_s^n - \sum_{r=1}^{P} \text{softmax}_{l'}(\boldsymbol{x}_r^{n\top} \boldsymbol{W} \boldsymbol{x}_{l'}^n) \boldsymbol{x}_r^n) \boldsymbol{x}_{l'}^{n\top} \boldsymbol{\mu}_1 \tag{117}$$

$$\cdot \frac{|\mathcal{R}_k^n|}{P - |\mathcal{S}_1^n|}.$$

Likewise, if $l \in \mathcal{R}_k^n$, $k \in [M]$, $\boldsymbol{V}_{(j,\cdot)} \sum_{s=1}^{P} \boldsymbol{x}_s^n \text{softmax}_l(\boldsymbol{x}_s^{n\top} \boldsymbol{W} \boldsymbol{x}_l^n) > 0$, $j \in \mathcal{U}_{n,l}$, $l' \in \mathcal{S}_1^n$, $l'' \in \mathcal{S}_2^n$,

$$0 \leq \boldsymbol{\mu}_2^\top \boldsymbol{V}_{(j,\cdot)} \sum_{s=1}^{P} \boldsymbol{x}_s^n \text{softmax}_l(\boldsymbol{x}_s^{n\top} \boldsymbol{W} \boldsymbol{x}_l^n) \cdot (\boldsymbol{x}_s^n - \sum_{r=1}^{P} \text{softmax}_l(\boldsymbol{x}_r^{n\top} \boldsymbol{W} \boldsymbol{x}_l^n) \boldsymbol{x}_r^n) \boldsymbol{x}_l^{n\top} \boldsymbol{v}_k$$

$$\leq \boldsymbol{\mu}_2^\top \boldsymbol{V}_{(j,\cdot)} \sum_{s=1}^{P} \boldsymbol{x}_s^n \text{softmax}_{l''}(\boldsymbol{x}_s^{n\top} \boldsymbol{W} \boldsymbol{x}_{l''}^n) \cdot (\boldsymbol{x}_s^n - \sum_{r=1}^{P} \text{softmax}_{l''}(\boldsymbol{x}_r^{n\top} \boldsymbol{W} \boldsymbol{x}_{l''}^n) \boldsymbol{x}_r^n) \boldsymbol{x}_{l''}^{n\top} \boldsymbol{\mu}_2, \tag{118}$$

$$\boldsymbol{\mu}_1^\top \boldsymbol{V}_{(j,\cdot)} \sum_{s=1}^{P} \boldsymbol{x}_s^n \text{softmax}_l(\boldsymbol{x}_s^{n\top} \boldsymbol{W} \boldsymbol{x}_l^n) \cdot (\boldsymbol{x}_s^n - \sum_{r=1}^{P} \text{softmax}_l(\boldsymbol{x}_r^{n\top} \boldsymbol{W} \boldsymbol{x}_l^n) \boldsymbol{x}_r^n) \boldsymbol{x}_l^{n\top} \boldsymbol{v}_k$$

$$= -\boldsymbol{\mu}_2^\top \boldsymbol{V}_{(j,\cdot)} \sum_{s=1}^{P} \boldsymbol{x}_s^n \text{softmax}_{l''}(\boldsymbol{x}_s^{n\top} \boldsymbol{W} \boldsymbol{x}_{l''}^n) \cdot (\boldsymbol{x}_s^n - \sum_{r=1}^{P} \text{softmax}_{l''}(\boldsymbol{x}_r^{n\top} \boldsymbol{W} \boldsymbol{x}_{l''}^n) \boldsymbol{x}_r^n) \boldsymbol{x}_{l''}^{n\top} \boldsymbol{\mu}_2, \tag{119}$$

$$\boldsymbol{v}_k^\top \boldsymbol{V}_{(j,\cdot)} \sum_{s=1}^{P} \boldsymbol{x}_s^n \text{softmax}_l(\boldsymbol{x}_s^{n\top} \boldsymbol{W} \boldsymbol{x}_l^n) \cdot (\boldsymbol{x}_s^n - \sum_{r=1}^{P} \text{softmax}_l(\boldsymbol{x}_r^{n\top} \boldsymbol{W} \boldsymbol{x}_l^n) \boldsymbol{x}_r^n) \boldsymbol{x}_l^{n\top} \boldsymbol{v}_k$$

$$\leq \boldsymbol{\mu}_1^\top \boldsymbol{V}_{(j,\cdot)} \sum_{s=1}^{P} \boldsymbol{x}_s^n \text{softmax}_{l'}(\boldsymbol{x}_s^{n\top} \boldsymbol{W} \boldsymbol{x}_{l'}^n) \cdot (\boldsymbol{x}_s^n - \sum_{r=1}^{P} \text{softmax}_{l'}(\boldsymbol{x}_r^{n\top} \boldsymbol{W} \boldsymbol{x}_{l'}^n) \boldsymbol{x}_r^n) \boldsymbol{x}_{l'}^{n\top} \boldsymbol{\mu}_1 \tag{120}$$

$$\cdot \frac{|\mathcal{R}_k^n|}{P - |\mathcal{S}_1^n|}.$$

Therefore, by the update rule, we know

$$\boldsymbol{W}^{(t+1)} \boldsymbol{\mu}_1 = \boldsymbol{W}^{(t)} \boldsymbol{\mu}_1 - \eta \frac{1}{B} \sum_{n \in \mathcal{B}_b} \frac{\partial \ell(\boldsymbol{X}^n, y^n; \Psi)}{\partial \boldsymbol{W}^{(t)}} \boldsymbol{\mu}_1$$

$$= \boldsymbol{W}^{(t)} \boldsymbol{\mu}_1 + K(t) \boldsymbol{\mu}_1 + \sum_{l=2}^{M} \iota_l' \boldsymbol{\mu}_l, \tag{121}$$

where

$$K(t) \gtrsim \eta \frac{1}{B} \sum_{n \in \mathcal{B}_b} \frac{m |\mathcal{S}_1^n|}{aP} \zeta_{1,t} p_n(t) \phi_n(t) (P - |\mathcal{S}_1^n|), \tag{122}$$

$$\iota_l' \leq K(t) \cdot \max_n \left\{ \frac{|\mathcal{S}_1^n| \phi_n(t)}{p_n(t)} \right\} \leq K(t) \cdot e^{-q_1(t)}. \tag{123}$$

We know that

$$\boldsymbol{W}^{(0)} \boldsymbol{\mu}_1 \approx 0. \tag{124}$$

Then,

$$
\begin{aligned}
q_1(t+1) &= \boldsymbol{\mu}_1^\top \boldsymbol{W}^{(t+1)} \boldsymbol{\mu}_1 \\
&= \boldsymbol{\mu}_1^\top \boldsymbol{W}^{(t)} \boldsymbol{\mu}_1 + K(t) \\
&= q_1(t) + K(t) \\
&= \sum_{b=0}^{t} K(b).
\end{aligned}
\tag{125}
$$

Similarly,

$$
\begin{aligned}
\boldsymbol{W}^{(t+1)} \boldsymbol{\mu}_2 &= \boldsymbol{W}^{(t)} \boldsymbol{\mu}_2 - \eta \frac{1}{B} \sum_{n \in \mathcal{B}_b} \frac{\partial \ell(\boldsymbol{X}^n, y^n; \Psi)}{\partial \boldsymbol{W}^{(t)}} \boldsymbol{\mu}_2 \\
&= \boldsymbol{W}^{(t)} \boldsymbol{\mu}_2 + K(t) \boldsymbol{\mu}_2 + \sum_{l \neq 2} \iota_l' \boldsymbol{\mu}_l.
\end{aligned}
\tag{126}
$$

$$
\boldsymbol{\mu}_2^\top \boldsymbol{W}^{(t+1)} \boldsymbol{\mu}_2 = \sum_{b=0}^{t} K(b).
\tag{127}
$$

For $k \in [M]$,

$$
\boldsymbol{W}^{(t+1)} \boldsymbol{v}_k = \boldsymbol{W}^{(t)} \boldsymbol{v}_k + J_1(t) \boldsymbol{\mu}_1 + J_2(t) \boldsymbol{\mu}_2 + \sum_{l=1}^{M} \iota_l' \boldsymbol{v}_l.
\tag{128}
$$

By Hoeffding's inequality (15), with high probability,

$$
\|\boldsymbol{\mu}_1^\top \boldsymbol{W}^{(t+1)} \boldsymbol{v}_k\| \leq \Theta(1) \cdot \sqrt{\frac{\log B}{B}} \sum_{b=0}^{t} K(b) \lesssim \epsilon \cdot \sum_{b=0}^{t} K(b),
\tag{129}
$$

where the second step holds if $B \geq \epsilon^{-2} \log M$. And for $j \neq k, j \in [M]$,

$$
\|\boldsymbol{v}_j^\top \boldsymbol{W}^{(t)} \boldsymbol{v}_k\| \leq K(t) e^{-q_1(t)}.
\tag{130}
$$

For any $\boldsymbol{\mu}'$ such that $\boldsymbol{\mu}_1^\top \boldsymbol{\mu}' = \alpha$ and $\boldsymbol{\mu}' \perp \{\boldsymbol{v}_1, \boldsymbol{v}_2, \cdots, \boldsymbol{v}_M\}$, we can write $\boldsymbol{\mu}'$ as $\alpha \boldsymbol{\mu}_1 \pm \sqrt{1 - \alpha^2} \boldsymbol{\mu}_\perp$ for some $\boldsymbol{\mu}_\perp \perp \{\boldsymbol{\mu}_1, \boldsymbol{v}_1, \boldsymbol{v}_2, \cdots, \boldsymbol{v}_M\}$. Therefore,

$$
\begin{aligned}
{\boldsymbol{\mu}'}^\top \boldsymbol{W}^{(t+1)} \boldsymbol{\mu}' &= (\alpha \boldsymbol{\mu}_1 \pm \sqrt{1 - \alpha^2} \boldsymbol{\mu}_\perp)^\top \boldsymbol{W}^{(t+1)} (\alpha \boldsymbol{\mu}_1 \pm \sqrt{1 - \alpha^2} \boldsymbol{\mu}_\perp) \\
&= \alpha^2 {\boldsymbol{\mu}_1}^\top \boldsymbol{W}^{(t+1)} \boldsymbol{\mu}_1 \pm \Theta(\epsilon) \cdot {\boldsymbol{\mu}_1}^\top \boldsymbol{W}^{(t+1)} \boldsymbol{\mu}_1.
\end{aligned}
\tag{131}
$$

### E.2 PROOF OF LEMMA 4

For ease of presentation, we sometimes use $\boldsymbol{\mu}_2$ to represent $-\boldsymbol{\mu}_1$ in the proof.

$$
\begin{aligned}
&\eta \frac{1}{B} \sum_{n \in \mathcal{B}_b} \frac{\partial \ell(\boldsymbol{X}^n, y^n; \Psi)}{\partial \boldsymbol{V}_{(i, \cdot)}} \\
={}&\eta \frac{1}{B} \sum_{n \in \mathcal{B}_b} \frac{\partial \ell(\boldsymbol{X}^n, y^n; \Psi)}{\partial f(\boldsymbol{X}^n; \Psi)} \frac{f(\boldsymbol{X}^n; \Psi)}{\partial \boldsymbol{V}_{(i, \cdot)}} \\
={}&\eta \frac{1}{B} \sum_{n \in \mathcal{B}_b} (-y^n) \frac{1}{P} \sum_{l=1}^{P} a_{(l)_i} \mathbb{1}[\boldsymbol{V}_{(i, \cdot)} \boldsymbol{X} \operatorname{softmax}_l({\boldsymbol{X}^n}^\top \boldsymbol{W} \boldsymbol{x}_l^n) \geq 0] \\
&\cdot \Big( \sum_{s=1}^{P} \boldsymbol{x}_s^n \operatorname{softmax}_l({\boldsymbol{x}_s^n}^\top \boldsymbol{W} \boldsymbol{x}_l^n) \Big).
\end{aligned}
\tag{132}
$$

For $n$ such that $y^n = +1$ and $i \in \mathcal{W}_{n,l}$, we have that

$$
\mathbb{1}[\boldsymbol{V}_{(i, \cdot)} \boldsymbol{X} \operatorname{softmax}_l({\boldsymbol{x}_s^n}^\top \boldsymbol{W} \boldsymbol{x}_l^n) \geq 0] = 1,
\tag{133}
$$

and for $l \in \mathcal{S}_1^n$,

$$\sum_{s=1}^{P} \boldsymbol{x}_s^n \text{softmax}_l(\boldsymbol{x}_s^{n\top} \boldsymbol{W} \boldsymbol{x}_l^n) = p_n(t)\boldsymbol{\mu}_1 + \sum_{l=1}^{M_2} \iota_l' \boldsymbol{v}_l + \iota_{M_2+1}' \boldsymbol{\mu}_2, \tag{134}$$

where

$$\iota_l' \leq (1 - p_n(t)) \cdot \frac{|\mathcal{R}_k^l|}{P - |\mathcal{S}_1^n|}. \tag{135}$$

If $l \in \mathcal{S}_2^n$, we have

$$\sum_{s=1}^{P} \boldsymbol{x}_s^n \text{softmax}_l(\boldsymbol{x}_s^{n\top} \boldsymbol{W} \boldsymbol{x}_l^n) = p_n'(t)\boldsymbol{\mu}_2 + \sum_{l=1}^{M_2} \kappa_l' \boldsymbol{v}_l + \kappa_{M_2+1}' \boldsymbol{\mu}_2, \tag{136}$$

where

$$p_n'(t) \leq p_n(t), \tag{137}$$

$$\kappa_l' \leq (1 - p_n(t)) \cdot \frac{|\mathcal{R}_k^l|}{P - |\mathcal{S}_2^n|}. \tag{138}$$

If $l \in \mathcal{R}_k^n$, $k \in [M]$, we have

$$\sum_{s=1}^{P} \boldsymbol{x}_s^n \text{softmax}_l(\boldsymbol{x}_s^{n\top} \boldsymbol{W} \boldsymbol{x}_l^n) = p_n'(t)\boldsymbol{\mu}_1 + p_n''(t)\boldsymbol{\mu}_2 + o_n(t)\boldsymbol{v}_k + \sum_{l \neq k} u_l' \boldsymbol{v}_l, \tag{139}$$

where

$$p_n'(t) \leq \frac{|\mathcal{S}_1^n|}{P} \cdot p_n(t), \tag{140}$$

$$p_n''(t) \leq \frac{|\mathcal{S}_2^n|}{P} \cdot p_n(t), \tag{141}$$

$$o_n(t) \leq \frac{|\mathcal{R}_k^n|}{P} \cdot p_n(t) \tag{142}$$

$$u_l' \leq (1 - \frac{|\mathcal{S}_1^n| + |\mathcal{S}_2^n| + |\mathcal{R}_k^n|}{|\mathcal{S}_1^n|} \cdot p_n(t)) \cdot \frac{|\mathcal{R}_k^l|}{P - |\mathcal{S}_1^n| - |\mathcal{S}_2^n| - |\mathcal{R}_k^n|}. \tag{143}$$

Therefore, we have

$$-\eta \frac{1}{B} \sum_{n \in \mathcal{B}_b} \frac{\partial \ell(\boldsymbol{X}^n, y^n; \Psi)}{\partial \boldsymbol{V}} = \sum_{l=1}^{M} u_l' \boldsymbol{v}_l + q_n(t)\boldsymbol{\mu}_1 + q_n'(t)\boldsymbol{\mu}_2, \tag{144}$$

where

$$q_n(t)' \gtrsim \eta \frac{1}{B} \sum_{n \in \mathcal{B}_b} \frac{|\mathcal{S}_1^n|}{aP} \cdot p_n(t), \tag{145}$$

$$|q_n'(t)| \lesssim \eta \frac{1}{B} \sum_{n \in \mathcal{B}_b} \frac{|\mathcal{S}_2^n|}{aP} \cdot p_n(t), \tag{146}$$

$$|u_k'| \lesssim \eta \frac{1}{B} \sum_{n \in \mathcal{B}_b} \frac{|\mathcal{R}_k^n|}{aP} \cdot (1 - p_n(t)) \frac{1}{M}. \tag{147}$$

Then,

$$\boldsymbol{V}_{(i,\cdot)}^{(t)} \boldsymbol{\mu}_1 \geq \eta \sum_{b=0}^{t-1} \frac{1}{B} \sum_{n \in \mathcal{B}_b} \frac{|\mathcal{S}_1^n|}{aP} \cdot p_n(b), \tag{148}$$

$$\boldsymbol{V}_{(i,\cdot)}^{(t)} \boldsymbol{\mu}_2 = -\boldsymbol{V}_{(i,\cdot)}^{(t)} \boldsymbol{\mu}_1, \tag{149}$$

$$\boldsymbol{V}_{(i,\cdot)}^{(t)} \boldsymbol{v}_k \leq \eta \sum_{b=0}^{t-1} \frac{1}{B} \sum_{n \in \mathcal{B}_b} \frac{|\mathcal{S}_1^n|}{aPM}, \tag{150}$$

for $k \in [M]$. For $i \in \mathcal{U}_{n,l}$, we similarly have

$$\boldsymbol{V}_{(i,\cdot)}^{(t)} \boldsymbol{\mu}_2 \geq \eta \sum_{b=0}^{t-1} \frac{1}{B} \sum_{n \in \mathcal{B}_b} \frac{|\mathcal{S}_2^n|}{aP} \cdot p_n(b), \tag{151}$$

$$\boldsymbol{V}_{(i,\cdot)}^{(t)} \boldsymbol{\mu}_1 = -\boldsymbol{V}_{(i,\cdot)}^{(t)} \boldsymbol{\mu}_2, \tag{152}$$

$$\boldsymbol{V}_{(i,\cdot)}^{(t)} \boldsymbol{v}_k \leq \eta \sum_{b=0}^{t-1} \frac{1}{B} \sum_{n \in \mathcal{B}_b} \frac{|\mathcal{S}_1^n|}{aPM}, \tag{153}$$

for some $k \in [M]$. For $i \notin \mathcal{W}_{n,l} \cup \mathcal{U}_{n,l}$, we have that

$$\boldsymbol{V}_{(i,\cdot)}^{(t)} \boldsymbol{v}_k \leq \sqrt{\frac{\log B}{B}} \boldsymbol{V}_{(j,\cdot)}^{(t)} \boldsymbol{v}_k, \tag{154}$$

$$\boldsymbol{V}_{(i,\cdot)}^{(t)} \boldsymbol{\mu}_1 \leq \sqrt{\frac{\log B}{B}} \boldsymbol{V}_{(j,\cdot)}^{(t)} \boldsymbol{\mu}_1, \tag{155}$$

where $k \in [M]$, $j \in \mathcal{W}_{n,l} \cup \mathcal{U}_{n,l}$.

### E.3 PROOF OF LEMMA 1

We know that by Lemma 3 and 4 in (Li et al., 2023a), for $i \in \mathcal{W}_{n,l}(0)$ and $l \in \mathcal{S}_1^n$, we have that

$$\mathbb{1}[\boldsymbol{V}_{(i,\cdot)}^{(t)} \boldsymbol{R}_l^n(t)] = 1, \tag{156}$$

and for $i \in \mathcal{U}_{n,l}(0)$ and $l \in \mathcal{S}_2^n$, we have that

$$\mathbb{1}[\boldsymbol{V}_{(i,\cdot)}^{(t)} \boldsymbol{R}_l^n(t)] = 1. \tag{157}$$

We also have that the size of $\mathcal{W}_{n,l}$ and $\mathcal{V}_{n,l}$ are larger than $\Omega(m)$. Therefore, for $y^n = +1$, by Lemma 4 and 3, we have

$$
\begin{aligned}
f(\boldsymbol{X}^n; \Psi) =& \frac{1}{P} \sum_{l=1}^{P} \sum_{i \in \mathcal{W}_{l,n}(0)} \frac{1}{a} \mathrm{Relu}(\boldsymbol{V}_{(i,\cdot)} \boldsymbol{X} \mathrm{softmax}_l(\boldsymbol{X}^{n\top} \boldsymbol{W} \boldsymbol{x}_l^n)) \\
&+ \frac{1}{P} \sum_{l=1}^{P} \sum_{i \notin \mathcal{W}_{l,n}(0), a_{(l)_i} > 0} \frac{1}{a} \mathrm{Relu}(\boldsymbol{V}_{(i,\cdot)} \boldsymbol{X} \mathrm{softmax}_l(\boldsymbol{X}^{n\top} \boldsymbol{W} \boldsymbol{x}_l^n)) \\
&- \frac{1}{P} \sum_{l=1}^{P} \sum_{i: a_{(l)_i} < 0} \frac{1}{a} \mathrm{Relu}(\boldsymbol{V}_{(i,\cdot)} \boldsymbol{X} \mathrm{softmax}_l(\boldsymbol{X}^{n\top} \boldsymbol{W} \boldsymbol{x}_l^n)).
\end{aligned} \tag{158}
$$

We know that

$$
\begin{aligned}
&\frac{1}{P} \sum_{l=1}^{P} \sum_{i \in \mathcal{W}_{l,n}(0)} \frac{1}{a} \mathrm{Relu}(\boldsymbol{V}_{(i,\cdot)}^{(T)} \boldsymbol{X} \mathrm{softmax}_l(\boldsymbol{X}^{n\top} \boldsymbol{W}^{(T)} \boldsymbol{x}_l^n)) \\
&\gtrsim \frac{|\mathcal{S}_1^n|}{P} \cdot \frac{m}{a} \cdot \zeta_T \cdot p_n(T) \\
&\gtrsim \frac{|\mathcal{S}_1^n|}{P} \cdot \frac{m}{a^2} \cdot \eta \sum_{b=0}^{T-1} \frac{1}{B} \sum_{h \in \mathcal{B}_b} \frac{|\mathcal{S}_1^h|}{P} p_h(b) \cdot p_n(T).
\end{aligned} \tag{159}
$$

We can derive that

$$
\begin{aligned}
q_1(T) =& \sum_{b=0}^{T-1} K(b) \\
\geq& \sum_{b=0}^{T-1} \eta \frac{1}{B} \sum_{n \in \mathcal{B}_b} \frac{m|\mathcal{S}_1^n|}{aP} p_n(b) \phi_n(b) (P - |\mathcal{S}_1^n|) \eta \sum_{c=0}^{b-1} \frac{1}{B} \sum_{h \in \mathcal{B}_c} \frac{|\mathcal{S}_1^h|}{aP} p_h(c) \\
\gtrsim& \delta_*^4 \eta \sum_{b=0}^{T-1} \frac{1}{e^{q_1(b)}}.
\end{aligned} \tag{160}
$$

Therefore, we have that when $q_1(T) \leq O(1)$ or $q_1(T) \geq \Theta(T^c)$ for $c = \Theta(1)$, (160) does not hold. When $q_1(T) = \Theta(\log T)$, we have that (160) holds. In this case,

$$p_n(T) \geq \frac{\delta_* T^C}{\delta_* T^C + 1 - \delta_*} \geq 1 - \frac{1 - \delta_*}{\delta_*} T^{-C}, \tag{161}$$

where $C > 1$. Meanwhile, for $l \in \mathcal{R}_k^n$, $k \in [M]$, and any $s \in [P]$,

$$\text{softmax}_l(\boldsymbol{x}_s^{n\top} \boldsymbol{W}^{(T)} \boldsymbol{x}_l^n) = \Theta(\frac{1}{P}). \tag{162}$$

We can then derive that as long as

$$T \gtrsim \eta^{-1} \delta_*^{-2}, \tag{163}$$

we have

$$\frac{|\mathcal{S}_1^n|}{P} \cdot \frac{m}{a^2} \cdot \eta \sum_{b=0}^{T-1} \frac{1}{B} \sum_{h \in \mathcal{B}_b} \frac{|\mathcal{S}_1^h|}{P} p_h(b) \cdot p_n(T) \geq 1. \tag{164}$$

Then,

$$f(\boldsymbol{X}^n; \Psi) \geq 1, \ell(\boldsymbol{X}^n, y^n; \Psi) = 0. \tag{165}$$

With (163), we can also derive that

$$\sum_{k=1}^M \|\boldsymbol{V}_{(i,\cdot)}^{(T)} \boldsymbol{v}_k\|^2 \lesssim \frac{1}{M} \|\boldsymbol{V}_{(i,\cdot)}^{(T)} \boldsymbol{\mu}_1\|^2, \tag{166}$$

which means that for $i \in \mathcal{W}_{n,l}$ with $l \in \mathcal{S}_1^n$, $\boldsymbol{V}_{(i,\cdot)}^{(T)}$ is mainly in the direction of $\boldsymbol{\mu}_1$. This verifies condition (B) of Lemma 1. Therefore, by Hoeffding's inequality (15), for any $\boldsymbol{W}' \in \Psi$,

$$\Pr\left(\left\|\frac{1}{|\mathcal{B}_b|} \sum_{n \in \mathcal{B}_b} \frac{\partial \ell(\Psi; \boldsymbol{P}^n, z^n)}{\partial \boldsymbol{W}'} - \mathbb{E}\left[\frac{\partial \ell(\Psi; \boldsymbol{P}^n, z^n)}{\partial \boldsymbol{W}'}\right]\right\| \geq \left\|\mathbb{E}\left[\frac{\partial \ell(\Psi; \boldsymbol{P}^n, z^n)}{\partial \boldsymbol{W}'}\right] \epsilon\right\|\right)$$
$$\leq e^{-B\epsilon^2} \leq M^{-C}, \tag{167}$$

as long as

$$B \gtrsim \epsilon^{-2} \log M. \tag{168}$$

Then,

$$\mathbb{E}_{(\boldsymbol{X},y) \sim \mathcal{D}_{\mathcal{T}}} \ell(\boldsymbol{X}, y; \Psi) \leq \epsilon. \tag{169}$$

## F EXTENSION TO MULTI-CLASSIFICATION

Define that a $2^c$-classification is achieved by $c$ times of binary classification with the orthonormal set $\{\boldsymbol{\mu}_{\mathcal{T}}^{(1)}, \cdots, \boldsymbol{\mu}_{\mathcal{T}}^{(c)}\}$ as the discriminative patterns for the task $\mathcal{T}$. We have $\boldsymbol{\mu}_{\mathcal{T}}^{(i)} \perp \boldsymbol{v}_m$, $m \in [M]$, $i \in [c]$. The label $\boldsymbol{y}$ is $c$-dimensional with each entry chosen from $\{+1, -1\}$. Specifically, each $(\boldsymbol{X} \in \mathbb{R}^{d \times P}, \boldsymbol{y} \in \mathbb{R}^c) \sim \mathcal{D}_{\mathcal{T}}$ is generated as follows:

- Randomly generate the $k$-th entry $y_k$, $k \in [c]$ of the label $\boldsymbol{y}$ from $\{+1, -1\}$ with an equal probability.
- Each token is randomly chosen from $\{\boldsymbol{\mu}_{\mathcal{T}}^{(i)}, -\boldsymbol{\mu}_{\mathcal{T}}^{(i)}\}_{i=1}^c \cup \{\boldsymbol{v}_1, \cdots, \boldsymbol{v}_M\}$. If $y_k = 1$ (or $-1$), the number of tokens corresponding to $\boldsymbol{\mu}_{\mathcal{T}_k}$ (or $-\boldsymbol{\mu}_{\mathcal{T}_k}$) is larger than that of $-\boldsymbol{\mu}_{\mathcal{T}_k}$ (or $\boldsymbol{\mu}_{\mathcal{T}_k}$). $\boldsymbol{\mu}_{\mathcal{T}}^{(i)}$ and $-\boldsymbol{\mu}_{\mathcal{T}}^{(i)}$ (or "$-\boldsymbol{\mu}_{\mathcal{T}}^{(i)}$ and $\boldsymbol{\mu}_{\mathcal{T}}^{(i)}$") are referred to label-relevant and confusion patterns for $y_k = 1$ (or $y_k = -1$), respectively. The average fractions of label-relevant and confusion tokens of $\boldsymbol{\mu}_{\mathcal{T}}^{(i)}$ are $\delta_*^{(i)}$ and $\delta_\#^{(i)}$, respectively.

We then need $c$ sets of our binary model (4) to generate the output for $2^c$-classification, i.e.,

$$f(\boldsymbol{X}; \Psi) = (f_1(\boldsymbol{X}; \Psi), f_2(\boldsymbol{X}; \Psi), \cdots, f_c(\boldsymbol{X}; \Psi))$$

$$f_i(\boldsymbol{X}; \Psi) = \frac{1}{P} \sum_{l=1}^P \boldsymbol{a}_{(l)_i}^\top \text{Relu}(\boldsymbol{W}_{O_i}^\top \sum_{s=1}^P \boldsymbol{W}_{V_i} \boldsymbol{x}_s \text{softmax}_l(\boldsymbol{x}_s^\top \boldsymbol{W}_{K_i}^\top \boldsymbol{W}_{Q_i} \boldsymbol{x}_l)), \tag{170}$$

with $\Psi = \{\{\boldsymbol{a}_{(l)_i}\}_{l=1}^P, \boldsymbol{W}_{O_i}, \boldsymbol{W}_{V_i}, \boldsymbol{W}_{K_i}, \boldsymbol{W}_{Q_i}\}_{i=1}^c$. The dimensions of $\boldsymbol{W}_{O_i}, \boldsymbol{W}_{V_i}, \boldsymbol{W}_{K_i}, \boldsymbol{W}_{Q_i}$, $i \in [c]$ follow Section 3.2.

The learning process is then $c$ independent and parallel binary classification problems for each entry of the $c$-dimensional output. After fine-tuning, the trained model of each output entry has a similar property to Lemma 1 for single binary classification. $\delta_*^{(i)}$, the fraction of label-relevant pattern $\mu_{\mathcal{T}}^{(i)}$, $i \in [c]$, may decrease by $c$ times in average from the binary classification scenario. Therefore, by condition (iii) of Theorem 1, the number of iterations and samples increases by $c^2$ times, which is a polynomial of log scale of the number of classes $2^c$. Then, for the disrminative patterns $\{\boldsymbol{\mu}_{\mathcal{T}_1}^{(i)}\}_{i=1}^c$ of task $\mathcal{T}_1$ and $\{\boldsymbol{\mu}_{\mathcal{T}_2}^{(i)}\}_{i=1}^c$ and $\mathcal{T}_2$ of task $\mathcal{T}_2$, if for any $\boldsymbol{\mu}_{\mathcal{T}_1}^{(i)}$, there exists a unique $\boldsymbol{\mu}_{\mathcal{T}_2}^{(i)}$ close to be orthogonal to $\boldsymbol{\mu}_{\mathcal{T}_1}^{(i)}$, then $\mathcal{T}_1$ and $\mathcal{T}_2$ are irrelevant. If for any $\boldsymbol{\mu}_{\mathcal{T}_1}^{(i)}$, there exists a unique $\boldsymbol{\mu}_{\mathcal{T}_2}^{(i)}$ with a small angle to (or almost opposite to) $\boldsymbol{\mu}_{\mathcal{T}_1}^{(i)}$, then $\mathcal{T}_1$ and $\mathcal{T}_2$ are aligned (or contradictory). We can then derive similar conclusions as our Theorems 1 and 2 by combining the results of all the output entries.

