# OpenReview forum: "When is Task Vector Provably Effective for Model Editing? A Generalization Analysis of Nonlinear Transformers"
_ICLR.cc/2025/Conference — ICLR 2025 Oral_

### Official Review · Reviewer_QzER · 2024-10-30

**Soundness:** 4
**Presentation:** 4
**Contribution:** 4
**Rating:** 8
**Confidence:** 3

**Summary:**

The paper produces the first theoretical result on the ability of task vectors to generalize to new tasks and ability to unlearn a task. The study is conducted under the lens of "aligned" vs "irrelevant" vs "contradictory" tasks. The authors studies the necessary fine-tuning hyperparameters and the task vector coefficients ($alpha$) that enable generalization and unlearning.

Additionally, the authors also verify the theory result with experiments on toy CMNIST dataset and on next token prediction task.

**Strengths:**

- Very exciting theoretical result that includes very practical aspects (e.g., hyperparameters of the fine-tuning, how to set alpha for each task vectors, etc)
- Novel characterization of generalization conditions on 1 layer transformer, building up on previous work that uses NTK assumptions
- Practical scenarios in terms of the relation between tasks (aligned vs irrelevant vs contradictory)
- Nicely written and relatively accessible to non-theory people. I particularly like the remark after each theorem that explains what the theory is and what does it imply
- Nice setup on CMNIST that reflects the aligned vs. irrelevant vs. contradictory condition, followed by a nice non-toy experiment.

**Weaknesses:**

N/A -- good paper overall :)

**Questions:**

Definition 2 -- what is the dimension of $mu_{\tau}$? is it the same as $v$?

---

> ### Author Response · Authors · 2024-11-24
> **Response to Reviewer QzER**
>
> We appreciate your valuable time for the evaluation.
>
> **Q1 (Questions)**: what is the dimension of $\mu\_{\mathcal{T}}$? is it the same as $v$?
>
> **A1**: The dimension of $\mu\_{\mathcal{T}}$ is $d$. Yes, it is the same dimension as $v$. Thank you for pointing this out. We have added $\mu\_{\mathcal{T}}\in\mathcal{R}^d$ at the beginning of Definition 2.

---

### Official Review · Reviewer_L24R · 2024-11-02

**Soundness:** 3
**Presentation:** 3
**Contribution:** 3
**Rating:** 8
**Confidence:** 2

**Summary:**

This paper provides a theoretical analysis of the effectiveness of task vector methods for model editing in transformers. The authors investigate the conditions under which task addition (for multi-task learning) and task negation (for unlearning) are effective, proving that task correlation plays a crucial role. They also establish conditions for successful out-of-domain generalization using task vectors. Experiments with both synthetic and real-world data validate the key concepts of the proposed theory.

**Strengths:**

- This paper is the first to theoretically examine the generalization of task vectors, filling a significant gap in current research.
- Writing is well-written and easy to follow.
- The theoretical contributions are well-supported by experiments, effectively bridging theory and empirical validation.
- The theoretical insights align well with intuitive expectations regarding the effects of task vectors across aligned, irrelevant, and contradictory tasks.

**Weaknesses:**

- Although insightful, the data model may be overly simplistic for capturing the complexities of real-world data. For example, even for simple yes/no questions, a negation word in a sentence may flip the relevance of certain words in the sentence, which cannot be captured by the proposed data model. I wonder if some theoretical aspects can be generalized independently of this data model.
- The analysis is restricted to a one-layer transformer with limited nonlinearity, despite claims in the title and introduction regarding the challenges of analyzing nonlinearity in task vectors.

**Questions:**

I found this paper highly engaging and believe it would attract even greater interest with an exploration of the theory across a broader set of tasks, particularly generative tasks. For example, could the proposed theoretical framework be extended to multiclass classification as an initial step? A discussion on how these insights might be applied to a wider range of tasks would substantially enhance the paper's appeal. I would be willing to increase my score if the authors could provide even preliminary ideas on these extensions.

---

> ### Author Response · Authors · 2024-11-24
> **Response to Reviewer L24R Part I**
>
> We thank you for your valuable time and efforts in the evaluation. We have added Appendix D to discuss the extension to the multi-classification case.
>
> **Q1 (Weakness 1)**: Although insightful, the data model may be overly simplistic for capturing the complexities of real-world data. A negation word in a sentence may flip the relevance of certain words in the sentence, which cannot be captured by the proposed data model.
>
> **A1**: This is a good question. **First, our data formulation follows the state-of-the-art theoretical study of model training on vision data [Deng et al., 2023; Bu et al., 2024; Jiang et al., 2024] published at top ML conferences ICML, ICLR, and Neurips**. For example, Section 2.2 of [Deng et al., 2023], Section 2.1 of [Bu et al., 2024], and Section 3 of [Jiang et al., 2024] consider opposite key patterns for opposite labels, respectively. The data formulation we use is consistent with the existing theoretical works. The simplified data formulation allows us to characterize the gradient updates in different directions for different patterns in the convergence and generalization analysis. Meanwhile, to the best of our knowledge, **the mathematical formulation of the state-of-the-art training and generalization analyses of Transformers learning language data [Li et al., 2023b; Chen et al., 2024] do not explicitly consider the impact of negative words either**. These works establish the data model as Markov Chain, N-Gram model, etc., which also consider input language tokens as orthonormal vectors like our work.
>
> **Second, our formulation considers word embeddings as the input instead of the raw data**. In the embedding space, a negative word may map the contexts to the negative direction of the same contexts without the negative word. Based on such embeddings, our analysis can be applied.
>
> Deng et al., 2023. Robust Learning with Progressive Data Expansion Against Spurious Correlation. At Neurips.
>
> Bu et al., 2024. Provably Neural Active Learning Succeeds via Prioritizing Perplexing Samples. At ICML.
>
> Jiang et al., 2024. Unveil Benign Overfitting for Transformer in Vision: Training Dynamics, Convergence, and Generalization. At Neurips.
>
> Chen et al., 2024. Unveiling Induction Heads: Provable Training Dynamics and Feature Learning in Transformers. Preprint.
>
> Li et al., 2023b. How do transformers learn topic structure: Towards a mechanistic understanding. At ICML.

---

> > ### Author Response · Authors · 2024-11-24
> > **Response to Reviewer L24R Part II**
> >
> > **Q2 (Weakness 2)**: The analysis is restricted to a one-layer transformer with limited nonlinearity, despite claims in the title and introduction regarding the challenges of analyzing nonlinearity in task vectors.
> >
> > **A2**: Thank you for the question. **First, the motivation of our analysis emphasizes the contribution of analyzing the nonlinear transforms to distinguish from the existing approach that analyzes task vectors in the tangent space which requires linearization of the transformer models [Ortiz-Jimenez et al., 2023; Wu et al., 2023]**. Note that [Ortiz-Jimenez et al., 2023] use the Neural Tangent Kernel (NTK) framework (Jacot et al., 2018) to study neural networks as linearized models under specific assumptions. Under this linearized setting, the use of linear arithmetic on task vectors for targeted model editing can be justified. In contrast, our work first characterizes the learning dynamics of Transformers out of the NTK region  Although our model is only one-layer Transformer, our analysis of the learning dynamics does not require linearization as the existing work, thus, we emphasize we investigate task arithmetic using nonlinear Transformers.
> >
> > **Second, regarding the simplification of the model, the state-of-the-art theoretical works [Li et al., 2023; Jiang et al., 2024; Makkuva et al., 2024] recently published at top ML conferences like ICLR and ICML on the theoretical generalization and/or learning dynamics of Transformers also focus on one-layer single-head Transformers only**. That is because the loss landscape for multi-layer/head Transformers is highly nonlinear and non-convex due to the interactions between multiple nonlinear functions. The extension to multi-layer multi-head Transformers requires a more complicated characterization of the gradient updates, while the simplified model architecture can benefit the theoretical analysis as mentioned by the reviewer.
> > For the theoretical generalization guarantees of Transformers beyond one-layer single-head, to the best of our knowledge, only [Tian et al., 2023; Chen et al., 2024a; Chen et al., 2024b] theoretically study learning with multi-head or multi-layer Transformers. To the best of our knowledge, no existing works theoretically study the generalization of merging multiple task vectors,  even with one-layer single-head Transformers. Therefore, we focus on the one-layer analysis to obtain some initial theoretical insights. The theoretical findings are justified with multi-layer multi-head real-world Transformer architectures ViT-Small/16 and Phi-1.5, respectively. The results are consistent with the findings in Theorems 1, 2, and 3 using one-layer single-head Transformers.
> >
> > Ortiz-Jimenez et al., 2023. Task arithmetic in the tangent space: Improved editing of pre-trained models. At Neurips.
> > Wu et al., 2023. On the Convergence of Encoder-only Shallow Transformers. At Neurips.
> >
> >
> > Li et al., 2023. A Theoretical Understanding of Shallow Vision Transformers: Learning, Generalization, and Sample Complexity. At ICLR.
> >
> > Jiang et al., 2024. Unveil Benign Overfitting for Transformer in Vision: Training Dynamics, Convergence, and Generalization. At Neurips.
> >
> > Tian et al., 2023. Scan and Snap: Understanding Training Dynamics and Token Composition in 1-layer Transformer. At Neurips.
> >
> > Makkuva et al., 2024. Local to Global: Learning Dynamics and Effect of Initialization for Transformers. At Neurips.
> >
> > Chen et al., 2024a. Training dynamics of multi-head softmax attention for in-context learning: Emergence, convergence, and optimality. Preprint.
> >
> > Chen et al., 2024b. Provably learning a multi-head attention layer. Preprint.

---

> > > ### Author Response · Authors · 2024-11-24
> > > **Response to Reviewer L24R Part III**
> > >
> > > **Q3 (Question 1)**: This paper would attract greater interest with an exploration of the theory across a broader set of tasks, particularly generative tasks. Could the proposed theoretical framework be extended to multiclass classification as an initial step?
> > >
> > > **A3**: This is a great question. We understand that the reviewer expects a more general setup of multi-classification. The reason why we only discuss binary classification in the main body of our paper is that all the existing theoretical works studying the optimization and generalization of Transformers [Jelassi et al., 2022; Li et al., 2023; Jiang et al., 2024] only consider binary classification.
> > >
> > > Our theoretical analysis for binary classification can be extended to multi-classification. We have added a related discussion in Section D in Appendix. Briefly speaking, one can define that a $2^c$-classification is achieved by $c$ times of binary classification with the orthonormal set $\\{\mu\_\mathcal{T}^{(1)}, \cdots, \mu\_\mathcal{T}^{(c)}\\}$ as the discriminative patterns for the task $\mathcal{T}$. The label is $c$-dimensional with each entry chosen from $\\{+1,-1\\}$. We then need $c$ sets of our binary model (equation 4) to generate the output for $2^c$-classification. The learning process is then $c$ independent binary classification problems for each entry of the $c$-dimensional output. After fine-tuning, the trained model of each output entry has a similar property to Lemma 1 for binary classification. Then, for the disrminative patterns $\\{\mu\_{\mathcal{T}\_1}^{(i)}\\}\_{i=1}^c$ of task $\mathcal{T}\_1$ and $\\{\mu_{\mathcal{T}\_2}^{(i)}\\}\_{i=1}^c$ of task $\mathcal{T}\_2$, if for any $\mu_{\mathcal{T}\_1}^{(i)}$, there exists a unique $\mu\_{\mathcal{T}\_2}^{(i)}$ close to be orthogonal to $\mu\_{\mathcal{T}\_1}^{(i)}$, then $\mathcal{T}\_1$ and $\mathcal{T}\_2$ are irrelevant. If for any $\mu_{\mathcal{T}\_1}^{(i)}$, there exists a unique $\mu_{\mathcal{T}\_2}^{(i)}$ with a small angle to (or almost opposite to) $\mu\_{\mathcal{T}\_1}^{(i)}$, then $\mathcal{T}\_1$ and $\mathcal{T}\_2$ are aligned (or contradictory). We can then derive similar conclusions as our Theorems 1 and 2 by combining the results of all the output entries.
> > >
> > > Jelassi et al., 2022. Vision transformers provably learn spatial structure. At Neurips.
> > >
> > > Li et al., 2023. A Theoretical Understanding of Shallow Vision Transformers: Learning, Generalization, and Sample Complexity. At ICLR.
> > >
> > > Jiang et al., 2024. Unveil Benign Overfitting for Transformer in Vision: Training Dynamics, Convergence, and Generalization. At Neurips.

---

> > > > ### Comment · Reviewer_L24R · 2024-11-24
> > > >
> > > > Thank you to the authors for your detailed response and for incorporating our feedback into the revised paper. Regarding the second point raised by A1, I am curious if there are any studies or papers that demonstrate ''In the embedding space, a negative word may map the contexts to the negative direction of the same contexts without the negative word'' empirically?

---

> ### Author Response · Authors · 2024-11-25
>
> **A**: Thank you for the question. By this statement, we mean that **the embeddings between a positive word and a corresponding negative word are distinct enough**. There are several empirical works [Engler et al., 2022; Liu et al., 2024] studying task vectors in the activation/embedding space or the interpretability of word embeddings that can verify this statement to some extent. Section 4.2 of [Engler et al., 2022] concludes that in the binary sentiment prediction task, **the word embeddings of antonymous words, such as** "**unpleasant**" and "**pleasant**", and "**tasteless**" and "**tasetful**", **are the most discriminative.** Note that the embeddings can also refer to the outputs of intermediate layers of a deep network. Table 5 of [Liu et al., 2024] illustrates that by adding the negation of a positive embedding to each layer of a deep Transformer, the model output will change from a positive generation to a negative one. This means that **two embeddings, where one is in the negative direction of the other, are decoded as antonymous generation** in the setting of [Liu et al., 2024], which justifies our statement.
>
> Engler et al., 2022. SensePOLAR: Word sense aware interpretability for pre-trained contextual word embeddings. At EMNLP findings.
>
> Liu et al., 2024. In-context Vectors: Making In Context Learning More Effective and Controllable Through Latent Space Steering. At ICML

---

> > ### Comment · Reviewer_L24R · 2024-11-25
> >
> > Thank you for finding the relevant papers. I would recommend including a discussion of them to justify Definition 2. I also have an additional question about A3: How do the extended bounds in Theorems 1 and 2 depend on the number of classes, $2^c$? Do they increase on a log scale? Specifically, are they proportional to $c$ or $2^c$ when the number of classes is $2^c$?

---

> ### Author Response · Authors · 2024-11-26
>
> Dear Reviewer L24R,
>
> We have added a footnote to introduce the empirical motivation of the studied data formulation for binary classification around Definition 2 on page 5 in the updated version. Thank you for the suggestion.
>
> It is a great question about how our derived bound in Theorems 1 and 2 changes when the number of classes becomes $2^c$ for $c>1$. We think that **to achieve the same generalization performance in Theorems 1 and 2, the required number of iterations and samples increase on a log or a polynomial of log scale of the number of classes**. Consider our extension to $2^c$ classification in Appendix E as an example. This $2^c$ classification is essentially solving $c$ parallel binary classification. However, $\delta\_*^{(i)}$, the fraction of label-relevant pattern $\mu\_{\mathcal{T}}^{(i)}$, $i\in[c]$, may decrease by $c$ times in average from the binary classification scenario. Therefore, by condition (iii) of Theorem 1, the number of iterations and samples increases by $c^2$ times, which is a polynomial of log scale of the number of classes $2^c$. Note that this derivation comes from our problem formulation of $2^c$ classification and our analytical tool of optimization and generalization of Transformers. Additional assumptions and advanced techniques might lead to a tighter bound.
>
> Authors

---

> > ### Comment · Reviewer_L24R · 2024-11-26
> >
> > Thank you for the detailed answer! The rebuttal effectively addressed my concerns and questions, and I enjoyed the discussion. I have raised my score accordingly.

---

> > > ### Author Response · Authors · 2024-11-26
> > > **Thank you for the discussion and raising the score!**
> > >
> > > Dear Reviewer L24R,
> > >
> > > We are delighted that our response effectively addresses your concerns. Thank you for the discussion and raising the score from 6 to 8.
> > >
> > > Authors

---

### Official Review · Reviewer_3aPS · 2024-11-02

**Soundness:** 3
**Presentation:** 2
**Contribution:** 3
**Rating:** 6
**Confidence:** 3

**Summary:**

This work focuses on the task vector in the context of task arithmetic, demonstrating its role in learning and unlearning tasks through experiments. It’s an interesting topic that needs more exploration. The authors find that task correlation affects the performance of the task vector and conduct theoretical justification on the low-rank approximation of the task vector and it's ability to adapt to out-of-domain tasks.

**Strengths:**

* Very comprehensive mathematical analysis and theoretical proofs
* Discussion on the task vector is extensive.

**Weaknesses:**

* There are some issues with the paper's writing (Formula 4 and Definition 2 in the Section 3.2 is confusing).
* In the language generation task, only a model with 1.5B parameters is used, and the experimental results are not fully meet expectations (also a noticeable performance loss in so-called irrelevant task).

**Questions:**

* Line 236-238, the conventional attention expression is $softmax(W_QXX^TW_K^T)W_VX$, why is it written as $W_VXsoftmax(X^TW_K^TW_QX)$ in Formula 4?
* Line 236-238, what is the meaning of $X^n$?
* Line 242, Why is $x_i$ used here, while $X$ is used in Formula 4?
* Line 261, Since $\mu_T$ and $v_j$ are orthogonal, what is the meaning of  tokens corresponding to $\mu_T$?
* How to quantify the relevance of different language generation tasks？Are semantically similar and task-related equivalent?

---

> ### Author Response · Authors · 2024-11-24
> **Response to Reviewer 3aPS Part I**
>
> We thank the reviewer for the valuable time and effort in the evaluation. We have made revisions in Equation 4, Equation 11, and Definition 2 and included Table 6 of key notations in Appendix B according to the review. We also improved our discussion of our language model experiment in lines 507-509 and added an illustration of the result of less-aligned task in lines 538-539.
>
> **Q1 (Weakness 1 & Question 1 & 2)**: There are some issues with the paper's writing (Formula 4 and Definition 2 in the Section 3.2 is confusing). Line 236-238, the conventional attention expression is $softmax(W_Q XX^\top W_K^\top)W_V X$, why is it written as $W_V X(X^\top W_K^\top W_Q X)$ in Formula 4? Line 236-238, what is the meaning of $X^n$?
>
> **A1**: Thank you for the question. We have added table 6 at the beginning of Appendix B to summarize important notations to improve the readability.
>
> Equation 4 should be revised as $$f( X; \Psi)=\frac{1}{P}\sum_{l=1}^P a_{(l)}^\top\text{Relu}(W_O W_V\sum_{s=1}^P x_s\text{softmax}_l({x_s}^\top W_K^\top W_Q x_l)).$$ In the original version, there is an $X^n$ by mistake, which should be $X$. We also change the vector version of attention in the original Equation 4 into a scalar version to facilitate understanding. We are sorry for the typo and the confusion.
>
> The formulation of our Transformer model is also used by [Li et al., 2023; Zhang et al., 2024; Huang et al., 2024]. We think the reviewer refers to the formulation of $softmax(X W_Q  W_K^\top X^\top) X W_V$ in Question 1 so that the softmax attention is computed based on the number of tokens. This formulation is essentially the same as our original correct version $ W_V X\text{softmax}_l({ X}^\top W_K^\top W_Q x_l))$ by transposing $X$, $W_Q$, $W_K$, and $W_V$.
> For Question 3, $X^n$ should be first introduced in line 245, which denotes the $n$-th input data $X\in\mathbb{R}^{d\times P}$.
>
> Li et al., 2023. A Theoretical Understanding of Shallow Vision Transformers: Learning, Generalization, and Sample Complexity. At ICLR.
>
> Zhang et al., 2024. Trained Transformers Learn Linear Models In-Context. At JMLR.
>
> Huang et al., 2024. In-context convergence of transformers. At ICML.

---

> ### Author Response · Authors · 2024-11-24
> **Response to Reviewer 3aPS Part II**
>
> **Q2 (Weakness 2)**: In the language generation task, only a model with 1.5B parameters is used, and the experimental results are not fully meet expectations (also a noticeable performance loss in so-called irrelevant task).
>
> **A2**: This is a good question. **First, we would like to clarify that task vector and even the state-of-the-art machine unlearning methods [Zhang et al., 2024; Jia et al., 2024; Maini et al., 2024] are empirically shown to result in a performance drop in the retain set, the set we aim to maintain the performance**. This is a trade-off between forgetting and retaining performance. For example, Table 3 of [Shi et al., 2024] illustrates that using commonly used machine unlearning techniques will lead to a loss in the utility. Especially, task vector method decreases $13.1\\%$ performance on the whole dataset BOOKS in average. Therefore, the performance loss on the less aligned task with “Pride and Prejudice” in our paper, which is $15.13\\%$ for full-rank task vector in Table 3, and $13.83\\%$ for low-rank task vector in Table 4, is not bad. This is an inherent and reasonable performance degradation of the task vector approach.
>
> **Second, we would like to emphasize that the key conclusion of the language generation task is the comparison between the forgetting performance of the aligned task $\mathcal{T}\_{HP2}$ and the less-aligned task $\mathcal{T}\_{PP}$**. Note that the performance decrease of $\mathcal{T}\_{PP}$ is much smaller than $\mathcal{T}\_{HP2}$, while the performance decrease of $\mathcal{T}\_{HP2}$ is close to that of $\mathcal{T}\_{HP1}$. This verifies Theorem 2 that unlearning a task $\mathcal{T}_{HP1}$ can significantly degrade the performance of the aligned task ($\mathcal{T}\_{HP2}$) while the accuracy of the less-aligned task ($\mathcal{T}\_{PP}$) decreases less. We have revised the sentence in lines 507-509 accordingly to help the understanding.
>
> We thank the suggestion of trying larger models. **We use Phi-3-small (7B) to repeat the experiment using LoRA**. The result is shown below in the Table. **We can see that the insight of Theorem 2 still holds, i.e., unlearning a certain task (HP1) can effectively forget the aligned task (HP2) with a performance loss of $52.29\\%$, but forget the less-aligned task less (PP) with a performance loss of $20.65\%$**. Moreover, by using a larger model, the unlearning performance of the aligned task HP2 is improved from a $37.23\\%$ decrease to a $55.61\\%$ decrease. In comparison, the performance difference on the less-aligned PP is much smaller, from a $15.13\\%$ decrease to a $20.65\\%$ decrease.
>
> | $\lambda$    | 0 (baseline)  | -0.2      | -0.4     | -0.6      | -0.8      | -1                              |
> | -------- | ------- | -------| -------| ------- | ------- | ------- |
> | $\mathcal{T}\_{HP1}$ | 0.2573         | 0.1989 | 0.1933 | 0.1888 | 0.1542 | 0.1142 ($55.61\\%$) |
> | $\mathcal{T}\_{HP2}$ | 0.2688         | 0.2113 | 0.1993 | 0.1938 | 0.1622 | 0.1563 ($52.29\\%$) |
> | $\mathcal{T}\_{PP}$   | 0.1942         | 0.1825 | 0.1644 | 0.1687 | 0.1592 | 0.1541 ($20.54\\%$) |
>
> Shi et al., 2024. MUSE: Machine Unlearning Six-Way Evaluation for Language Models.
>
> Zhang et al., 2024. From Catastrophic Collapse to Effective Unlearning. COLM 2024.
>
> Jia et al., 2024. WAGLE: Strategic Weight Attribution for Effective and Modular Unlearning in Large Language Models. Neurips 2024.
>
> Maini et al., 2024. Tofu: A task of fictitious unlearning for llms. Preprint.
>
> -----------
>
> **Q3 (Question 3)**: Line 242, Why is x_i used here, while X is used in Formula 4?
>
> **A3**: Line 242 computed the attention weight between $x_l$ and $x_i$, which is a scalar. The original formula 4 computes the vector version of the attention, i.e., the vector of attention weights between $x_l$ and every $x_j$ for $j\in[P]$. We have changed Formula 4 to the scalar version to be consistent with that in lines 242-243 and help the understanding. Thank you for pointing it out.
>
> -------------
>
> **Q4 (Question 4)**: Line 261, Since $\mu\_\mathcal{T}$ and $v\_j$ are orthogonal, what is the meaning of tokens corresponding to $\mu\_\mathcal{T}$?
>
> **A4**: We define that “each token is randomly chosen from $\\{\mu_{\mathcal{T}}, -\mu_{\mathcal{T}}\\}\cup\\{v_1,\cdots,v_M\\}$“. Then, “tokens corresponding to $\mu\_\mathcal{T}$” refers to “tokens equal to $\mu\_\mathcal{T}$”. We have made revisions accordingly to avoid the confusion.

---

> ### Author Response · Authors · 2024-11-24
> **Response to Reviewer 3aPS Part III**
>
> **Q5 (Question 5)**: How to quantify the relevance of different language generation tasks？Are semantically similar and task-related equivalent?
>
> **A5**: This is a good question. One possible practical method to approximate the relevance $\alpha$ between two different language generation tasks is as follows. This is illustrated around Equation 11 of the updated submission. Consider two models $\Psi^*\_{\mathcal{T}\_1}$ and $\Psi^*\_{\mathcal{T}\_2}$ finetuned on tasks $\mathcal{T}\_1$ with test set $\mathcal{D}\_1$ and $\mathcal{T}\_2$ with dataset $\mathcal{D}\_2$, respectively. Then, compute $\hat{\alpha}(\Psi\_{\mathcal{T}\_1}^*, \Psi\_{\mathcal{T}\_2}^*)=1/2(\hat{\alpha}(\Psi\_{\mathcal{T}\_1}^*, \Psi\_{\mathcal{T}\_2}^*, \mathcal{D}\_1)+\hat{\alpha}(\Psi\_{\mathcal{T}\_1}^*, \Psi\_{\mathcal{T}\_2}^*, \mathcal{D}\_2))$ where $\hat{\alpha}(\Psi\_{\mathcal{T}\_1}^*, \Psi\_{\mathcal{T}\_2}^*, \mathcal{D}\_j)=\frac{1}{|\mathcal{D}\_j|}\sum_{i\in \mathcal{D}\_j}\cos\left\langle \tilde{y}\_{1,j}^i, \tilde{y}\_{2,j}^i\right\rangle$, $j=1,2$, which represents the cosine similarity between the centered output of the input $\mathcal{D}\_j$ using the two finetuned models $\Psi\_{\mathcal{T}\_1}^*$ and $\Psi\_{\mathcal{T}\_2}^*$. Here $\tilde{y}\_{l,j}^i=\hat{y}\_{l,j}^i-\frac{1}{|\mathcal{D}\_j|}\sum\_{i\in \mathcal{D}\_j}\hat{y}\_{l,j}^i$ for $l,j\in\\{1,2\\}$ represents the $i$-th output of the fine-tuned model $\Psi_{\mathcal{T}_l}^*$ on the test set $\mathcal{D}_j$.
>  Note that to compute $\hat{\alpha}(\Psi\_{\mathcal{T}\_1}^*, \Psi\_{\mathcal{T}\_2^*})$, we do not require the availability of extra models or datasets except $\Psi\_{\mathcal{T}\_1}^*$, $\Psi\_{\mathcal{T}\_2}^*$, and the test set $\mathcal{D}\_1$ and $\mathcal{D}\_2$. By computation, $\hat{\alpha}(\Psi\_{\mathcal{T}\_{HP1}}^*, \Psi\_{\mathcal{T}\_{HP2}}^*)=0.498$ and $\hat{\alpha}(\Psi\_{\mathcal{T}\_{HP1}}^*, \Psi\_{\mathcal{T}\_{PP}}^*)=0.239$, which indicates that HP1 is more aligned with HP2 than PP. .
>
> Since semantic similarity does not have a formal mathematical formulation, we cannot say it is equivalent to task relation, which we describe in this paper through the mathematical quantity $\alpha$. However, we would like to illustrate that these two notions are highly correlated in language generation tasks. This is because learning the next token prediction task is essentially learning the semantic structure from the distribution of the language data. We say HP1 and HP2 are semantically similar since they are from the same author and the same series, which implies a similar writing style and sets of vocabulary.  Then, given two similar inputs, the outputs are more likely to be close to each other. Note that this is consistent with the similarity between the two tasks by our definition in Section 3.3. In our theoretical setup, related or aligned tasks mean that two similar label-relevant patterns that correspond to the same label. In our theoretical formulation, we define that $\mathcal{T}\_1$ and $\mathcal{T}\_2$ are aligned if their correlation $\alpha=\mu\_{\mathcal{T}\_1}^\top\mu\_{\mathcal{T}\_2}>0$, where $\mu\_{\mathcal{T}\_1}$ and $\mu\_{\mathcal{T}\_2}$ are the discriminative patterns of $\mathcal{T}\_1$ and $\mathcal{T}\_2$ defined in Definition 2, respectively. Therefore, semantic similarity is a close notion to task relevance/alignment in our language generation task.

---

> > ### Comment · Reviewer_3aPS · 2024-11-25
> >
> > I have read the author rebuttal and made any necessary changes to my review.

---

> > > ### Author Response · Authors · 2024-11-25
> > > **Thank you!**
> > >
> > > Dear Reviewer 3aPS,
> > >
> > > Thank you for raising the score. We are delighted that our rebuttal helps to address your concerns.
> > >
> > > Authors

---

### Official Review · Reviewer_3a5G · 2024-11-03

**Soundness:** 3
**Presentation:** 3
**Contribution:** 3
**Rating:** 8
**Confidence:** 2

**Summary:**

This paper explores the theoretical aspects of task vector arithmetic as a model editing technique for multi-task learning, unlearning, and out-of-domain generalization. The authors provide a theoretical analysis to justify why and when task vector methods are effective in nonlinear Transformer models, especially for binary classification tasks. They prove that task addition facilitates multi-task learning for aligned or irrelevant tasks, while task negation can effectively unlearn contradictory or irrelevant tasks. Additionally, they offer generalization guarantees for out-of-domain tasks and theoretical justification for task vector approximations. These findings are empirically validated through various experiments.

**Strengths:**

- The paper is very well-written and easy to follow.
- It provides a guideline for when and why task arithmetic works in multi-task learning, machine unlearning, and generalization to new tasks.
- The discussion of low-rank approximations and magnitude-based pruning of task vectors supports the use of efficient approximation techniques in task arithmetic fine-tuning.
- This is the first known theoretical generalization analysis of task vector arithmetic in nonlinear Transformer-based models, filling a notable gap in the literature.
- The theoretical claims are validated through empirical experiments on the Phi-1.5 language model and Colored-MNIST image classification, adding practical credibility to the proposed framework.

**Weaknesses:**

- The theoretical analysis relies on a single-head, one-layer Transformer model, which may limit the applicability of the results to more complex multi-layer Transformer architectures.
- While the empirical validation includes a large language model and a basic image classification task, the study could benefit from a broader set of tasks, including more complex or structured tasks beyond binary classification.
- Although the theoretical framework outlines conditions for selecting arithmetic coefficients, more practical guidelines or analyses for tuning these coefficients in real-world applications would be beneficial.

typos:
- line 288, "fine-turning" -> "fine-tuning"
- line 388, "are are" -> "are"

**Questions:**

I don't have questions.

---

> ### Author Response · Authors · 2024-11-24
> **Response to Reviewer 3a5G Part I**
>
> We thank the reviewer for the valuable time in the evaluation. We have included a more practical guideline for approximating correlations between tasks and tuning task vector hyperparameters by revising Equation 11 in the updated submission.
>
> **Q1 (Weakness 1)**: The theoretical analysis relies on a single-head, one-layer Transformer model, which may limit the applicability of the results to more complex multi-layer Transformer architectures.
>
> **A1**: This is a good question. **First, the state-of-the-art theoretical works [Li et al., 2023; Makkuva et al., 2024] recently published at top ML conferences like ICLR, Neurips, and ICML on the theoretical generalization and/or learning dynamics of Transformers also focus on one-layer single-head Transformers only**. That is because the loss landscape for multi-layer/head Transformers is highly nonlinear and non-convex due to the interactions between multiple nonlinear functions. The extension to multi-layer multi-head Transformers requires a more complicated characterization of the gradient updates, while the simplified model architecture can benefit the theoretical analysis. For the theoretical generalization guarantees of Transformers beyond one-layer single-head, to the best of our knowledge, only [Tian et al., 2023; Chen et al., 2024a; Chen et al., 2024b] theoretically study learning with multi-head or multi-layer Transformers. **To the best of our knowledge, no existing works theoretically study the generalization of merging multiple task vectors,  even with one-layer single-head Transformers.** Therefore, we focus on the one-layer analysis to obtain some initial theoretical insights.
>
>
> **Second, the simplification of one-layer single-head Transformers enables us to make contributions under our theoretical settings**. Our work is the first one to prove the effectiveness of task addition and negation with Transformer models using feature-learning analysis. Our work also proves guaranteed generalization of task arithmetic in out-of-domain generalization. The simplification also enables the theoretical proof of the validity of low-rank approximation and magnitude-based pruning for task vectors.
>
>
> **Third, the theoretical conclusions of one-layer single-head Transformers are empirically verified by multi-layer multi-head Transformers to some extent**. We justify our theoretical findings on multi-layer multi-head real-world Transformer architectures ViT-Small/16 and Phi-1.5, respectively. Experiments in Figures 1, 2, Tables 3, and 4 justify our Theorem 1, 2, and 3. This implies that our generalization results of task arithmetic on different tasks can potentially be extended to multi-layer multi-head cases. We leave the detailed theoretical analysis of the extension as future works.
>
> Li et al., 2023. A Theoretical Understanding of Shallow Vision Transformers: Learning, Generalization, and Sample Complexity. At ICLR.
>
> Jiang et al., 2024. Unveil Benign Overfitting for Transformer in Vision: Training Dynamics, Convergence, and Generalization. At Neurips.
>
> Tian et al., 2023. Scan and Snap: Understanding Training Dynamics and Token Composition in 1-layer Transformer. At Neurips.
>
> Makkuva et al., 2024. Local to Global: Learning Dynamics and Effect of Initialization for Transformers. At Neurips.
>
> Chen et al., 2024a. Training dynamics of multi-head softmax attention for in-context learning: Emergence, convergence, and optimality. Preprint.
>
> Chen et al., 2024b. Provably learning a multi-head attention layer. Preprint.

---

> ### Author Response · Authors · 2024-11-24
> **Response to Reviewer 3a5G Part II**
>
> **Q2 (Weakness 2)**: While the empirical validation includes a large language model and a basic image classification task, the study could benefit from a broader set of tasks, including more complex or structured tasks beyond binary classification.
>
> **A2**: Thank you very much for the insightful question. First, we would like to emphasize that the proposed LLM unlearning experiment is a generative task, which goes beyond binary classification.
>
> Second, we sincerely appreciate the encouragement to explore more complex or structured tasks. Since this work primarily focuses on theoretical contributions and insights, we prioritized experiments on the binary classification task using the Colored-MNIST dataset. This simpler setup was chosen because it effectively allows us to validate and illustrate the theoretical insights derived in our study. For example, we prove that task addition is effective for multi-task learning when the tasks are either irrelevant or aligned (Theorem 1), while task negation is provably successful for unlearning tasks that are either irrelevant or contradictory (Theorem 2). A linear combination of these task vectors can generalize to a wide range of new tasks by properly selecting the arithmetic coefficients (Theorem 3). To further substantiate our theoretical results, we extended our experiments to include machine unlearning tasks on LLMs. We believe that the selected tasks and datasets are representative, ensuring that our theory is both well-supported and practically applicable. Thank you again for your valuable feedback, which will guide future extensions of this work.
>
> -------------
>
> **Q3 (Weakness 3)**: Although the theoretical framework outlines conditions for selecting arithmetic coefficients, more practical guidelines or analyses for tuning these coefficients in real-world applications would be beneficial.
>
> **A3**: This is a great question. **Our Theorems 1 and 2 show that, given the correlation $\alpha$ between tasks $\mathcal{T}\_1$ and $\mathcal{T}\_2$, we can determine whether the merged model $\Psi=\Psi^{(0)}+\Delta\Psi_{\mathcal{T}\_1}+\lambda \Delta\Psi_{\mathcal{T}\_2}$ can take effect and what $\lambda$ to choose based on Theorems 1 and 2**.
>
> One possible practical method to approximate $\alpha$ is as follows. Consider two models $\Psi^*\_{\mathcal{T}\_1}$ and $\Psi^*\_{\mathcal{T}\_2}$ finetuned on tasks $\mathcal{T}\_1$ with test set $\mathcal{D}\_1$ and $\mathcal{T}\_2$ with dataset $\mathcal{D}\_2$, respectively. Then, compute $\hat{\alpha}(\Psi\_{\mathcal{T}\_1}^*, \Psi\_{\mathcal{T}\_2}^*)=1/2(\hat{\alpha}(\Psi\_{\mathcal{T}\_1}^*, \Psi\_{\mathcal{T}\_2}^*, \mathcal{D}\_1)+\hat{\alpha}(\Psi\_{\mathcal{T}\_1}^*, \Psi\_{\mathcal{T}\_2}^*, \mathcal{D}\_2))$ where $\hat{\alpha}(\Psi\_{\mathcal{T}\_1}^*, \Psi\_{\mathcal{T}\_2}^*, \mathcal{D}\_j)=\frac{1}{|\mathcal{D}\_j|}\sum_{i\in \mathcal{D}\_j}\cos\left\langle \tilde{y}\_{1,j}^i, \tilde{y}\_{2,j}^i\right\rangle$, $j=1,2$, which represents the cosine similarity between the centered output of the input $\mathcal{D}\_j$ using the two finetuned models $\Psi\_{\mathcal{T}\_1}^*$ and $\Psi\_{\mathcal{T}\_2}^*$. Here $\tilde{y}\_{l,j}^i=\hat{y}\_{l,j}^i-\frac{1}{|\mathcal{D}\_j|}\sum\_{i\in \mathcal{D}\_j}\hat{y}\_{l,j}^i$ for $l,j\in\\{1,2\\}$ represents the $i$-th output of the fine-tuned model $\Psi_{\mathcal{T}_l}^*$ on the test set $\mathcal{D}_j$.
>
> Note that to compute $\hat{\alpha}(\Psi\_{\mathcal{T}\_1}^*, \Psi\_{\mathcal{T}\_2^*})$, we do not require the availability of extra models or datasets except $\Psi\_{\mathcal{T}\_1}^*$, $\Psi\_{\mathcal{T}\_2}^*$, and the test set $\mathcal{D}\_1$ and $\mathcal{D}\_2$. We have revised Equation 11 accordingly. **We find that the values of $\bar{\alpha}$ for the aligned, irrelevant, and contradictory tasks studied in Figure 1 for Colored-MNIST of our paper are $0.891$, $0.164$, and $-0.849$,** **which is aligned with our formulation** of $\alpha>0$, $\alpha\approx0$, and $\alpha<0$ for the three scenarios. For our language generation tasks, $\hat{\alpha}(\Psi\_{\mathcal{T}\_{HP1}}^*, \Psi\_{\mathcal{T}\_{HP2}}^*)=0.498$ and $\hat{\alpha}(\Psi\_{\mathcal{T}\_{HP1}}^*, \Psi\_{\mathcal{T}\_{PP}}^*)=0.239$, which indicates that HP1 is more aligned with HP2 than PP.
>
> Then, our Theorems 1 and 2 indicate that by trying some large $\lambda>1-\alpha$, the merged model can perform well on both two tasks if $\hat{\alpha}\geq 0$. By using some $\lambda\leq 0$, the merged model can forget $\mathcal{T}_2$ while maintain the performance on $\mathcal{T}_1$ if $\hat{\alpha}\leq 0$.
>
> ---------------------
>
> **Q4**: typoes.
>
> **A4**: Thank you for pointing them out. We have corrected them in the updated submission.

---

> > ### Author Response · Authors · 2024-11-28
> > **Thank you!**
> >
> > Dear Reviewer 3a5G,
> >
> > Thank you for raising the score from 6 to 8. We are encouraged by your support.
> >
> > Authors

---

### Author Response · Authors · 2024-11-27
**Update and Summary of Revisions**

Dear Reviewers/AC/SAC/PC,

We appreciate the evaluation and suggestions and the discussion with you. We uploaded a revision of our manuscript according to the review. In this revision, in addition to the changes discussed in the previous response, we also added the result of the correlation between language datasets HP1 and HP2 and between HP1 and PP as promised earlier to Reviewers 3a5G and 3aPS in the response and the manuscript. This shows that HP1 is more aligned with HP2 than PP, which verifies our experimental results in Section 4.2.

We summarize major changes in the revision during the rebuttal process.

1. **More discussion and experiments**: We have improved the discussion of our language model experiment in lines 507-509 and added the language generation experiment using Phi-3-small (7B) in Appendix A and an illustration of the results of the less-aligned task in footnote 4 as suggested by Reviewer 3aPS. We have added footnote 1 to introduce the empirical motivation of the data formulation and Appendix D to discuss the extension to the multi-classification case as suggested by Reviewer L24R.

2. **Presentation**: We have corrected Equation 4, Equation 11, and Definition 2 and included Table 6 of key notations in Appendix B as suggested by Reviewers 3a5G, 3aPS, and QzER.

3. **Typos**: We have corrected typos as mentioned by Reviewer 3a5G and 3aPS.

Thanks,

Authors

---

### Meta-Review · Area_Chair_hYmC · 2024-12-19

**Metareview:**

The submission theoretically analyses task vector arithmetic in transformers, demonstrating the extent to which multi-task learning, unlearning, and out-of-distribution generalisation can be achieved. The theoretical analysis is corroborated by experiments. All reviewers were excited by the theoretical results—the first such results developed in the context of task vectors in transformers. The additional experiments were well received, and the overall presentation of the work was praised. The reviewers did not point out any significant weaknesses in the work.

**Additional Comments On Reviewer Discussion:**

Due to the generally positive reviews, there was not much discussion required for this paper. The authors rebuttal was well-received and several reviewers increased their score.

---

### Decision · Program_Chairs · 2025-01-22

Accept (Oral)